

# An exploratory study on the aerosol height retrieval from OMI measurements of the 477 nm $O_2-O_2$ spectral band, using a Neural Network approach

Julien Chimot[1], Joris Pepijn Veefkind[1,2], Tim Vlemmix[1], Johan de Haan[2], Vassilis Amiridis[3], Emmanouil Proestakis[3,4], Eleni Marinou[3,5], and Pieternel Felicitas Levelt[1,2]

[1]Department of Geoscience and Remote Sensing (GRS), Civil Engineering and Geosciences, TU Delft, the Netherlands
[2]Royal Netherlands Meteorological Institute, De Bilt, the Netherlands
[3]Institute for Astronomy, Astrophysics, Space Applications and Remote Sensing, National Observatory of Athens, Athens 15236, Greece
[4]Laboratory of Atmospheric Physics, Department of Physics, University of Patras, 26500, Greece
[5]Department of Physics, Aristotle University of Thessaloniki, Thessaloniki, 54124, Greece

*Correspondence to:* J. Chimot (j.j.chimot@tudelft.nl)

**Abstract.** This paper presents an exploratory study on the retrieval of aerosol layer height (ALH) from the OMI 477 nm $O_2-O_2$ spectral band. We have developed algorithms based on the Multilayer Perceptron (MLP) Neural Network (NN) approach and applied them on 3-year (2005-2007) OMI cloud-free scenes over North-East Asia, collocated with MODIS-Aqua aerosol product. In addition to the importance of aerosol altitude for climate and air quality objectives, the main motivation of

5     this study is to evaluate the possibility of retrieving ALH for potential future improvements of trace gas retrievals (e.g. $NO_2$, $HCHO$, $SO_2$, etc..) from UV-Vis air quality satellite measurements over scenes including high aerosol concentrations. ALH retrieval relies on the analysis of the $O_2-O_2$ slant column density (SCD) and requires an accurate knowledge of the aerosol optical thickness $\tau$. Using the MODIS-Aqua aerosol optical thickness at 550 nm as a prior information, comparison with the LIdar climatology of vertical Aerosol Structure for space-based lidar simulation (LIVAS) shows that ALH average biases over

10     scenes with MODIS $\tau \geq 1.0$ are in the range of 260-800 m. These results depend on the assumed aerosol single scattering albedo (sensitivity up to 600 m) and the chosen surface albedo (variation less than 200 m). Scenes with $\tau \leq 0.5$ are expected to show too large biases due to the little impacts of particles on the $O_2-O_2$ SCD changes. In addition, NN algorithms also enable aerosol optical thickness retrieval by exploring the OMI reflectance in the continuum. Comparisons with collocated MODIS-Aqua show agreements between $-0.02 \pm 0.45$ and $-0.18 \pm 0.24$ depending on the season. Improvements may be

15     obtained from a better knowledge of the surface albedo, and higher accuracy of the aerosol model. This study shows the first encouraging aerosol layer height retrieval results over land from satellite observations of the 477 nm $O_2-O_2$ spectral band.

## 1 Introduction

The ability to monitor air quality and climate from UltraViolet-Visible (UV-Vis) satellite spectral measurements requires accurate trace gas (e.g. $NO_2$, $SO_2$, $HCHO$, $O_3$) and aerosol observations. Aerosols and trace gases often share similar anthro-





pogenic sources sources, and their concentrations, as shown by the satellite observations, often exhibit significant correlations (Veefkind et al., 2011). The reason is that trace gases are often precursors for aerosols. The importance of measuring vertical distribution of atmospheric aerosols on a global scale is triple. Firstly, aerosols directly impact the radiation budget of the Earth-atmosphere system through the scattering and absorption of solar and terrestrial radiation (Feingold et al., 1999). High

concentrations of fine particles lead to reduced clouds droplet size, enhanced cloud reflectance (Twomey et al., 1984), and reduced precipitation (Rosenfeld, 2000; Ramanathan et al., 2001; Rosenfeld et al., 2002). Therefore, large uncertainties of aerosol optical properties limit our climate predictive capabilities (IPCC: Solomon et al., 2007). In spite of more robust climate predictions in the last years, radiative forcing (RF) induced by aerosols still contributes to the largest uncertainty to the total RF estimate (IPCC: The Core Writing Team Pachauri and Meyer, 2014). The vertical distribution and relative location are

determining factors of aerosol radiative forcing in the long-wave spectral range (Dufresne et al., 2002; Kaufman et al., 2002). Secondly, aerosols play a significant role in air quality, in particular near the surface. Due to the rapid growth of both population and economic activity, such as in Asian region, increase in fossil fuel emissions gives rise to concerns about fine particles formation and dispersion. Aerosols include a variety of hazardous organic and inorganic substances, reduce visibility, lead to reductions in crop productivity and strongly affect health of inhabitants in urban regions (Chameides et al., 1999; Prospero,

1999; Eck et al., 2005).

Thirdly, slant column densities (SCD) of trace gases, derived from UV-Vis air quality space-borne sensors, have a high sensitivity to aerosol heights. For partly cloudy conditions, clouds are the main error source of trace gas measurements. But, in the absence of clouds, vertical distribution of aerosols, combined with their scattering and absorbing properties, modifies the length of the average light path of the detected photons, and therefore affects the computation of trace gas Air Mass Factor

(AMF). The application of the AMF is crucial for the conversion of slant column densities (SCD) from satellite line-of-sight measurements into vertical column densities. Then, aerosols strongly contribute to the uncertainties of trace gas retrievals from space-borne observations. For example, uncertainties in the computed tropospheric $NO_2$ AMF for the Ozone Monitoring Instrument (OMI) are the dominant source of errors in the retrieved tropospheric $NO_2$ column over polluted areas (Boersma et al., 2007). Negative biases on OMI tropospheric $NO_2$ columns, between $-26\%$ and $-50\%$, are found in urban and very

polluted areas in cases of high aerosol pollution and particles located at elevated altitude (Shaiganfar et al., 2011; Ma et al., 2013; Kanaya et al., 2014). HCHO AMF for GOME-2 and SCanning Imaging Absorption spectroMeter for Atmospheric CHartographY (SCIAMACHY) shows about 20-50% sensitivity to aerosols, depending if they are located within or above the boundary layer (Barkley et al., 2012; Hewson et al., 2015). Dust aerosols (large particles, with strong absorption in UV) can reduce the AMF in the $SO_2$ wavelengths (310 - 330 nm) by half, thus doubling the retrieved $SO_2$ (Krotkov et al., 2008).

This impacts the ability of sensors like OMI to monitor Planetary Boundary Layer (PBL) $SO_2$ with a sensitivity to local anthropogenic sources. Over regions of enhanced columns, aerosols highly contribute to the total $SO_2$ AMF error (Lee et al., 2009). Therefore, aerosol parameters (or retrievals) are a pre-requisite before retrieving trace gas vertical column densities.

State-of-the-art trace gas retrieval algorithms correct for aerosol effects either explicitly using modeled aerosol vertical profiles (Barkley et al., 2012, 2013; Kuhlmann et al., 2015; Lin et al., 2014, 2015), or alternatively implicitly via cloud algorithms.

For example, the OMI $O_2-O_2$ spectral band at 477 nm has been widely exploited to derive cloud information (Acarreta et al.,





2004; Sneep et al., 2008). However, cloud algorithm is sensitive to aerosols, and thus the retrieved effective cloud parameters are modified in their presence (Boersma et al., 2007; Castellanos et al., 2015; Chimot et al., 2016). The OMI $O_2-O_2$ spectral band at 477 nm contains significant information on aerosol properties and height. The retrieved effective clouds are then used to correct the computed AMF (de Smedt et al., 2008; Boersma et al., 2011). In spite of these well considered perturbations,

the use of the effective cloud parameters, assuming that the opaque Lambertian cloud model can reproduce the distribution of scattering fine particle effects, does not yet completely correct for the aerosol effects when computing the AMF, in particular for the tropospheric $NO_2$ columns (Castellanos et al., 2015; Chimot et al., 2016).

Characterizing the aerosol vertical distribution, in addition to the associated optical properties, using passive space-borne measurements is challenging due to the absence of spectral features in the aerosol optical properties and the combined influ-

ences of surface and cloud reflection. Contrary to effective cloud retrievals, aerosol retrieval is a more complex problem mainly because of the variability of particle microphysical properties and the lower optical thickness (typically 1-2 orders of magnitude). As a consequence, methods assuming large multiple scattering contributions, such as a simple cloud model assuming Lambertian properties, cannot be used. Passive radiometers like Moderate Resolution Imaging Spectroradiometer (MODIS) can only retrieve a limited amount of independent information from their measurements, usually aerosol optical thickness $\tau$

and the extinction Ångström exponent $\alpha$, as a proxy for the particle size distribution (Levy et al., 2007, 2013). The near-UV technique has been widely used to map the daily global distribution of UV-absorbing aerosols such as desert dust particles as well as carbonaceous aerosols generated by anthropogenic biomass burning and wildfires. It allows to retrieve $\tau$, Single Scattering Albedo $\omega_0$ and the qualitative Aerosol Absorbing Index (AAI) in the 330-388 nm of the Total Ozone Mapping Spectrometer (TOMS) and OMI sensors (Torres et al., 1998; Torres et al., 2002; Torres et al., 2007). However, this technique

is highly affected by the dependency of the measured radiances on the height of the absorbing aerosol layer (de Graaf et al., 2005). The Cloud-Aerosol Lidar with Orthogonal Polarization (CALIOP) has been providing vertical profiles of aerosols but with limited spatial coverage because of its measurements characteristics (Omar et al., 2009). Park et al. (2016) evaluated the sensitivity of the $O_2-O_2$ slant column density to changes in aerosol layer height over ocean. It is demonstrated that the $O_2-O_2$ spectral band at 477 nm is the most sensitive to the aerosol layer effective height (compared to the $O_2-O_2$ absorption bands

at 340, 360 and 380 nm), due to the largest $O_2-O_2$ absorption and reduced Rayleigh scattering. Veihelmann et al. (2007) determined that the OMI UV-Vis reflectance measurements contain between 2 and 4 Degrees of Freedom of Signal (DFS). The 477 nm $O_2-O_2$ band adds more information than any other individual band. This relative large number of DFS for UV-Vis satellite solar backscatter observations is explained by the sensitivity of the reflectance to the aerosol layer height. Detailed $O_2-O_2$ radiative transfer simulations performed by Dirksen et al. (2009) revealed the availability of the altitude information

about smoke aerosol plume, released by intense forest fires and transported over long distance, under specific conditions: high AAI and no clouds. In spite of all these efforts, no aerosol height retrieval has been done at this moment from $O_2-O_2$ satellite measurements at 477 nm over land.

Since aerosol altitude, in addition to aerosol optical thickness $\tau$, is one of the key parameters affecting the computation of AMF for trace gases retrievals such as $NO_2$ (Leitão et al., 2010; Chimot et al., 2016), the motivation of this exploratory

study is to evaluate the capability of retrieving it from the satellite $O_2-O_2$ absorption band at 477 nm. This study follows





the conclusions of previous works focused on the sensitivity of this spectral band and the observed links between the $O_2-O_2$ effective cloud retrievals and aerosol parameters. In this paper, quite a few algorithm concepts are developed, based on the Neural Network (NN) approach, and then tested on a high number of OMI observations over land. The primary focus of this exploratory study is the retrieval performance of aerosol layer pressure (ALP) associated with scattering and fine particles over

large urban, industrialized and highly polluted area and cloud-free scenes. In addition, the sensitivity of the algorithms to $\tau$ knowledge is investigated and, therefore, the capability of $\tau$ retrievals from the same OMI band is evaluated. The considered OMI observations are described in Sect. 2, with a particular emphasize on the available $O_2-O_2$ Differential Optical Absorption Spectroscopy (DOAS) parameters and their link with ALH and $\tau$. The development of the different NN algorithms are described in Sect. 3. Their performances are evaluated in Sect. 4 on synthetic and independent data set with a characterization

of the main limiting factors. Finally, these algorithms are applied to 3 years (2005-2007) of cloud-free OMI observation over the North-East Asia, where large amounts of aerosols are emitted from both natural and anthropogenic sources (Lee et al., 2012). They are then compared with other observation products, namely MODIS Aqua $\tau$ and the LIdar climatology of vertical Aerosol Structure for space-based lidar simulation (LIVAS). This allows to address the potential expectation of $O_2-O_2$ aerosol layer height (ALH) retrievals for future AMF computations.

## 2 OMI $O_2-O_2$ DOAS analysis and aerosols

### 2.1 OMI satellite data

The Dutch-Finnish mission OMI (Levelt et al., 2006) is a nadir-viewing push-broom imaging spectrometer launched on the National Aeronautics and Space Administration (NASA) Earth Observing System (EOS)-Aura satellite. It provides daily global coverage of key air quality components through observations of the backscattered solar radiation that are captured during

daylight in the UV-Vis spectral domain. Based on a two dimensional detector array concept, radiance spectra are simultaneously measured on a 2600 km wide swath within a nadir pixel size of 13x24 km$^2$ (28 x 150 km$^2$ at extreme off-nadir). OMI has a higher spatial resolution than any other UV-Vis hyperspectral spectrometers. It measures in the wavelength range of 270 nm to 500 nm with a spectral resolution of 0.45 nm in the UV-2 band (310-360 nm) and 0.63 nm in the visible band (360-500 nm). Retrieval of atmospheric parameters are usually obtained from the reflectance spectrum defined as:

$$R(\lambda) = \frac{\pi \cdot E(\lambda)}{cos(\theta_o) \cdot F(\lambda)}, \tag{1}$$

with $\lambda$ the wavelength, $E(\lambda)$ and $F(\lambda)$, the radiance and the solar irradiance respectively, and $\theta_o$ the solar zenith angle.

Starting mid-2007, an external obstruction to the sensor's field of view has been perturbing OMI measurements of the Earthshine radiance at all the wavelengths. At this moment, about half of the sensor's sixty viewing positions are affected by this so-called "row anomaly", referring to the row numbers of the viewing positions on the CCD detectors. Details on the onset and

progression of this anomaly are given on the site http://www.knmi.nl/omi/research/product/rowanomaly-background.php. For practical reasons, this study only used the OMI data acquired during 2005-2007, i.e. before the development of this anomaly.





OMI has not been optimized for aerosol monitoring. However, various studies demonstrated the sensitivity of the reflectance measurements to aerosol properties in the UV. The main advantage of this spectral domain is the reduced impacts due to uncertainties in surface reflectance, partly due to the low values of both ocean and land surface albedo in the UV (Torres et al., 1998). Absorbing aerosols are detected and distinguished from the other aerosol types by analyzing their spectral contrast to

Rayleigh scattering. The OMI near-UV aerosol algorithm (OMAERUV) independently retrieves atmospheric total columns of aerosol optical thickness $\tau$ and single scattering albedo $\omega_0$ from the 2 UV wavelengths, 354 nm and 388 nm (Torres et al., 2007, 2013). The OMI multi-wavelength algorithm (OMAERO) uses up to 19 wavelength bands between 331 nm and 500 nm to derive aerosol characteristics (i.e. AOD, aerosol type, AAI as well as ancillary information) from the OMI cloud-free spectral reflectance, based on the best fitting aerosol type models provided. Three main aerosol types are considered: dust, carboneceous

aerosol associated with biomass burning, or weakly absorbing sulfate based aerosol. The wavelengths were selected such that they are essentially free from strong Raman scattering. The $O_2-O_2$ collision complex at 477 nm is included to enhance the sensitivity to the aerosol layer height (Veihelmann et al., 2007).

In comparison with 44 Aerosol Robotics Network (AERONET) sites, evaluated OMAERUV $\tau$ yield a root mean square error (RMSE) of 0.16 and a correlation coefficient of 0.81 over the years 2005-2008 (Ahn et al., 2014). About 65 % of these

retrievals lie within the expected uncertainty. The OMAERUV SSA products agree with AERONET to within 0.03 in 46,% of the collocated pairs, and to within 0.05 in 69 % of the cases (Jethva et al., 2014). Park et al. (2016) used a Look-Up-Table (LUT) approach to retrieve, from the OMI $O_2-O_2$ 477 nm spectral band, aerosol effective height over ocean, close to East Asia, within the error range of 1 km (compared to CALIOP). This approach was applied to 7 case studies, each of them covering a few days. No aerosol optical thickness was retrieved. No study has yet explicitly used this satellite band to directly

retrieve aerosol layer height and $\tau$ over land.

## 2.2 DOAS analysis of the OMI $O_2-O_2$ 477 nm spectral band

The various DOAS techniques rely on the same key concept: a simultaneous extraction of several trace gas slant column densities from the fine spectral features (i.e. the high frequency part) present in passive UV-Vis spectral measurements of atmospheric radiation (Platt and Stutz, 2008). The OMI $O_2-O_2$ cloud algorithm exploits the 460-490 nm spectral band, focused

on the $O_2-O_2$ absorption line at 477 nm. The initial purpose of this algorithm is the retrieval of cloud parameters, namely effective cloud fraction and effective cloud pressure (Acarreta et al., 2004; Veefkind et al., 2016). The first step of this algorithm is a DOAS spectral fit in which the absorption cross-section spectrum of $O_2-O_2$ is fitted together with a first order polynomial:

$$-\ln(R(\lambda)) = \gamma_1 + \gamma_2 \cdot \lambda + N^s_{O_2-O_2}(\lambda) \cdot \sigma_{O_2-O_2} + N^s_{O_3}(\lambda) \cdot \sigma_{O_3}, \tag{2}$$

where $\gamma_1 + \gamma_2 \cdot \lambda$ defines the first order polynomial, $\sigma_{O_2-O_2}$ and $\sigma_{O_3}$ are the $O_2-O_2$ absorption cross-section spectrum

(at 253 K) and the $O_3$ absorption cross section spectrum respectively, convoluted with the OMI slit function, $N^s_{O_3}$ is the $O_3$ slant column density and $N^s_{O_2-O_2}$ is the $O_2-O_2$ slant column density. The $O_3$ cross section spectrum is included because it overlaps with the $O_2-O_2$ spectrum. The fitted parameters are $\gamma_1$, $\gamma_2$, $N^s_{O_2-O_2}$, and $N^s_{O_3}$. In the absence of absorbers, one may




define the continuum reflectance $R_c$ at the reference wavelength $\lambda_0$:

$$R_c = \exp(-\gamma_1 - \gamma_2 \cdot \lambda_0). \tag{3}$$

The reference wavelength is specified as the middle of the DOAS fit window at $\lambda_0 = 475\,\mathrm{nm}$.

The OMI cloud algorithm employs, in a second step, a LUT to convert the retrieved $N_{O_2-O_2}^s$ and $R_c$ into effective cloud
parameters. In this paper, the aerosol retrieval algorithms differ by replacing this second step by a Neural Network (NN)
approach, described in Sect. 3. This approach allows the conversion of the DOAS fit parameters into $\tau$ and ALP (in hPa). As a
consequence, the NN retrievals rely on the way how these 2 aerosol parameters modify the $O_2-O_2$ DOAS variables and thus
the photons average light path.

## 2.3  On the impact of aerosols on $R_c$ and $O_2-O_2$ SCD

The feasibility to exploit the UV and Vis spectral bands to derive aerosol information has been widely investigated the last 15
years by the ground-based Multi-Axis Differential Optical Absorption Spectroscopy (MAX-DOAS) community. Three types
of passive UV-Vis spectral measurements can be related to the presence of aerosols: radiance measurements (Frieß et al.,
2006), measurements of absorption by the $O_2-O_2$ complex (Wagner et al., 2004; Vlemmix et al., 2010), and Ring absorption
measurements (Wagner et al., 2009). The sensitivity to aerosols and their vertical distribution is different for these three types
of measurements, due to differences in the underlying physical processes.

Figure 1 illustrates how aerosol particles directly drive the OMI $O_2-O_2$ DOAS parameters at 477 nm assuming cloud-free
space-borne observations. These effects are obtained from radiative transfer simulations including aerosols and no clouds. The
detailed generation of such simulations is given in Sect. 3.2. The DOAS fit equations following Eq. (3) and Eq. (4) are then
applied on these simulations. In this paper, the aerosol layer height (ALH) is expressed by the aerosol layer pressure (ALP), in
hPa, defined as the mid-pressure of an homogeneous and finite aerosol layer.

The OMI continuum reflectance at 475 nm $R_c$ is directly affected by the total column $\tau$ of fine particles present in the
observed scene as well as their optical properties (e.g. single scattering albedo $\omega_0$). Indeed, reflectance values increase with
increasing $\tau$ independently of the ALP (cf. Fig. 1a). This mostly results from the influence of aerosols on the ensemble of
detected photons and on the additional scattering effects observed in the scene compared to an aerosol-free scene. However,
$R_c$ does not only depend on the total $\tau$, but results as well from the optical properties of the particles, described for instance
by $\omega_0$ and the phase function, and the surface albedo $A$. The importance of these parameters is further discussed in Sect. 4 and
Sect. 5. In addition, the reflectance is also driven by the geometry angles: i.e. viewing zenith angles $\theta$, solar zenith angles $\theta_0$
and relative azimuth angle $\phi - \phi_0$. An increase of $\theta$ or $\theta_0$ will lead to longer average light path, and thus will amplify aerosol
related additional scattering effects (for a given $\tau$).

$O_2-O_2$ absorption measurements determine the shielding or enhancement of photons by the $O_2-O_2$ complex in the visible
spectral range, due to the presence of particles. $O_2-O_2$ absorption increases with increasing path lengths. From that per-
spective, the aerosol layer height directly influences the $O_2-O_2$ absorption since particles at higher altitude partly shield the
$O_2-O_2$ complex located below (i.e. decrease of the length of the average light path). Such an effect has a direct impact on the





$O_2-O_2$ slant column density (SCD) $N^s_{O_2-O_2}$ values. As observed in Fig. 1 and also reported by Wagner et al. (2004); Park et al. (2016); Chimot et al. (2016), $N^s_{O_2-O_2}$ generally increases with increasing aerosol layer pressure (or decreasing aerosol layer height).

However, as depicted in Fig. 1c, not only ALP but also $\tau$ influences the slant $O_2-O_2$ absorption since both parameters simultaneously affect the photon path distribution. An increase of $\tau$, for particles with $\omega0 \leq 0.95$, leads to a decrease of $N^s_{O_2-O_2}$. The slope of this decrease depends on the aerosol altitude (i.e. higher for particles at high altitude). This means that $\tau$ also drives the $O_2-O_2$ shielding effect, while the aerosol height does not significantly affect $R_c$. As a consequence:

- the 2 parameters $R_c$ and $N^s_{O_2-O_2}$ have a non negligible correlation (see Fig. 1b) as they share, in part, same information: the length of the average light path and the aerosol optical thickness $\tau$. They are not two independent variables and combining them does not provide with two independent pieces of aerosol information.

- the variable $N^s_{O_2-O_2}$ contain both information, $\tau$ and ALP that cannot be easily separated. As a consequence, if $\tau$ is not accurately known, there will likely be an ambiguity when analysing $N^s_{O_2-O_2}$ to retrieve ALP since contributions from both aerosol height and absorption or scattering of the sunlight due to aerosol amounts cannot be separated.

Moreover, potential uncertainties in surface albedo and aerosol optical properties may amplify this ambiguity and thus make the retrieval problem more complex. To avoid these constraints, it may be then relevant to test the ALP retrievals by replacing $R_c$ with a prior $\tau$, provided this information is known with a high enough accuracy (e.g. MODIS $\tau$).

Overall, the impact of aerosol particles on the OMI $O_2-O_2$ spectral band is similar to cloud particles. This explains in part the difficulty to distinguish aerosols from clouds. In cases with a mix of aerosols and clouds, there is an ambiguity between $R_c$, the $\tau$ and the OMI effective cloud fraction on the one hand, and $N^s_{O_2-O_2}$, the aerosol layer pressure, $\tau$, the OMI effective cloud pressure and fraction on the other hand (Boersma et al., 2011; Castellanos et al., 2015; Chimot et al., 2016). Therefore, this study only focuses on cloud-free reflectance to avoid this complexity.

## 3 Design of the neural network retrieval algorithms

The retrieval algorithms developed for this paper are based from the Pybrain software (Schaul et al., 2010). Pybrain is a versatile machine learning library written in Python designed to facilitate both the application of and research on premier learning algorithms such as recurrent NNs. It includes several functions such as supervised learning algorithms, feed forward network design and error back-propagation computations. Only the main developments specific to the present study are explained and discussed in the next sub-sections. For more details related to the Pybrain specificities, the reader is encouraged to read Schaul et al. (2010).

The Multilayer Perceptron (MLP) neural networks have been widely used and acknowledged for decades in the field of remote sensing (Atkinson and Tatnall, 1997). Indeed, most retrieval problems in this field are ill-posed and nonlinear, in particular the atmospheric ones. Thus, the associated inverse problems can only be addressed by including a priori information and relying on statistical analysis. Since aerosol retrieval from passive spectral measurements is well-known as a non-linear



inverse problem, the MLP technique represents then a powerful approach to design a retrieval algorithm in a fast and robust way. The basic idea is to build an optimal interpolator system making the link between the OMI 477 nm $O_2-O_2$ measurements and the retrieved aerosol layer pressure and optical thickness (i.e. between the ellipses of Fig. 1). However, knowledge must be acquired by the network by means of a supervision database. The following sections summarize then the design of the

developed algorithms (see Sect. 3.1), the generated supervision database (see Sect. 3.2) and the employed learning process (Sect. 3.3).

### 3.1 Multilayer Perceptron network approach: application to the OMI $O_2-O_2$ aerosol retrievals

Artificial neural Networks are a family of models related to the machine learning and the artificial intelligence domain (Luger and Stubblefield, 1998). They are used to reduce the number of calculations of functions requiring a large number of inputs and

being generally unknown (or not well defined). The idea is to approximate them by parameterized and more simple functions. Input and output signals are then interconnected by a set of activation functions and a set of weights associated with each of them (Luger and Stubblefield, 1998). In the context of this work, the link between the OMI $O_2-O_2$ measurements and aerosol parameters cannot be exactly described with accurate functions (see Sect. 2.3). Thus, the idea of developing neural networks here is to identify input-output relationships directly from a well-known training ensemble. The choice of a NN approach relies

on their advantages compared to more conventional methods such as linear regression, linear interpolation in a LUT or the Optimal Estimation Method (OEM). In particular, it enables 1) very fast computations with modern computers in spite of the number of required parameters, 2) optimized interpolation technique even in case of non linear statistical modelling and so potentially lower systematic biases compared to a linear interpolation, 3) reduced memory use compared to a LUT with a very high sampling.

As illustrated in Fig. 2, the designed NNs rely on a multi-layer architecture, based on the Multilayer Perceptron technique (Rumelhart et al., 1986), composed of parallel processors (i.e. neurons) organized in distinct layers. Such an architecture allows to separate non-linear data, and generally consists of 3 (or more) types of layers. The first layer includes all the required input variables. The last layer includes all the desired output data (or here retrievals). The intermediate layers are usually referred as hidden layers and contain the activation functions. All these layers are connected via neural links: two nodes or neurons $i$ and

$j$ between two consecutive layers have synaptic connections associated with a synaptic weight $\omega_{ij}$. Each neuron $j$ computes a weighted sum of its $N$ $x_i$ information sent from the neurons of the previous layer. Then, this weighted sum is transported through a non-linear mathematical function before being passed to the next layer. Here use is made of the classical sigmoid function:

$$\sigma(x) = \frac{1}{1 + \exp(-x)}. \tag{4}$$

The output $z_j$ of the neuron $j$ in the hidden layer is thus given by:

$$z_j = \sigma(\sum_{i=1}^{N} \omega_{ij} \cdot x_i). \tag{5}$$



The overall set $W$ of synaptic weights $\omega_{ij}$ contain all the information about the network (i.e. its neural architecture defined by a specified number of layers, neurons and connections). When the transport reaches the output layer, it forms the network output.

The chosen neural architecture is the following (see Fig. 2). The input layer is composed of 7 parameters that include (1) $\theta$, $\theta_0$, $\phi - \phi_0$, surface pressure, surface albedo, and (2):

- either $R_c$ and $N^s_{O_2-O_2}$ for $\tau$ and ALP retrievals: this configuration is named $NN_{R_c, N^s_{O_2-O_2}}$,

- or $\tau$ and $N^s_{O_2-O_2}$ for ALP retrieval: this configuration is named $NN_{\tau, N^s_{O_2-O_2}}$.

The output layer is, for each NN retrieval algorithm, composed of only one output variable: either $\tau$, or aerosol layer pressure $ALP$. In total, three NN retrieval algorithms are then selected and used at the end: one $NN_{R_c, N^s_{O_2-O_2}}$ for $\tau$ retrieval, one $NN_{R_c, N^s_{O_2-O_2}}$ and one $NN_{\tau, N^s_{O_2-O_2}}$ for $ALP$ retrieval.

The choice to use either $NN_{R_c, N^s_{O_2-O_2}}$ or $NN_{\tau, N^s_{O_2-O_2}}$ will impact the accuracy of the $ALP$ retrieval results (see Sect. 4.2, 5.2 and 5.3).

### 3.2 Generation of the supervision database: aerosol properties and simulations

The MLP neural networks must be trained in order to build models that can learn from a well-known data set, and then generalize the inverse problem by predicting the aerosol retrievals from input observations that have never been seen before. For that purpose, a learning database must be carefully designed and generated. It must be representative of the entire distribution of (input-output) values that can likely be encountered in the OMI observations. As a consequence for the MLP algorithms, a large quantity of data is often required for the learning process. However, very large learning data set can be extremely time consuming in terms of generation and then NN training.

The NNs are here trained with simulated data. This study uses the Determining Instrument Specifications and Analyzing Methods for Atmospheric Retrieval (DISAMAR) software of KNMI (de Haan, 2011). DISAMAR includes a radiative transfer model and different retrieval methods. The radiative transfer model is based on the Doubling Adding KNMI (DAK) model (de Haan et al., 1987; Stammes, 2001) and thus computes the reflectance and transmittance in the atmosphere using the adding/doubling method. This method calculates the internal radiation field in the atmosphere at levels to be specified by the user and takes into account Rayleigh, aerosol and cloud scattering and trace gas and aerosol absorption. Scattering by aerosols is simulated with a Henyey–Greenstein scattering phase function $\Phi(\Theta)$ (Hovenier and Hage, 1989):

$$\Phi(\Theta) = \frac{1-g^2}{(1+g^2-2g\cos\Theta)^{3/2}}. \tag{6}$$

where $\Theta$ is the scattering angle. The phase function is then parameterized by the asymmetry parameter $g$, which is the average of the cosine of the scattering angle, such its variation $-1 \leq g \leq 1$ ranges from back-scattering through isotropic scattering to forward scattering. Following the DISAMAR configuration, $\tau$ values in the simulations are specified at the referent wavelength of 550 nm. The Ångström exponent $\alpha$ describes the spectral dependence of $\tau$.



ALP is the main target parameter since this is one of the main parameters describing the average light path distribution in the tropospheric $NO_2$ AMF computation. The second target is $\tau$ since this information may be requested for a good ALP retrieval quality. We thus assume we do not need at this level to define more realistic aerosol models for every aerosol scene. With a referent asymmetry parameter of $g = 0.7$, intermediate value typically observed (Dubovik et al., 2002), the Henyey-

Greenstein function is known to be smooth and reasonably well reproduce the Mie scattering functions for most of aerosol types. This approach is also used for the preparation of the operational aerosol layer height retrieval algorithm from Sentinel-5 Precursor (Sanders et al., 2015) and for explicit aerosol corrections in the AMF calculation when retrieving trace gases such as tropospheric $NO_2$ (Spada et al., 2006; Wagner et al., 2007; Castellanos et al., 2015).

The ensemble of parameters and associated values used for generating the learning database is detailed in Table 1. About 460

000 spectral simulations, over the $O_2-O_2$ spectral band (460-490 nm), were generated, assuming different satellite viewing and solar geometries, surface albedo, surface pressure and aerosol pollution levels. Scenes with too large angles (i.e. $\theta_0 \geq 65°$) and too bright surfaces (i.e. $A > 0.1$) are excluded. For each of these simulations, $R_c$ and $N^s_{O_2-O_2}$ were deduced from the DOAS fit equations Eq. (2) and Eq. (3). Aerosols are specified for a standard case, assuming fine particles with a unique value of Ångström coefficient $\alpha = 1.5$ and $g = 0.7$. Aerosol profiles are parameterized by scattering layers with constant aerosol

volume extinction coefficient and aerosol single scattering and with a fixed pressure thickness. The ALP corresponds to the mid-pressure of the assumed scattering aerosol layer. In order to investigate the assumptions related to the single scattering albedo properties $\omega_0$, two typical values are considered: $\omega_0 = 0.95$ and 0.9. Contrary to the other variables, $\omega_0$ is not known for each OMI pixel and thus cannot be used as an explicit input parameter by the designed NNs. Moreover, it cannot be retrieved from this band since it is supposed to affect $R_c$ and $N^s_{O_2-O_2}$, similarly as $\tau$. Therefore, 2 sets of NN models are developed for

different purposes:

– one set of three MLP NN algorithms ($NN_{R_c,N^s_{O_2-O_2}}$ for $\tau$ retrieval, $NN_{R_c,N^s_{O_2-O_2}}$ and $NN_{\tau,N^s_{O_2-O_2}}$ for $ALP$ retrieval) is trained assuming $\omega_0 = 0.95$;

– one set of three MLP NN algorithms ($NN_{R_c,N^s_{O_2-O_2}}$ for $\tau$ retrieval, $NN_{R_c,N^s_{O_2-O_2}}$ and $NN_{\tau,N^s_{O_2-O_2}}$ for $ALP$ retrieval) is trained assuming $\omega_0 = 0.9$.

The choice to use one of these sets will impact the accuracy of the retrieval results.

### 3.3 Optimization of the learning process and selection of the best NN architecture

Prediction of the optimal number of neurons and the hidden layers is generally not possible as these values are strongly specific to the given problem (Atkinson and Tatnall, 1997). While it has been theoretically proven that a single hidden layer MLP networks with non linear activation functions may represent any nonlinear continuous function (Haykin, 1999), a 2-hidden

layer MPL may approximate any function to any degree of non-linearity taking also into account discontinuities (Sontag, 1992). To identify the best NN architecture for each aerosol retrieval parameter ($\tau$ and ALP) and for each configuration ($NN_{R_c,N^s_{O_2-O_2}}$ or $NN_{\tau,N^s_{O_2-O_2}}$, see Sect. 3.1), several architectures are trained and then evaluated: one single hidden layer with a variable number of neurons (between 9 and 70) and two hidden layers (between 15 and 70 neurons on the 1st layer,





and between 10 and 40 on the 2nd layer). Then, the most relevant NN architecture is selected based on the best computed evaluation score. In total, about 96 different MLP architectures, for each configuration, were evaluated.

For one given NN architecture, the training process is the optimization technique that estimates the optimal network parameters $W$ of synaptic weights $\sigma_{ij}$ (see Sect. 3.1). For that purpose, a positive-definite error function $E$ must be minimized. This

error function measures, for a set of $p$ representative situations, for which inputs and outputs (i.e. $\tau$ and ALP) are known, the mismatch between the neural network outputs $o_i$ and the true outputs $t_i$ as follow:

$$E = \frac{1}{2} \sum_{i=1}^{P} (o_i - t_i)^2. \tag{7}$$

This error function minimization follows here the Error Back-Propagation algorithm as specified by Rumelhart et al. (1986). It is a stochastic steepest descent algorithm well adapted to the MLP hierarchical architecture. The learning step is made sample

by sample, iteratively and stochastically selected in the training data set. The network is initialized with random synaptic weights. At each iteration, the error is computed and then propagated backwards from the output layer. The processes of error back-propagating and feeding forward signals are repeated iteratively until the error function is minimized or the maximum number of iterations is reached (i.e. 500).

During the training phase, the considered network architecture must obtain an optimal generalization performance: i.e.

the network performance should not degrade significantly when data set other than the training one is analyzed. Standard NN architectures, like the fully connected MLP, generally have a too large parameter space, and are prone to over-fitting. Although the network performance seems to constantly improve on the training sets at each iteration, it can actually begin to worsen (in terms of errors) on unseen data set. Therefore, a verification step is performed, over the last 15 iterations, to detect this overfitting moment (i.e. no significant variation of $E$) and stop the training phase. This process is called early stopping.

Finally, to ensure that the system is not trapped in local minima during the cost function minimization, the learning phase (training+verification) is repeated three times, the synaptic weights being randomly initialized at the beginning of each training phase. The network system presenting the best evaluation score (see Eq. 7) is then selected. All these precautions are carried out by randomly splitting the learning data (see Tab. 1) into 3 independent sets: training, verification and evaluation. They respectively consisted of 70 %, 15 % and 15 %.

Figure 3 depicts the box-whisker plots of the square of errors obtained over the ensemble of training-verification-evaluation data set for $\tau$ and ALP retrieval and for $NN_{R_c, N^s_{O_2-O_2}}$ configuration, assuming aerosol $\omega_0$ of 0.95. For $\tau$ retrievals, although the NNs with 40 and 70 neurons on one hidden layer do reasonably well, the scores show improved values when two hidden layers are used. The ALP retrieval scores are significantly larger than for $\tau$ ( a factor of 100). This is a direct consequence that ALP is less well constrained by the spectral measurements: lower pieces information are available compared to $\tau$, in particular

for scenes dominated by low $\tau$ values. While the NNs with one hidden layer do not show any significant improvements when increasing number of neurons, better scores are only obtained with 2 hidden layers. Overall, the similar behavior of training-verification-evaluation scores validate that the trained NNs are generalized enough to be able to reproduce similar variation of the scores on other independent data set. The identified best NN algorithms are thus found with 2 hidden layers, with



between 25 and 70 neurons on the first layer, and between 10 and 20 neurons on the second layer depending on the considered configuration (see Sect. 2.3) and retrieved parameter.

## 4 Sensitivity analyses on synthetic data set

The robustness of the trained and selected NN retrieval algorithms is assessed by applying them to independent simulations, not present in the learning (training-verification-evaluation) database. Simulated reflectance are noise-free and only include aerosol particles (no clouds). The sensitivity of $\tau$ and ALP retrievals is verified for different surface albedo, aerosol properties ($\omega_0$, $\tau$, ALP). $R_c$ and $N_{O_2-O_2}^s$ are derived from the spectra and provided as inputs to the NNs. The impact of uncertainties on surface albedo, aerosol model and $N_{O_2-O_2}^s$ are analysed. Consistent geophysical conditions (temperature, $NO_2$ and $O_3$ profiles) are considered between these simulations and those included in the learning database. All the analyses performed here are summarized in Tab. 2.

### 4.1 Aerosol Optical Thickness retrievals

Figure 4 depicts the derived $\tau$ values from the simulated spectral measurements, compared with the true values, and how much inaccurate assumptions about the aerosol single scattering albedo $\omega_0$ or the asymmetry parameter $g$ degrade the retrieval quality. Figure 4a shows that retrieved and true $\tau$ are very well correlated for all the types of surface, assuming no error in the assumed surface albedo and aerosol properties. This confirms the success of the learning process implemented in Sect. 3.2 and Sect. 3.3 and the use of the NN approach.

The assumed aerosol properties (single scattering albedo $\omega_0$ and phase function through the asymmetry parameter $g$), and so the choice of the trained NN algorithm, are of high importance. They change the slope between retrieved and true $\tau$ values. If the true aerosol $\omega_0$ value is lower ($\omega_0 = 0.9$) than in the simulations of the learning database ($\omega_0 = 0.95$), retrieved $\tau$ is smaller. This is a direct result of reduced scattering efficiency and thus more photons are absorbed instead of being scattered back towards the satellite sensor (see Fig. 4a). The bias reaches a maximum value of about $0.8$ for $\tau = 2$, and is lower than $0.1$ for $\tau \leq 0.5$ (see Fig. 3b). This is directly related to the impact of aerosols and their optical properties on the measured $R_c$.

Figure 4b illustrates retrieved $\tau$ bias due to the difference between the assumed asymmetry parameter $g$ in the learning database and in the synthetic spectra. This gives the direct impact of uncertainties associated with the aerosol scattering phase function characterization. While $g = 0.7$ is the referent value for most of aerosols, scenes with $g = 0.6$ are related to finer and weakly absorbing particles with a somewhat reduced forward scattering direction such as carbonaceous aerosols, desert dust and volcanic dash models as given by the ESA aerosol CCI-project (de Leeuw et al., 2013). Values of $g = 0.8$ are associated with larger particles and an increased forward scattering direction such as cirrus (Sanders et al., 2015). An overestimation of $g$ (i.e. true $g = 0.6$ while assumed $g = 0.7$) leads to an increased retrieved $\tau$ value (i.e. positive bias) as a result of more photons scattered back towards the satellite sensor, and less towards the surface. Reciprocally, an underestimation of $g$ (i.e. true $g = 0.8$ while assumed $g = 0.7$) leads to a decreased retrieved $\tau$ value (i.e. negative bias) as a result of less photons scattered back towards the satellite sensor, and more towards the surface. Absolute bias values can exceed $0.5$ for $\tau = 1.5$ while they stay





close to 0.25 for $\tau = 0.5$. Overall, uncertainties in aerosol models can drastically impact the retrieved $\tau$ accuracy and modify the slope between retrieved and true $\tau$ values

Figure 5a depicts the box-whisker distribution of $\tau$ biases due to uncertainties in surface albedo. Estimations are obtained for fine and scattering particles ($\alpha = 1.5$, $\omega_0 = 0.95$, $g = 0.7$). The $\tau$ uncertainty is obtained from the half of $\tau$ differences between adding and deducting uncertainties of the variables as follows:

$$\epsilon(\partial x) = \frac{1}{2} \mid \tau(x + \partial x) - \tau(x - \partial x) \mid . \tag{8}$$

where $\epsilon(\partial x)$ is the $\tau$ bias due to the variable error source $x$ (here surface albedo), $\partial x$ is the uncertainty applied to $x$. $\epsilon(\partial x)$ values are computed for all combinations of surface albedo [0.03-0.05-0.07] and $\theta_0$-$\theta = [25°$-$25°, 50°$-$25°, 25°$-$45°]$.

Errors in surface albedo lead to biases in derived $\tau$ values. Retrieved $\tau$ bias lies in the range of 0.04-0.48 on average, for surface albedo uncertainties $\partial x_{alb}$ between 0.005 and 0.05 (see Fig. 5a). To limit the $\tau$ bias values to an average of 0.2, $\partial x_{alb}$ should not exceed 0.025. Typical differences of climatological surface albedo between that obtained from the total ozone monitoring spectrometer (TOMS) and the global ozone monitoring experiement (GOME) (Koelemeijer et al., 2003), or between the Lambertian equivalent reflectance (LER) of OMI and the MODIS black sky albedo (Kleipool et al., 2008), are known to be up to 0.02. It was observed that surface albedo uncertainty mostly impacts cases with low $\tau$ (lower than 0.5), as emphasized as well by Park et al. (2016). The measured $R_c$ over scenes with low $\tau$ is a combination of aerosol scattering effects and surface reflection. But, in case of high $\tau$, aerosols strongly attenuate the surface contribution, and their scattering proprieties dominate. Differences in the standard deviation lie in the range of 0.07-0.17 depending on the surface albedo uncertainty.

## 4.2 Aerosol Layer Pressure retrievals

Figure 6 and Figure 7 depict the expected performances of the developed NN algorithms for Aerosol Layer Pressure (ALP) retrievals. For scenes dominated by $\tau$ in the range of 0.5-1.0, ALP retrievals are quite stable presenting biases close to 100 hPa. Only for $\tau \geq 1.0$, biases are smaller than 50 hPa. The accuracy of the retrieved ALP generally increases with increasing $\tau$. Indeed, assuming true ALP of 850 hPa and no error in the employed surface albedo and aerosol model (see Fig. 7), positive biases larger than 400 hPa are found for $\tau \leq 0.5$. Note that this behavior is observed for all the NN configurations ($NN_{R_c, N^s_{O_2 - O_2}}$ and $NN_{\tau, N^s_{O_2 - O_2}}$). A box-whisker plot, in Figure 8, illustrates the variability of the ALP biases as a function of $\tau$ over all the simulations contained in the entire learning database. This confirms that, in spite of the strict training-verification-evaluation process achieved in Sect. 3.3, the NN ALP retrievals are not expected to be accurate for small $\tau$ values, especially below 0.5. The reason is directly linked to the magnitude of the $O_2 - O_2$ shielding effect and its dependence on the aerosol amount (in addition to the aerosol altitude). Low amount of aerosols have very limited effects on the $O_2 - O_2$ absorption (see Fig. 1). Thus, even advanced interpolation techniques like NNs have difficulties for these cases. When $\tau$ increases, the $O_2 - O_2$ shielding effect amplifies and the algorithms are more able to link it to the ALP variation. Overall, even for small $\tau$ values (like 0.5), the retrieved aerosol pressures correlate with the aerosol layer height in spite of very poor accuracy (see Fig. 7a and Fig. 7c) .





The advantages of using $\tau$ information as input instead of $R_c(475\,\mathrm{nm})$ are multiple, provided this information is given with a high enough accuracy. Firstly, it allows to improve the accuracy of the ALP retrieval for cases dominated by low $\tau$ and particles located at high altitude (above 800 hPa or 2 km). Indeed, on Fig. 6b, for $\tau = 0.5$ and ALP between 750 and 850 hPa, biases are reduced from [250:350] to [150:250] hPa. For particles higher than 650 hPa (or 3.5 km), no improvements are

however observed. The low sensitivity to retrieve ALP when particles are located at a very high altitude is directly due to the $O_2-O_2$ complex and its vertical distribution. This was demonstrated by Park et al. (2016): $O_2-O_2$ concentration exponentially decreases with increasing atmospheric altitude.

Secondly, impacts due to uncertainties related to the chosen surface albedo and aerosol model are reduced. Assumptions on aerosol $\omega_0$ drive the interpretation of the shielding of the $O_2-O_2$ dimers by aerosols. If the assumed $\omega_0$ is too high (i.e. assumed

$\omega_0 = 0.95$ while $\omega_0$ true = 0.9), $ALP$ retrievals obtained with the $NN_{R_c,N^s_{O_2-O_2}}$ configuration are perturbed by 100 hPa (see Fig. 6a). These perturbations are reduced over scenes with high $\tau$ values (larger than 1) only for particles close to the surface, i.e. true $ALP \geq 850\mathrm{hPa}$ (see Fig. 7a and Fig. 7c). Using the $NN_{\tau,N^s_{O_2-O_2}}$ configuration, assuming true $\tau$ as input, helps to mitigate theses biases. The ALP retrievals are almost superposed to those with true $\omega_0$ assumptions and present the same behaviors with respect to the particles altitude and $\tau$ (see Fig. 6b). Similar conclusions are observed regarding uncertainties

on the asymmetry parameter. Higher $g$ values impact the ALP retrievals from $NN_{R_c,N^s_{O_2-O_2}}$ over scenes with $\tau \leq 1.0$. Such a bias is largely reduced with the $NN_{\tau,N^s_{O_2-O_2}}$ configuration.

Surface albedo variability contributes to the length of the average light path and thus leads to changes in $N^s_{O_2-O_2}$. Retrieved $ALP$ bias (Eq. 11) varies between 23 hPa and 141 hPa on average, for surface albedo uncertainties $\partial x_{alb}$ in the range of 0.005-0.05 with the $NN_{R_c,N^s_{O_2-O_2}}$ configuration (see Fig. 6b). To limit the values below 100 hPa on average, $\partial x_{alb}$ should

not exceed 0.025. Using the $NN_{\tau,N^s_{O_2-O_2}}$ configuration with true $\tau$ value, not only reduces the average but also the standard deviation of the bias: i.e. for $\partial x_{alb} = 0.05$, the aerosol pressure bias decreases from $141 \pm 96$ hPa with $NN_{R_c,N^s_{O_2-O_2}}$ to $69 \pm 79$ hPa with $NN_{\tau,N^s_{O_2-O_2}}$. The reasons of all these improvements are: 1) the strong correlation between $R_c$ and $N^s_{O_2-O_2}$ provides with a limited number of independent pieces of information to the retrieval system (see Sect. 2.3), 2) uncertainties about the surface albedo and aerosols $\omega_0$ impact the interpretation of both of $R_c$ and $N^s_{O_2-O_2}$ and mislead then the interpolation

process, 3) using $\tau$ as input is more consistent with the definition of aerosol models in the learning database and allows a better distinction the effects of $\tau$ and ALP on the $O_2-O_2$ slant column density.

A precision of $N^s_{O_2-O_2}$ lying in the range of 0.05-0.25 $10^{-43}$ mol$^2$cm$^{-5}$ results in similar ALP bias between $19 \pm 29$ hPa and $57 \pm 31$ hPa on average for both NN configurations (see Fig. 5c). Note a temperature correction, for real observations, must be taken into account to correctly interpret $N^s_{O_2-O_2}$ (see Sect. 5.1).

Overall all the estimated NN retrieval uncertainties are in line with the theoretical sensitivity analyses of Park et al. (2016) who found that the $O_2-O_2$ at 477 nm is significantly influenced by aerosol optical properties (including single scattering albedo), $\tau$, particle size and surface albedo. In particular, a $\omega_0$ uncertainty of 10 % was demonstrated to lead to the aerosol effective height (AEH) retrieval error ranging from 270 to 1440 m, depending on the aerosol types. Errors were found larger for high particle altitude and low $\tau$ cases. A surface albedo uncertainty of 0.02 was expected to impact AEH retrievals between

154 m and 434 m on average. AEH error was frequently larger only for low $\tau$ ($\leq 0.4$) and high AEH ($\geq 1$ km).



## 5 Application to OMI observation measurements

### 5.1 Methodology

Aerosol retrievals, as described in the previous sections, are performed on the OMI $O_2-O_2$ observations over large industrialized areas in North-East Asia over 3 years (2005-2007) and cloud-free scenes. All the associated results are summarized in Tab. 2.

Only observations collocated with MODIS Aqua Level 2 (L2) aerosol product collection 6 (Xiao et al., 2016) are considered. Their spatial resolution, 10 km x 10 km, is relatively close to the OMI nadir spatial resolution. The MODIS instrument, launched on the NASA EOS-Aqua platform in May 2002, is a spectrometer delivering continuous images of the Earth in the visible, solar infrared and thermal infrared approximately 15 min prior to OMI on-board EOS-Aura. The improved calibration of MODIS Aqua instrument is included in the reprocessing of the collection 6 aerosol product (Levy et al., 2013; Lyapustin et al., 2014). The reason to consider the MODIS aerosol products is here triple: 1) to maximize the probability of the selection of cloud-free OMI observation pixels dominated by aerosol pollution, 2) to evaluate the retrieved OMI $\tau$ products by comparing with collocated MODIS $\tau$(550 nm), 3) to use the MODIS $\tau$(550 nm) as input of the $NN_{\tau, N^s_{O_2-O_2}}$ algorithm for retrieving the OMI ALP product, assuming then this is the most accurate $\tau$ information available for each collocated OMI-MODIS observation pixel.

MODIS data are paired on a pixel-by-pixel basis if the distance between pixel centers is smaller than 5 km and if both observations are acquired within 15 min. A threshold of 0.1 is applied to both OMI and MODIS cloud fraction: i.e. if the OMI effective cloud fraction and/or the MODIS geometric cloud fraction has a cloud fraction value higher than 0.1, the pixels are filtered out. Applying such a threshold increases the probability of identifying cloud-free scenes. Furthermore, the availability of the MODIS aerosol products is a good confirmation of cloud-free scenes as MODIS Aqua $\tau$ variable is exclusively retrieved over sufficient cloud-free pixels. However, since the OMI effecive cloud fraction is sensitive to the scattering aerosols, it is well recognized that cloud-free observations with large presence of scattering aerosols are frequently excluded as well. In addition, a threshold of 0.1 is applied on the OMI surface albedo (i.e. OMLER database) in order to filter out too bright surfaces (either desert of snow covered pixels).

The NN retrieval algorithms developed and selected in Sect. 3 are used on the OMI DOAS $O_2-O_2$ observations, available in the OMCLDO2 product (Acarreta et al., 2004) which can be downloaded here: $http: //disc.sci.gsfc.nasa.gov/Aura/data - holdings/OMI/omcldo2_v003.shtml$. As explained in Veefkind et al. (2016), the $O_2-O_2$ slant column density depends on the temperature profile due to the nature of dimers of which the absorption scales with the pressure-squared instead of being linear with pressure. Therefore, a simple temperature correction is here applied by using seasonal mean temperature profiles given by the National Centers for Environmental Prediction (NCEP) analysis data. This correction is performed through the computation of the $\gamma$ factor (Veefkind et al., 2016):

$$\gamma = \frac{N^{sRef}_{O_2-O_2}(\lambda)}{N^{sMeas}_{O_2-O_2}(\lambda)}. \tag{9}$$





with $N_{O_2-O_2}^{sRef}$, the $O_2-O_2$ SCD associated with the reference temperature profile employed in the learning database and $N_{O_2-O_2}^{sMeas}$, the measured $O_2-O_2$ SCD related to the actual temperature conditions. As a first and simple assumption, no prior knowledge on aerosols is considered here. The main reason is the little sensitivity to aerosol loading of this $\gamma$ factor compared to the change of temperature profiles for the considered OMI observations.

Finally, retrievals are performed based on different assumed surface albedo databases. The standard and referent product is the OMLER surface reflectance climatology derived from several years of OMI observations at the spatial resolution of 0.5 ° x 0.5 ° longitude-latitude grid for each calendar month (Kleipool et al., 2008). The OMLER algorithm is based on temporal histograms of the observed LER values per geophysical location. Potential small residual cloud and aerosol contaminations are expected to remain in the OMLER product. As an alternative, the global and spatially complete MODIS black sky surface

albedo in the band 3 (459-479 nm) is considered. It is defined as the directional hemispherical reflectance and is a function of $\theta_0$ (Schaaf et al., 2002). It is derived by integrating the atmospheric corrected Bidirectional Reflectance Distribution Function (BRDF), derived from combined MODIS-Aqua and Terra observations over every 16-day period. The downwelling flux in the MODIS black sky albedo has no diffuse component. The Collection 6 of MCD43C3 product is given on a 0.05° (5.6 km) latitude/longitude Climate Modeling Grid (CMG), and is then resampled to match the OMI pixel resolution by calculating the

average of all MODIS pixels falling within the processed OMI pixel. Note that (Kleipool et al., 2008) demonstrated that the OMLER data set is closer to the black sky than to the white sky by evaluating the ratio between diffuse and direct illumination.

### 5.2  Aerosol optical thickness accuracy: on the importance of the surface albedo and the assumed aerosol properties

Figure 9 shows collocated retrieved OMI and MODIS $\tau$ over North-East Asia in 2005-2007 and for cloud-free scenes. As seen in Sect. 4.1, the change of aerosol $\omega_0$, mostly perturbs retrievals of high $\tau$ values, and thus the slope between OMI and MODIS

$\tau(550\,\mathrm{nm})$. Increasing $\omega_0$ from 0.9 to 0.95 reduces the $\tau$ retrieval values of about 0.5 for MODIS $\tau$ = 1.5. This is in line with the analyses on simulations in Sect. 4.1. Overall a very good agreement is obtained assuming $\omega_0$ = 0.9 for the seasons spring, autumn and winter (see Fig. 9 and Fig. 13): differences (OMI-MODIS) of $\tau(550\,\mathrm{nm})$ lie between $-0.18 \pm 0.24$ in winter and $-0.02 \pm 0.45$ in spring. In summer time, the best agreements are found assuming $\omega_0$ = 0.95 with differences in the range of $-0.06 \pm 0.31$ (see Fig. 8 and Fig. 12).

From the end of autumn to spring, westerly winds transport mineral dust from the Taklimakan and Gobi deserts in northern China and Mongolia region. These dust particles are then frequently mixed with the local anthropogenic aerosols released from the industrial activities, vehicle emissions and coal burning (Eck et al., 2005). Southeast Asia is affected in spring by biomass-burning activity (mostly over the peninsular) which is a major source of carbonaceous aerosols in the world. Overall, Jethva et al. (2014) show that AERONET and OMAERUV retrieve aerosol $\omega_0$ values on average between 0.9 and 0.95 in these

regions: most of the sulphate particles have $\omega_0$ close to 0.95, while smoke and dust present lower values (closer to 0.9, even below in some cases). These observations confirm that the employed aerosol model with $\omega_0 = 0.95$ should be considered as an upper limit for the OMI $\tau$ retrievals in autumn, winter and spring times, while a lower $\omega_0$ (i.e. 0.9) is likely more appropriate and thus allows, on average, more reliable $\tau$ retrievals. In summer time, because of reduced amounts of dust particles, $\tau$ values are more representative of local anthropogenic urban pollution, with more daily variabilities in the optical and scattering



properties. Lee et al. (2007); Lin et al. (2015) also found higher $\omega_0$ values over North-Eas Asia in summer (0.95-0.96) and lower for the other seasons (0.88-0.92). Overall, assuming same constant value (i.e. average) for all the acquired OMI pixels probably lead to some errors since aerosol scattering and absorption properties likely vary day-to-day, even month-to-month.

Figure 11 depicts a dependence of the retrieved OMI $\tau$ to the OMLER surface albedo values. As discussed in Sect. 4.1, error in surface albedo directly creates a bias on the retrieved $\tau$. In summer and spring, higher values are found over darker surfaces (i.e. OMLER surface albedo between 0.05 and 0.06). In the range of OMI surface albedo values 0.05-0.1, such behavior should not be observed assuming no systematic bias on the surface properties. Moreover, OMI $\tau$ shows too small values for scenes with MODIS $\tau \leq 0.4$ in autumn and winter using OMLER (see Fig. 11 and Fig. 13). Using the MODIS black sky surface albedo allows to reduce this dependence in summer and spring (see Fig. 11d) and reasonably increases OMI $\tau$ retrievals over scenes with low MODIS $\tau$ values (cf. Fig. 11b, Fig. 13c and Fig. 13g) in autumn and winter. Furthermore, standard deviation of differences (OMI-MODIS) $\tau$(550 nm) shows a net improvement of the retrievals precision, with a general reduction of 0.05 from OMLER to MODIS black sky. An exception is however noticed in winter, which may be due to remaining snow covered pixels in spite of the applied filtering (see Sect. 5.1). Overall spatial patterns better match between collocated MODIS and OMI pixels when employing MODIS black sky albedo with higher values over the high density population areas (i.e. North-East and South-West of selected Chinese region, South-West of Korea) and lower values over South-East of China (see Fig. 12 and Fig. 13). These improvements may be due to a more accurate atmospheric correction in the MODIS black sky surface albedo and potential remaining aerosol residuals present in the OMLER database.

In spite of these improved precisions, using the MODIS black sky albedo does not always improve the accuracy of the OMI $\tau$ retrievals. In particular, summer and spring seasons present too high $\tau$ values compared to the use of OMLER. This emphasizes that applying the MODIS black sky albedo to OMI measurements may be not fully optimal as: 1) MODIS albedo is the integral value over the full hemisphere which is not in line with the range of angles ($\theta_0$ and $\theta$) encountered by OMI, 2) the MODIS black sky albedo is valid for local solar noon zenith angle of each location which does not match the 1345 ascending node equator crossing time of OMI. An ideal surface albedo database should be aerosol and cloud free, and representative of the viewing and solar angles encountered by the space-borne sensor. Problems related to uncertainties in surface albedo climatologies for the aerosol retrieval problem are well known, and has recently been highlighted by Sanders et al. (2015), although a different spectral band is used ($O_2-A$ at 758-770 nm).

Furthermore, errors in the phase function or not taking into account the effect of polarization, can play a role. These aspects should be further investigated.

## 5.3 Long-term analyses of the aerosol layer pressure retrievals

Figure 14 shows the behavior of the retrieved OMI aerosol layer pressure as a function of collocated MODIS $\tau$(550 nm). MODIS $\tau$ is considered for OMI ALP retrievals since, at this stage, they are considered as the best prior information available with higher accuracy than OMI $\tau$ (see Sect. 5.1). While ALP retrievals over scenes with MODIS $\tau \leq 0.5$ exhibit large variability (more than 400 hPa) and are systematically very high, they start converging to more realistic values with increasing MODIS $\tau$. At MODIS $\tau \geq 1.0$, retrieved ALP lies in the range of 800-1000 hPa depending on the season, with lower variability (between





hPa and 200 hPa maximum). As discussed in Sect. 4.2, scenes with $\tau \leq 0.5$ are expected to present substantial large biases because of the minor impacts on the $O_2-O_2$ changes. Part of the variability can be related to uncertainties of surface albedo and non-constant and inhomogeneous aerosol properties from pixel-to-pixel (e.g. aerosol $\omega_0$ in the OMI observations).

When considering the $NN_{\tau, N^s_{O_2-O_2}}$ configuration with MODIS $\tau(550\,\mathrm{nm})$, from Dark target Algorithm over land, as input,
the retrievals globally show a reduced variability, especially for $\tau$ values in the range of 0.6:2.0 compared to the $NN_{R_c, N^s_{O_2-O_2}}$ configuration (see Fig. 14). Over scenes with MODIS $\tau \geq 1.0$ (see Fig. 15), the variability of the OMI Aerosol Layer Height (ALH), derived from Eq. 10 as explained in the next subsection, greatly decreases from the range of [1.1:2.7] km ($NN_{R_c, N^s_{O_2-O_2}}$) to [0.7:1.9] km ($NN_{\tau, N^s_{O_2-O_2}}$) depending on the season. When the OMLER is replaced by the MODIS black sky albedo database, the ALH variability continue to decrease of about 0.1 km.

## 5.4 Comparison of OMI aerosol layer height with LIVAS climatology

The results of 3 years of OMI ALP retrievals over North-East Asia can be statistically compared to a climatology. The LIdar climatology of Vertical Aerosol Structure for space-based lidar simulation studies (LIVAS) is a 3-D multi-wavelength global aerosol and cloud optical database (Amiridis et al., 2015). This database provides averaged profiles of aerosol optical properties over 9 years (1 january 2007 - 31 December 2015) from the Cloud Aerosol Lidar and Infrared Pathfinder Satellite Observations
(CALIPSO) data on a uniform grid of $1° \times 1°$. LIVAS addresses the wavelength dependency of aerosol properties for many laser operating wavelengths including 532 nm. LIVAS data set has been evaluated against AERONET in Amiridis et al. (2015) showing realistic and representative mean state aerosol optical depth values in 532 nm making this data set ideal for synergistic use with other satellite products. Although the years of the OMI "climatology" and LIVAS do not strictly overlap, it is assumed that the average aerosol layer height (ALH) does not significantly change between both periods. The comparison is done per
season. Spatial average of LIVAS ALH is done over the same area where retrievals are performed. Since large biases are expected at low $\tau$, only OMI retrievals acquired for MODIS $\tau(550\mathrm{nm}) \geq 1.0$ are taken into account and then spatially and temporally averaged per season. About 17 % in summer and spring, and between 5 % and 6 % in winter and autumn, of the OMI retrievals over the 3 years were then selected. As a first and simple approximation, OMI ALP retrievals are converted into ALH in km above sea level, assuming the atmosphere is in hydrostatic balance, scale height of 8 km and a surface pressure at
the sea level of 1013 hpa:

$$ALH = -8 * \ln(ALP/1013). \tag{10}$$

Figure 16 illustrates that the sensitivity to the choice of the aerosol $\omega_0$ is higher when using the $NN_{R_c, N^s_{O_2-O_2}}$ configuration. A minimum bias of 650 m (in winter) and a maximum of 1140 m (in Autumn) compared to LIVAS are found with $\omega_0 = 0.95$. On the other hand, ALH retrievals assuming $\omega_0 = 0.9$ already show a remarkable agreement with LIVAS (see Fig. 16b). The
OMI ALH can differ up to 1200 m due to the chosen aerosol $\omega_0$. The choice of the surface albedo mostly perturbs the OMI ALH, between 400 m and almost 500 m, when assuming $\omega_0 = 0.95$. Using the MODIS black sky surface albedo, instead of OMLER, allows to decrease the maximum ALH bias from 1140 m to 500 m (see Fig. 16b). As discussed in Sect. 5.2, an exception is noticed for the winter results with MODIS black sky albedo that may be due to remaining snow covered pixels.





Overall, retrievals from $NN_{\tau, N^s_{O_2-O_2}}$ configuration, with MODIS $\tau(550\,\mathrm{nm})$ as input, exhibit lower biases with $\omega_0 = 0.95$: in the range of [260:800] m (see Fig. 16d). The choice of the aerosol model also has reduced impacts with a maximum dispersion of 600 m, between $\omega_0 = 0.9$ and $\omega_0 = 0.95$. The assumptions about surface albedo show very little perturbation on the retrievals, less than 200 m. This result looks more consistent. Indeed, since selected OMI cases include a very high amount

of particles, surface reflection is expected to be largely attenuated. Mostly aerosol scattering and shielding effects should dominate the scenes. If the $NN_{R_c, N^s_{O_2-O_2}}$ configuration is used, any error in the surface albedo leads to a misinterpretation of $R_c$ and rises ambiguities when analysing $N^s_{O_2-O_2}$ without information on the amount of particles or their optical thickness. Using MODIS $\tau$ as input allows to mitigate this problem.

Applying the temperature correction on $N^s_{O_2-O_2}$ (see Eq. 12) leads to changes between 50 and 300 m on the ALH. Major

impacts are, however, expected on cases with lower $\tau$ cases associated with smaller $N^s_{O_2-O_2}$ changes (Veefkind et al., 2016). Another benefit of this temperature correction is to allow a more consistent seasonal patterns between OMI and LIVAS ALH, with higher values in spring and summer, due to long-range transport aerosols during the maximum dust activity from the deserts (see Sect. 5.2), and smaller values in winter and autumn (see Fig. 16c and Fig. 16d). Nevertheless, while the LIVAS ALH depict that aerosols should be at higher altitude in spring than in summer, the OMI ALH show the opposite. Several

explanations are possible: 1) exclusion of OMI scenes with strong aerosol pollutions because of a too strict threshold applied on the OMI effective cloud fraction (see Sect. 5.1), 2) a more rigorous temperature correction should be applied on measured $N^s_{O_2-O_2}$ (assuming daily instead of seasonal temperature profiles), 3) inaccuracies in the MODIS $\tau$ values, etc... All these elements should be further investigated.

These results seem to present higher accuracy than the exercise of Park et al. (2016) showing a bias of 1 km of between

retrieved OMI and the CALIPSO AEH values over ocean, during the Asian dust event on 31 March 2007. The reasons can be multiple: use of NNs instead of linear interpolation within a LUT, aerosol retrieved over land instead of ocean surfaces, consideration of variable surface albedos as inputs instead of a single value, application of a temperature correction on $N^s_{O_2-O_2}$, use of longer data records etc...

# 6 Conclusions

In this study, different Multilayer Perceptron Neural Network algorithms were developed and evaluated in order to retrieve aerosol layer height (ALH) over land from the OMI 477 nm $O_2-O_2$ spectral band. The aerosol height was here retrieved as aerosol layer pressure (ALP) and defined as the mid-pressure of an homogeneous and fixed scattering layer. The focus was on North-East Asia and cloud-free scenes dominated by scattering aerosol particles with $\omega_0$ in the range of 0.9-0.95. The algorithms were trained with a large synthetic data set and several precautions were taken into account to avoid problems such

as over-training or local minima. The key concept of ALP retrievals is the link between the $O_2-O_2$ slant column density (SCD) and the aerosol altitude as a result of shielding effect applied by the particles on the $O_2-O_2$ dimer complexes at lower altitudes. ALP was retrieved on 3 years (2005-2007) of OMI cloud-free observations collocated with MODIS-Aqua aerosol product in North-East Asia. The main objective of this work is to evaluate the feasibility of a direct retrieval of this key aerosol





parameter necessary to calculate air mass factors for trace gas retrievals. All the evaluated performances are summarized in Tab. 2

Accurate knowledge of aerosol optical thickness $\tau$ is required for a good ALP retrieval. Indeed, both $\tau$ and ALP parameters simultaneously contribute to the shielding of $O_2-O_2$ dimers. The analyses of the measured $O_2-O_2$ SCD alone leads to

an ambiguity since aerosol extinction and aerosol altitude cannot be distinguished. Without knowledge of $\tau$, the continuum reflectance (475nm) $R_c$ can be used as an alternative. However, this configuration is not optimal as 1) both $O_2-O_2$ SCD and $R_c$ contain overlapping information (i.e. $\tau$ and the length of the average light path) and 2) other parameters such as surface albedo, aerosol properties (e.g. aerosol single scattering albedo $\omega_0$ and phase function), and their associated uncertainties, impact as well $R_c$.

Different NN configurations were tested. Sensitivity analysis on simulations show that ALP accuracy lies in the range of 50-100 hPa (i.e. about 500 m and 1 km) over aerosol scenes with $\tau \geq 1.0$. Using accurate $\tau$ information, instead of $R_c$, reduces the impact of uncertainties due to 1) aerosol model: bias from 100 hPa to almost zero, if $\tau \leq 1.0$, for a difference of 0.05 in $\omega_0$ or for a difference of 0.1 in $g$, 2) surface albedo (bias from $141 \pm 96$ hPa to $69 \pm 79$ hPa for a surface albedo uncertainty of 0.05). When comparing the 3-year retrievals with the LIVAS climatology database, the Aerosol Layer Height (ALH) results

using MODIS-Aqua $\tau$(550 nm), over cloud-free scenes with $\tau \geq 1.0$, show the best accuracy: maximum biases in the range of 260-800 m depending on the season and assuming $\omega_0 = 0.95$. Assumed $\omega_0$ (either 0.9 or 0.95) impacts ALH retrievals up to 600 m on average, while changes due to the the assumed surface albedo database (OMLER or MODIS black sky) do not exceed 200 m.

In addition, algorithms should take into account that the $O_2-O_2$ SCD precision, resulting from the DOAS spectral fitting, affects the ALP retrieval. $O_2-O_2$ SCD precision lying in the range of 0.05-0.25 $10^{-43}$ mol$^2$cm$^{-5}$ leads to ALP bias between

$19 \pm 29$ and $57 \pm 31$ hPa. Due to the nature of the $O_2-O_2$ collision complex, a temperature correction must be applied on the SCD prior to retrievals. This impacts the retrieved ALH between 50 and 300 m and enables to reproduce more consistent seasonal patterns. Other parameters should be further investigated such as polarization effects and assumptions about the vertical distribution of particles.

Because low amount of aerosols have very little impacts on $O_2-O_2$ SCD changes, large biases are expected over scenes with $\tau$(550 nm) $\leq 0.5$. This $\tau$ value should be considered as a threshold for a good ALP retrieval quality. Moreover, the algorithms are expected to present a very low sensitivity to particles located at an altitude higher than 4 km. This is because of the nature of the $O_2-O_2$ complex of which the absorption scales with the pressure-squared instead of being linear with pressure,

The capability of deriving a $\tau$ information from the OMI 477 nm $O_2-O_2$ spectral band was also investigated. Accuracy

of $\tau$ retrievals relies on the assumed parameters affecting $R_c$. An overestimation of aerosol single scattering albedo, from $\omega_0 = 0.9$ to 0.95, induces a negative bias of 0.8 for $\tau = 2$. The impact is much lower for smaller $\tau$ (lower than 0.1 for $\tau$s $\leq 0.5$). Similar conclusions were found regarding uncertainty of the asymmetry parameter and thus the phase function characterization. Another major challenge when retrieving aerosol properties from passive satellite sensors is to separate the atmospheric and surface contributions in the total observed reflectance. Similarly as aerosol $\omega_0$, an overestimation of surface reflection leads to

an underestimation of retrieved $\tau$. Surface albedo uncertainty should be limited to 0.025 to ensure $\tau$ bias $\leq 0.2$. Comparisons of





OMI retrievals with collocated MODIS $\tau$ show agreements between $-0.02\pm0.45$ and $-0.18\pm0.24$ depending on the seasons. Further improvements should be made before being able to use these OMI $\tau$ products as prior information to ALP retrievals.

The NN approach presents, at this stage, quite promising results for a future operational processing of the OMI $O_2-O_2$ spectral band and the next UV-Vis satellite missions such as the TROPOspheric Monitoring Instrument (TROPOMI) (Veefkind et al., 2012). In spite of the high computing time due to the learning database creation and the training of these algorithms, very fast operational processing is allowed. Such processing is much faster than approaches relying on the Optimal Estimation Method and employs more optimized interpolation techniques than a classical linear interpolation within a LUT. For future processing of the OMI data, the OMLER climatology database should be optimized by filtering out small aerosol residuals.

The results described in this paper indicate that it is worthwhile to design and evaluate aerosol height retrieval algorithm exploiting the satellites 477 nm $O_2-O_2$ spectral band. This could lead to a replacement of the effective clouds by aerosol parameters in the computation of trace gas AMF. This has to be evaluated in the context of all the UV-Vis satellite missions devoted to air quality monitoring.

*Acknowledgements.* This work was funded by the Netherlands Space Office (NSO) under the OMI contract. The authors thank Piet Stammes, Folkert Boersma and Maarten Sneep from KNMI for the discussions about aerosol simulations and measurements, and sharing their experience with respect to the DISAMAR software.

We thank all the developers of the PyBrain software. PyBrain is a joint project of PostDocs, PhD and master level students. The core programmers are (or were) students of Prof. Jurgen Schmidhuber at the Dalle Molle Institute for Artificial Intelligence in Switzerland and the Technische Universitat Munchen in Germany.

The LIVAS products have been collected from the LIVAS database (http://lidar.space.noa.gr:8080/livas), and were produced by the LIVAS team under the European Space Agency (ESA) study contract No. 4000104106/11/NL/FF/fk. We thank all the developers who have provided us with all the data set and for sharing their discussions and recommendations.

The developers of LIVAS would like to acknowledge support through the project MarcoPolo under grant agreement n° 606953 from the European Union Seventh Framework Program (FP7/2007-2013) and the research program ACTRIS-2 under grant agreement no. 654109 from the European Union's Horizon 2020 research and innovation program.





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





**Table 1.** Ensemble of parameters and values associated with the simulated learning data set (see Sect. 3.2). Aerosols are simulated with a Henyey-Greenstein scattering phase function (Hovenier and Hage, 1989)

| Parameter | List of values |
| --- | --- |
| Solar Zenith Angle ($\theta_0$) [°] | 9.267, 21.167, 32.892, 44.217, 54.940, 64.814 |
| Viewing Zenith Angle ($\theta$) [°] | 0.0, 9.267, 21.167, 32.892, 44.217 |
| Relative azimuth angle ($\phi - \phi_0$) [°] | 0., 30., 60., 90., 120., 150., 180. |
| Surface pressure ($Ps$) [hPa] | 1013., 963. |
| Surface albedo ($A$) | 0.025, 0.05, 0.075, 0.1 |
| $\tau$ | 0.0, 0.05, 0.1, 0.2, 0.4, 0.5, 0.6, 0.9, 1.25, 2.0, 3.0 |
| Aerosol layer pressure (ALP) [hPa] | 975., 925., 850., 750., 700., 650., 550., 350., 150. |
| Aerosol Single Scattering Albedo ($\omega_0$) | 0.9, 0.95 |
| Ångström coefficient ($\alpha$) | 1.5 |
| asymmetry parameter ($g$) | 0.7 |

Axis Differential Optical Absorption Spectroscopy (MAX-DOAS) geometries calculated from different UV/visible radiative transfer models, Atmospheric Chemistry and Physics, 7, 1809–1833, doi:10.5194/acp-7-1809-2007, http://www.atmos-chem-phys.net/7/1809/2007/, 2007.

Wagner, T., Deutschmann, T., and Platt, U.: Determination of aerosol properties from MAX-DOAS observations of the Ring effect, Atmospheric Measurement Techniques, 2, 495–512, doi:10.5194/amt-2-495-2009, http://www.atmos-meas-tech.net/2/495/2009/, 2009.

Xiao, Q., Zhang, H., Choi, M., Li, S., Kondragunta, S., Kim, J., Holben, B., Levy, R. C., and Liu, Y.: Evaluation of VIIRS, GOCI, and MODIS Collection 6 AOD retrievals against ground sunphotometer observations over East Asia, Atmospheric Chemistry and Physics, 16, 1255–1269, doi:10.5194/acp-16-1255-2016, http://www.atmos-chem-phys.net/16/1255/2016/, 2016.





**Figure 1.** $\tau$ and aerosol layer pressure (ALP) as a function of $O_2-O_2$ slant column density $N^s_{O_2-O_2}$, continuum reflectance at 475 nm $R_c$ and true $\tau$ values for the following conditions: climatology mid-latitude summer temperature, $NO_2$, $O_3$ and $H_2O$ profiles, surface albedo $= 0.05$, $\theta_0 = 32°$, $\theta = 32°$, surface pressure $= 1013\,\text{hPa}$ and fine scattering aerosol particles ($\alpha = 1.5$, $\omega_0 = 0.95$, $g = 0.7$). The ellipses represent the $R_c$ and $N^s_{O_2-O_2}$ results derived from the simulations. The background colors then result from the radial basis interpolation of the DOAS fit results depicted by the ellipses: **(a)** $\tau$ and aerosol pressure as a function of DOAS fit parameters, **(b)** $\tau$ as a function of ALP and continuum reflectance at 475 nm, **(c)** ALP as a function of true $\tau$ and $O_2-O_2$ slant column density.





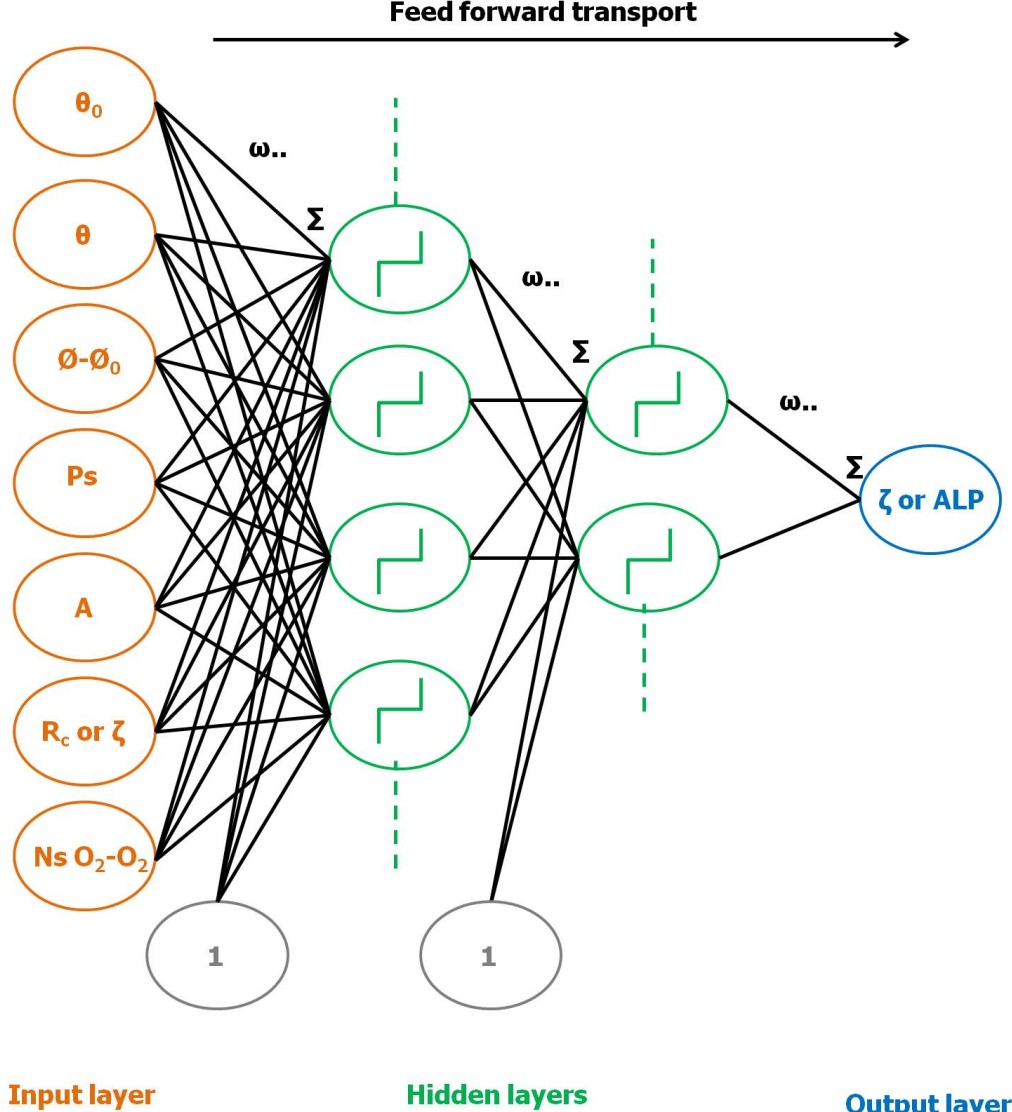

**Figure 2.** Diagram of Multilayer Perceptron (MLP) Neural Network (NN) architecture designed for Aerosol Layer Pressure (ALP) and aerosol optical thickness $\tau$ retrieval algorithms from the OMI $O_2-O_2$ spectral band at 477 nm. The input parameters are based on the list given in Table 1. The different considered approaches for the MLP design and their applications are more detailed in Sect. 3. Each circle represent a specific processor (named neuron) including either an input / output variable (in the input / output layer) or the activation function (i.e. sigmoid function in the hidden layer). The synaptic weights $\omega..$ ensure the connections of neurons between two consecutive layers. A weighted sum $\sum$ is performed before the transport through the activation function. Note the presence of the bias neurons, prior to the activation functions in the hidden layers. For simplicity, bias neurons are commonly visualized as values added to each neuron in the input and hidden layers of a network, but in practice are treated in exactly the same manner as other weights: all biases are simply weights associated with vectors that lead from a single node whose location is outside of the main network and whose activation is always 1. While the synaptic weights essentially change the steepness of the activation functions, the bias neurons allow to modify the origin of these functions from 0 to positive or negative values.





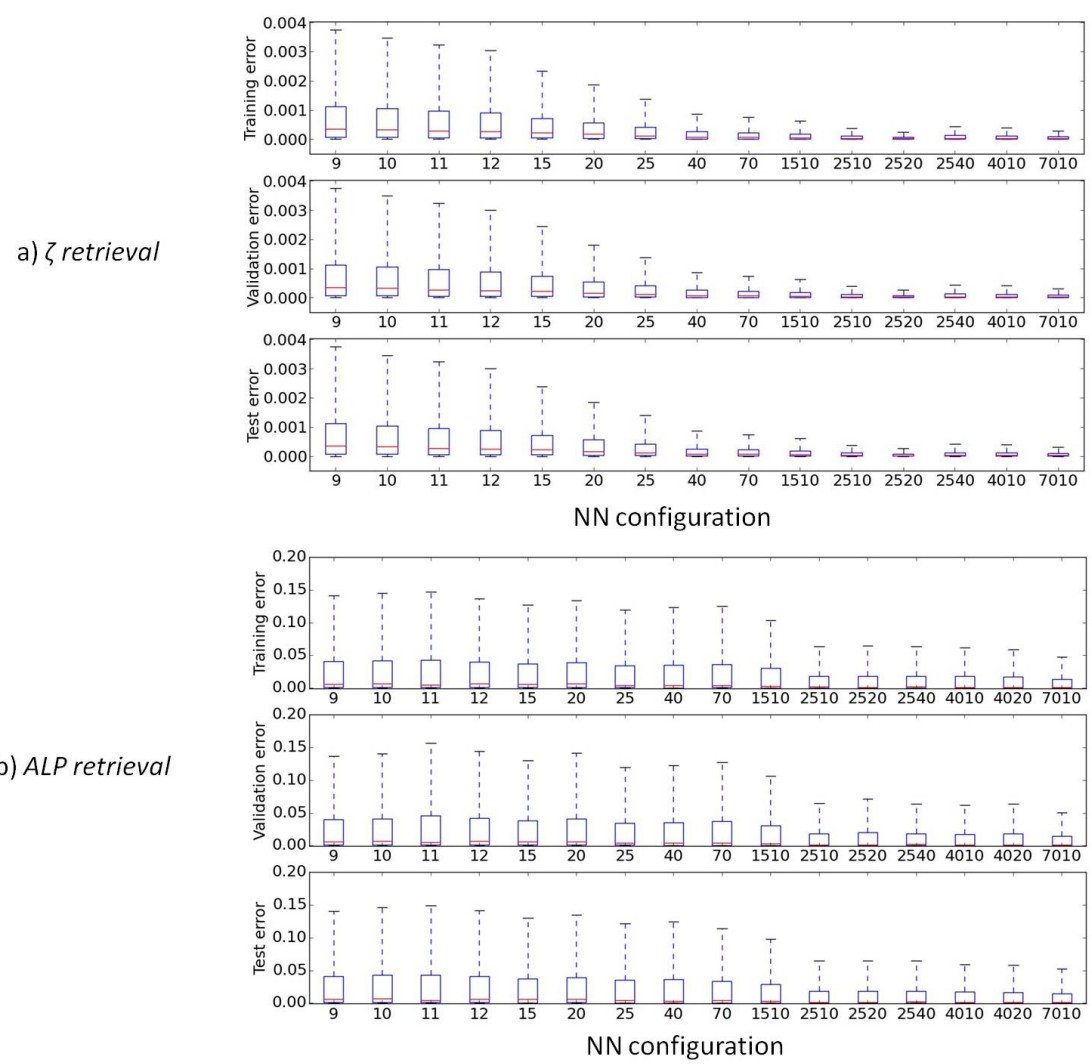

**Figure 3.** Box-whisker plots of the square of errors (see Eq. 7) obtained for different Neural Network (NN) configurations, at the end of their training, over the supervised data set (training-validation-test). The NNs XX have one hidden layer where XX indicate the number of neurons. The NNs YYXX have two hidden layers where YY and XX are the number of neurons in the 1st and 2nd hidden layer respectively: **(a)** NNs for $\tau$ retrieval, **(b)** NNs for $ALP$ retrieval. Note that errors are computed over normalized output and true $\tau$ and $ALP$ values (between -1 and 1) due to the definition of the sigmoid functions (see Sect. 3.1).



**Table 2.** Summary of OMI $\tau$(550 nm) and $ALP$ retrieval error sources and budget as evaluated by the sensitivity analyses on synthetic data set (see Sect. 4) or on 3-year (2005-2007) observation measurements over North East-Asia (see Sect. 5)

| Error source | $\tau$(550 nm) | From $NN_{R_c,N^s_{O_2-O_2}}$: $ALP$ in [hPa] $ALH$ in [m] | From $NN_{\tau,N^s_{O_2-O_2}}$ $ALP$ in [hPa] $ALH$ in [m] |
|---|---|---|---|
| $\tau$ values: | | | |
| True $\tau \leq 0.5$ (see Sect. 4) | - | [250:400] [hPa] | [150:400] [hPa] |
| True $\tau = [0.5 : 1.0]$ (see Sect. 4) | - | 100 [hPa] | 100 [hPa] |
| True $\tau \geq 1.0$ (see Sect. 4) | - | $\leq 50$ [hPa] | $\leq 50$ [hPa] |
| MODIS $\tau \geq 1.0$ (see Sect. 5) | - | [650:1140] m | [260:800] m |
| | | | |
| Surface albedo: | | | |
| $\partial A = 0.05$ (see Sect. 4) | 0.48 | $141 \pm 96$ [hPa] | $69 \pm 79$ [hPa] |
| OMLER vs. MODIS black sky (see Sect. 5) | [0.05:0.1] | $\leq 640$ m (MODIS ($\tau \geq 1.0$) | $\leq 200$ m (MODIS ($\tau \geq 1.0$) |
| | | | |
| Aerosol Single Scattering Albedo ($\partial\omega_0 = 0.05$) | | | |
| (see Sect. 4) | 0.8 (true $\tau = 2$) | 100 [hPa] (true $\tau = [0.5 : 1.5]$) | Almost zero (true $\tau \geq 0.5$) |
| | 0.1 (true $\tau = 0.5$) | | |
| (see Sect. 5) | 0.5 (MODIS $\tau = 1.5$) | 1200 m (MODIS ($\tau \geq 1.0$) | 600 m (MODIS ($\tau \geq 1.0$) |
| | | | |
| Asymmetry parameter ($\partial g = 0.1$) (see Sect. 4) | 0.5 (true $\tau = 1.5$) | [200:400] [hPa] (true $\tau = [0.5 : 1.0]$) | Almost zero (true $\tau \geq 0.5$) |
| | 0.25 (true $\tau = 0.5$) | 50 [hPa] (true $\tau \geq 1.0$) | |
| | | | |
| $O_2-O_2$ SCD ($\partial N^s_{O_2-O_2}$) (see Sect. 4): | | | |
| $\partial N^s_{O_2-O_2} = 0.05$ mol$^2$cm$^{-5}$ | - | $19 \pm 29$ [hPa] | $19 \pm 29$ [hPa] |
| $\partial N^s_{O_2-O_2} = 0.25$ mol$^2$cm$^{-5}$ | - | $57 \pm 31$ [hPa] | $57 \pm 31$ [hPa] |
| | | | |
| $O_2-O_2$ SCD temperature correction (see Sect. 5) | - | [50:300] m (MODIS ($\tau \geq 1.0$) | [50:300] m (MODIS ($\tau \geq 1.0$) |



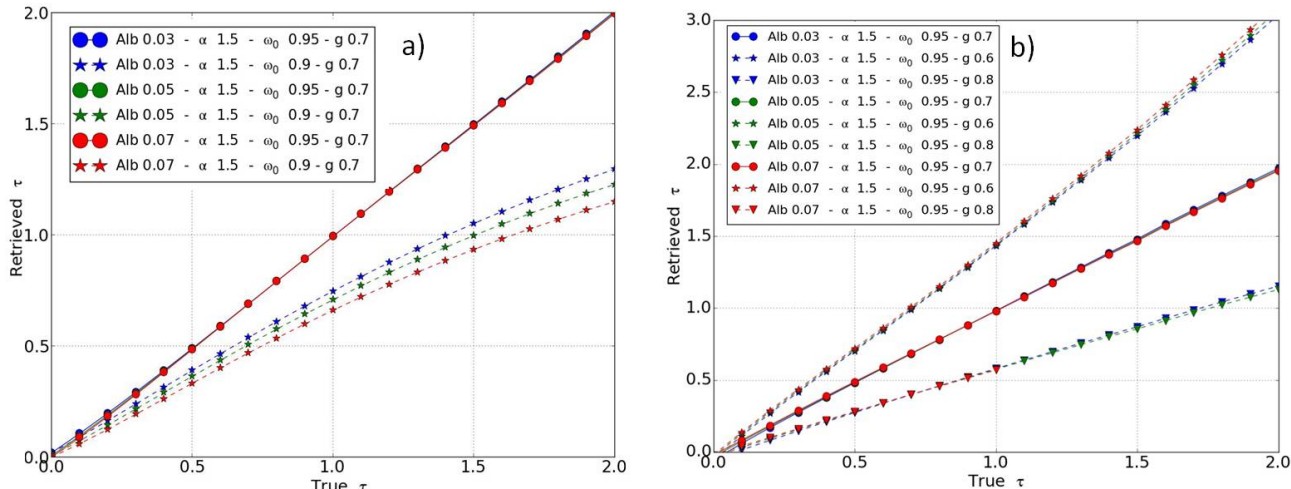

**Figure 4.** Simulated $\tau(550\,\mathrm{nm})$ retrievals, based on noise-free synthetic spectra with aerosols, as a function of true $\tau$. The assumed geophysical conditions are: temperature, $H_2O$, $O_3$, and $NO_2$ from climatology mid latitude summer, $\theta_0 = 25°$ and $\theta = 25°$, $Ps = 1010\,\mathrm{hPa}$. The referent aerosol scenario assumes fine scattering particles ($\alpha = 1.5$, $\omega_0 = 0.95$, $g = 0.7$) located between 800 and 900 hPa. All the retrievals are achieved with the NN algorithm trained with $\omega_0 = 0.95$: **(a)** Sensitivity of $\tau(550\,\mathrm{nm})$ retrievals to the aerosol single scattering albedo ($\omega_0 = 0.95$ or 0.9) in the synthetic spectra, **(b)** Sensitivity of $\tau(550\,\mathrm{nm})$ retrievals to the aerosol asymmetry parameter ($g = 0.6$, 0.7 or 0.8) in the synthetic spectra.





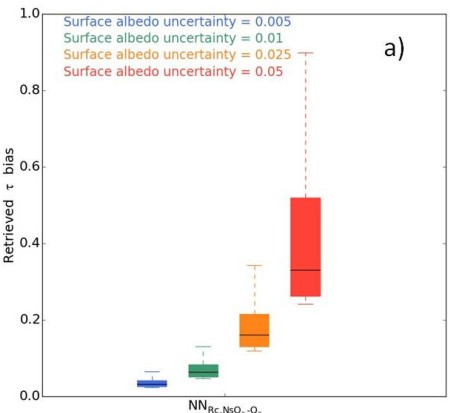

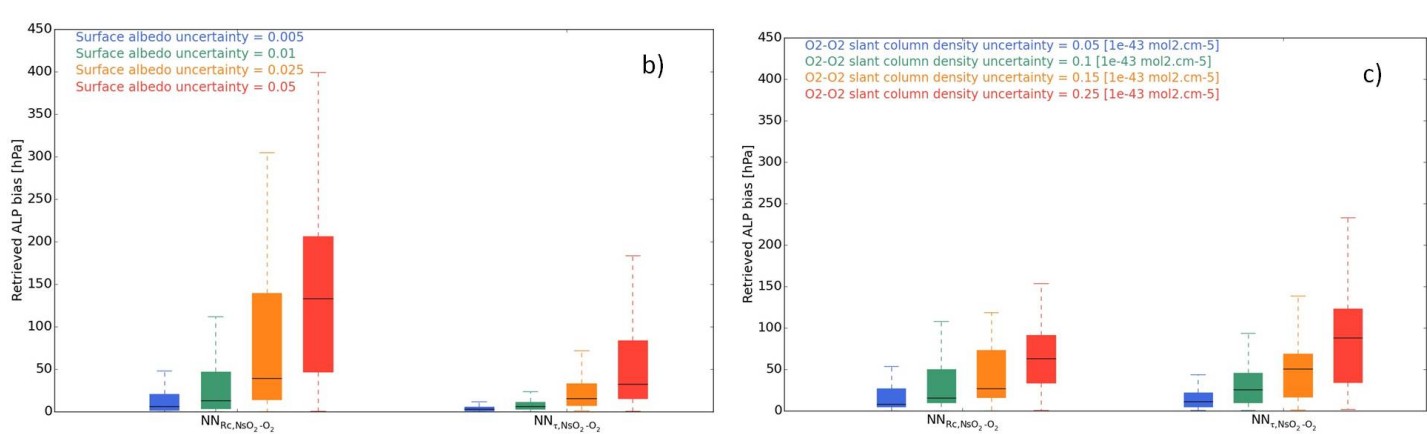

**Figure 5.** Box-whisker plots of aerosol retrieval biases induced by surface albedo and $O_2-O_2$ SCD $N^s_{O_2-O_2}$ uncertainties. The assumed conditions are: fine scattering aerosols ($\alpha = 1.5$, $\omega_0 = 0.95$, $g = 0.7$), climatology mid-latitude summer temperature, $NO_2$, $O_3$ and $H_2O$ profiles, surface pressure $= 1010\,\text{hPa}$, surface albedo $=[0.03\text{-}0.05\text{-}0.07]$ and combination of $\theta_0 - \theta = [25°\text{-}25°,\ 50°\text{-}25°,\ 25°\text{-}45°]$: **(a)** Retrieved $\tau(550\,\text{nm})$ bias due to surface albedo uncertainty, **(b)** Retrieved ALP bias due to surface albedo uncertainty, **(c)** Retrieved $\tau$ bias due to $O_2-O_2$ SCD uncertainty".





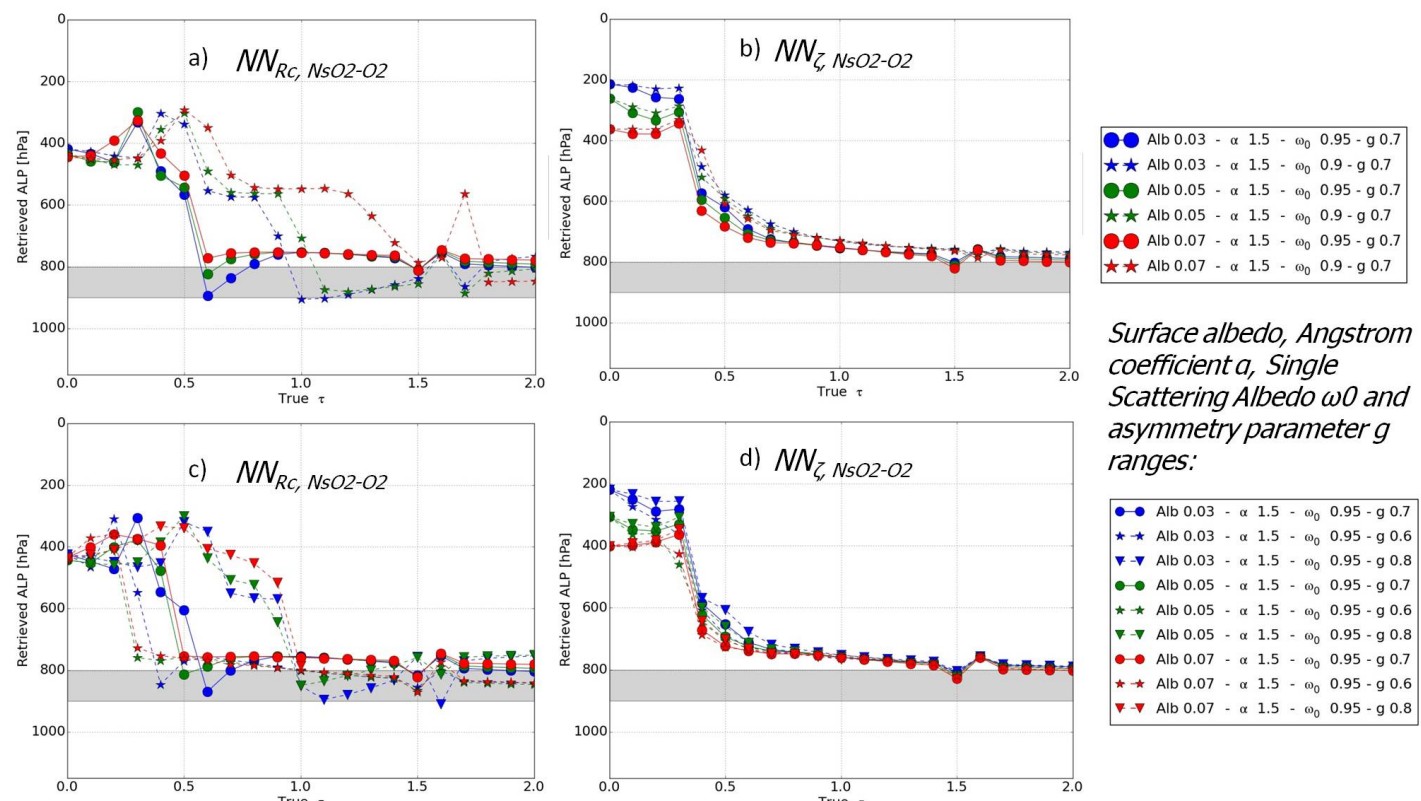

**Figure 6.** Simulated ALP retrievals, based on noise-free synthetic spectra with aerosols, as a function of true $\tau$. All the retrievals are achieved with the two NN configurations ($NN_{R_c,N^s_{O_2-O_2}}$ and $NN_{\tau,N^s_{O_2-O_2}}$) (see Sect. 3.1) trained with $\omega_0 = 0.95$. The assumed geophysical conditions are: temperature, $H_2O$, $O_3$, and $NO_2$ from climatology mid latitude summer, $\theta_0 = 25°$ and $\theta = 25°$, $Ps = 1010\,\text{hPa}$. The referent aerosol scenario assumes fine scattering particles ($\alpha = 1.5$, $\omega_0 = 0.95$, $g = 0.7$) located between 800 and 900 hPa: **(a)** and **(b)** Sensitivity of ALP retrievals to the aerosol single scattering albedo ($\omega_0 = 0.95$ or 0.9) in the synthetic spectra, **(c)** **(d)** Sensitivity of ALP retrievals to the aerosol asymmetry parameter ($g = 0.6$, 0.7 or 0.8) in the synthetic spectra.





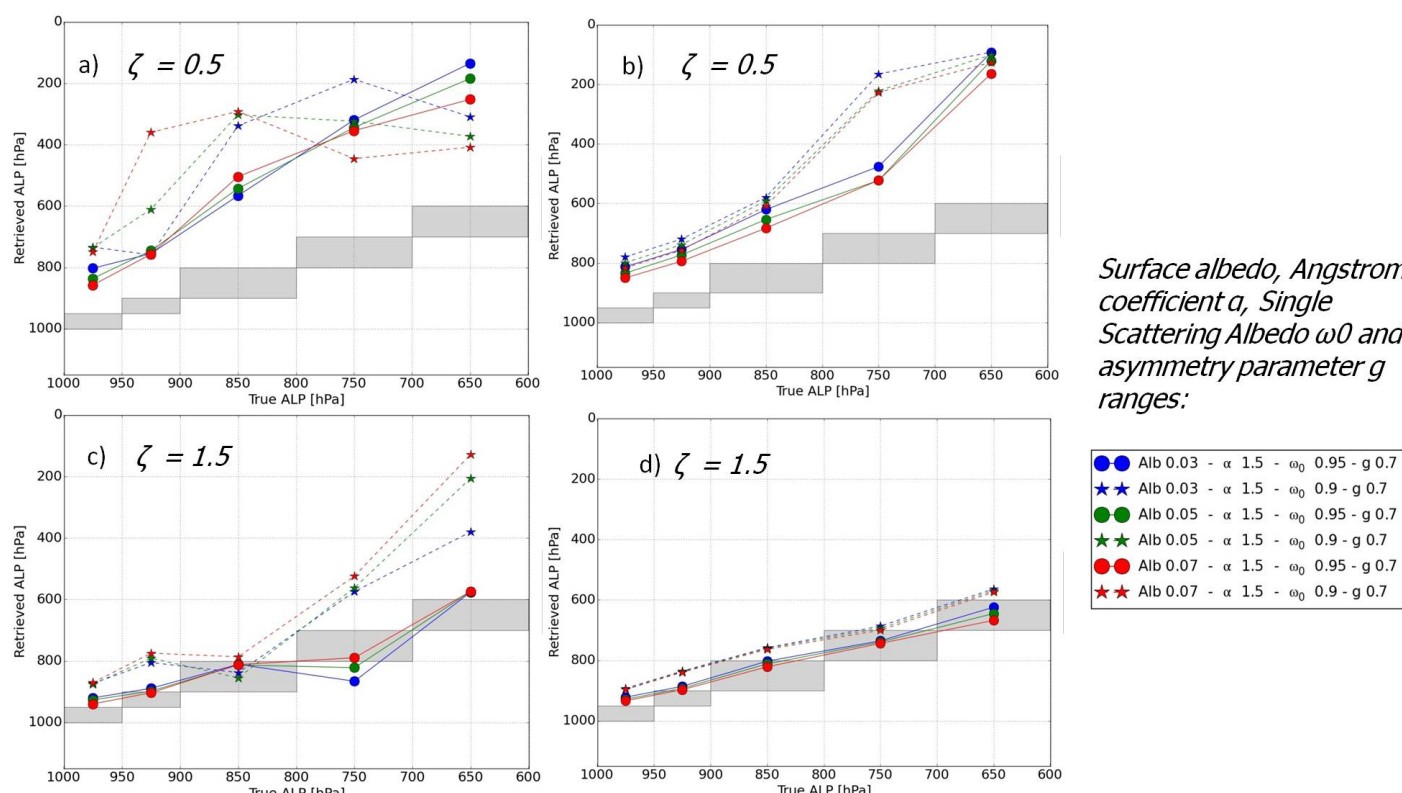

**Figure 7.** Simulated $ALP$ retrievals vs. true $ALP$ for 2 $\tau$ values (0.5 and 1.5) and the two NN configurations (see Sect. 3.1) and for the following conditions: temperature, $H_2O$, $O_3$, and $NO_2$ from climatology mid latitude summer, $\theta_0 = 25°$, $\theta = 25°$, surface pressure $= 1010\,\mathrm{hPa}$ and fine scattering aerosol particles ($\alpha = 1.5$, $\omega_0 = 0.95$, $g = 0.7$): **(a)** $NN_{R_c, N^s_{O_2-O_2}}$ and $\tau = 0.5$, **(b)** $NN_{\tau, N^s_{O_2-O_2}}$ with true $\tau$ value as input and $\tau = 0.5$, **(c)** $NN_{R_c, N^s_{O_2-O_2}}$ and $\tau = 1.5$, **(d)** $NN_{\tau, N^s_{O_2-O_2}}$ with true $\tau$ value as input and $\tau = 1.5$







**Figure 8.** Box-whisker plots of retrieved Aerosol Layer Pressure (ALP) biases as a function of true $\tau$ from $NN_{R_c, N^s_{O_2-O_2}}$ configuration over the the whole learning data set



**Figure 9.** Collocated MODIS Aqua and retrieved OMI $\tau(550\ \mathrm{nm})$ based on the OMLER surface albedo, over East China for cloud-free scenes and summer, winter and spring seasons. Statistics are computed over 3 years (2005, 2006, 2007): **(a)**, **(c)** and **(d)** assumed aerosol model with $\omega_0 = 0.95$, **(b)**, **(d)** and **(f)** assumed aerosol model with $\omega_0 = 0.9$.



**Figure 10.** Collocated retrieved OMI and MODIS Aqua $\tau$(550 nm) from Dark Target algorithm over land, over East China for cloud-free scenes. Retrievals are depicted as a function of OMLER surface albedo ranges (Kleipool et al., 2008) and for 2 seasons (winter and autumn). Statistics are computed over 3 years (2005, 2006,2007) : **(a)** and **(b)** OMI retrievals in winter based on OMLER and MODIS black sky surface albedo (see Sect. 5.1) respectively, **(c)** and **(d)** OMI retrievals in autumn based on OMLER and MODIS black sky surface albedo (see Sect. 5.1) respectively.





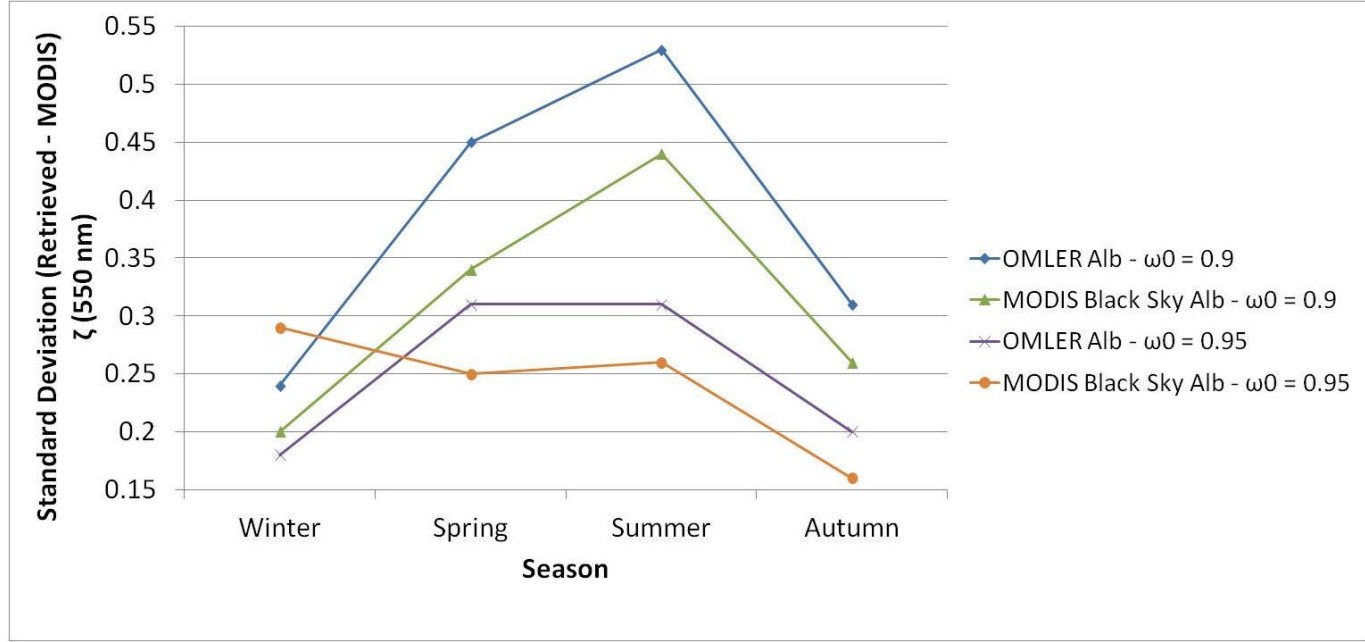

**Figure 11.** Standard deviation of the differences between OMI $\tau$ retrievals and MODIS $\tau(550\,\mathrm{nm})$ from Dark Target Land algorithm over land for all the individual cloud-free observations over North-East Asia. The retrievals are obtained over 3 years (2005, 2006,2007), and for the 4 seasons. Aerosol single scattering albedo $\omega_0 = 0.95$ and $\omega_0 = 0.9$ are assumed. OMLER and MODIS black sky surface albedo are alternatively considered.

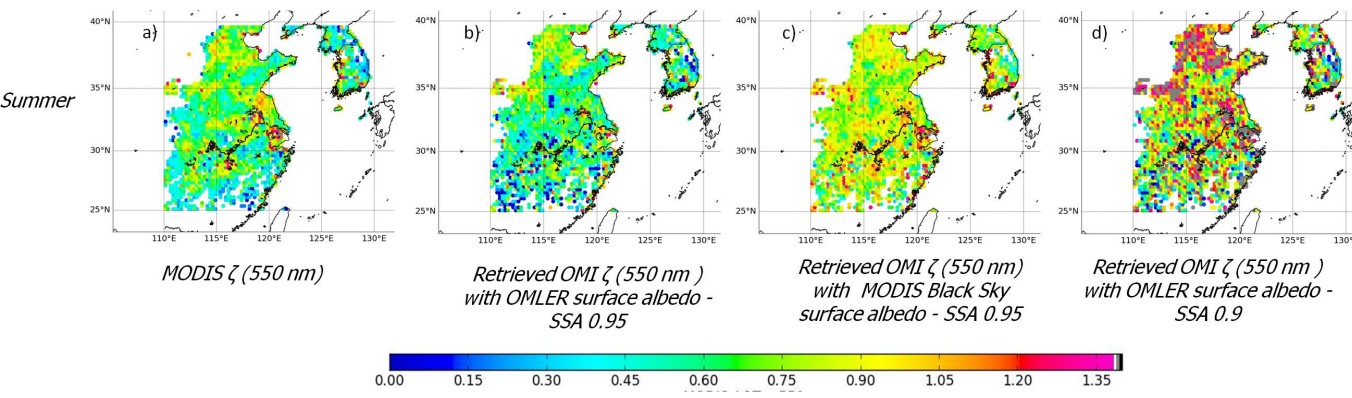

**Figure 12.** Spatial averages of $\tau(550\,\mathrm{nm})$ values, over East China for cloud-free scene. Statistics are computed over 3 years (2005, 2006,2007) for summer.





**Figure 13.** Spatial averages of $\tau(550\,\mathrm{nm})$ values, over East China for cloud-free scene. Statistics are computed over 3 years (2005, 2006,2007) for the 3 seasons, autumn, winter and spring.







**Figure 14.** Collocated retrieved OMI ALP (assumed aerosol model with $\omega_0 = 0.95$ and based on the OMLER surface albedo) and MODIS Aqua $\tau(550\,\text{nm})$, over North-East Asia for cloud-free scenes and summer, winter and spring seasons. Statistics are computed over 3 years (2005, 2006, 2007): **(a)**, **(b)** and **(c)** OMI retrievals are from the $NN_{R_c, N_{O_2-O_2}^s}$ configuration, **(d)**, **(e)** and **(f)** OMI retrievals are from the $NN_{\tau, N_{O_2-O_2}^s}$ configuration with MODIS Aqua $\tau(550\,\text{nm})$, from Dark Target Land algorithm, as input (see Sect. 5.1).





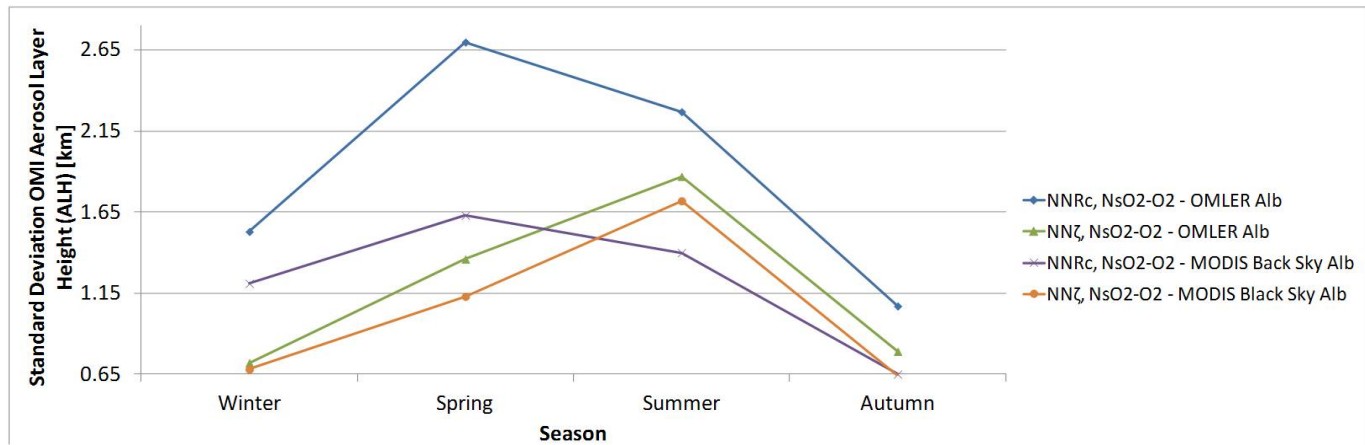

**Figure 15.** Standard deviation of the OMI aerosol layer height (ALH) retrievals obtained for the cloud-free scenes with MODIS $\tau(550\text{nm}) \geq$ 1.0 over North-East Asia. The retrievals are achieved over East China for cloud-free scene, over 3 years (2005, 2006,2007), and for the 4 seasons. Aerosol single scattering albedo $\omega_0 = 0.95$ is assumed. OMLER and MODIS black sky surface albedo, and the NN configurations ($NN_{R_c, N_{O_2-O_2}^s}$ and $NN_{\tau, N_{O_2-O_2}^s}$ with MODIS $\tau(550\text{ nm})$ as input) are alternatively considered.





**Figure 16.** Comparison of the average of the OMI aerosol layer height (ALH) retrievals obtained for the higher collocated MODIS $\tau$ values, with the LIVAS ALH climatology database. The retrievals are achieved over East China for cloud-free scene, over 3 years (2005, 2006, 2007), and for the 4 seasons. OMI and MODIS black sky surface albedo are considered alternatively. The 2 NN configurations are considered (see Sect. 5.1). Cases with and without the temperature correction applied on the $O_2-O_2$ are illustrated: **(a)** and **(b)** OMI ALH from the $NN_{R_c, N^s_{O_2-O_2}}$ configuration with and without temperature correction respectively, **(c)** and **(d)** OMI ALH from the $NN_{\tau, N^s_{O_2-O_2}}$ configuration with MODIS $\tau$(550 nm), from Dark Target Land algorithm, as input, with and without temperature correction respectively.