# Peer review of "An exploratory study on the aerosol height retrieval from OMI measurements of the 477 $\rm nm~O_2-O_2$ spectral band, using a Neural Network approach"

_Atmospheric Measurement Techniques, 2016_

## Referee Comment (RC1) · Anonymous Referee #1 · 16 Dec 2016

Review of An exploratory study on the aerosol height retrieval from OMI measurements of the 477 nm O2-O2 spectral band, using a neural network approach

The authors present a study exploring the retrieval of an aerosol layer height parameter from OMI measurements of the O2-O2 band in the visual wavelength range. They follow basically the same approach as the OMI O2-O2 cloud algorithm but use a neural network to replace the traditional look-up table. Sensitivities of the retrieval to assumptions related to the aerosol optical properties and the surface albedo are investigated. The algorithm setup has been applied to three years of OMI data and seasonal averages are compared with MODIS aerosol optical thickness and climatological aerosol heights from the LIVAS CALIPSO climatology.

In my opinion, the paper presents interesting and substantial work that in principle warrants publication in AMT. However, I feel that there are a number of major issues that should be addressed before I can commend publication. Major and minor comments are listed below. I would appreciate if these are addressed point-by-point and a diff-version of the manuscript is provided together with the replies. In addition, I struggled with the style and the structuring of the manuscript, which made it hard at times for me to quickly understand what the authors are trying to say. I give examples of this below. I realise that this is partly a matter of personal taste, but I strongly encourage the authors to use these comments to critically look at the entire manuscript. It would help the reader to better understand and appreciate the authors' valuable work.

Major comments

-A look-up table is generated and used to train a neural network, which is in turn used in the retrieval instead of the original look-up table. This lowers memory demands and increases computational speed. In my opinion this is a very good application of a neural network. However, in your extensive general discussion of neural networks you suggest that you use a neural network to resolve the ill-conditioning of the inverse problem. For example, p.7,l.30-p.8,l.2., particularly in p.8,l.16 and again in the conclusion on p.21,l.6. This is clearly not what you do. Your neural network is trained with simulated data, so all the physical relations between input and output contained in your neural network are already explicitly described in the RT model, and strong assumptions on the aerosol model, the vertical distribution of the aerosols and the ground surface are still a priori inputs for your retrieval. I think the text should be changed to really avoid giving the reader this wrong impression. Also, I don't quite see why interpolation in a look-up table should be less accurate than 'interpolation' in a look-up table with a neural network even in the case of non-linear data.

-An important figure is Figure 16. You are showing the seasonal dependence of ALH for your OMI retrievals and as derived from many years of CALIPSO retrievals. The amplitude of the seasonal variability in the LIVAS ALHs is only about 0.5 km and I am wondering whether that seasonal variability is picked up by your OMI retrievals. After all, biases due to wrong assumptions on the aerosol model are of the same order of magnitude or even larger. What makes Figure 16 a bit deceptive in my view is that you plot four lines showing OMI ALP retrievals for slightly different settings. These lines are obviously dependent and therefore correlate. At first sight, the overall consistency between the collection of plot lines suggested to me that you are indeed picking up the seasonal variability. But when having a closer look, I am not sure... the apparent consistency may be visually driven by these OMI ALP lines. In practice, you would pick some optimal retrieval setting based on sensitivity studies, literature etc. and do the retrieval, or you would perhaps run the retrieval for several retrieval settings and then take the average. I really think that for a proper comparison of LIVAS ALHs with your OMI ALHs, a figure like Figure 16 should therefore basically contain one panel with only two lines: one for LIVAS ALP and one for your best OMI ALP. Can you change Figure 16 (the effect of temperature corrections is not a key point of your paper and the effect of surface albedo and SSA could then be moved to a separate plot)? I fear that the agreement between OMI ALH and LIVAS ALH will look less convincing with only four pairs of points, but it is more realistic.

-In your analysis you assume that climatological CALIPSO profiles are the truth and you focus then on biases. I am willing to follow your assumption for the moment but I think presenting the story like this is too optimistic on the error in retrieved ALH for an individual pixel (which is what you would need for scatter corrections). (Again, I am making a point of this because you repeat bias values in the abstract, p.1.,l.9-10.) In the sensitivity analysis you identify several error sources that affect retrieved ALP (aerosol optical properties, surface albedo, neural network implementation). Together they form something more like a random error. When calculating biases as you do these random errors are averaged out. In my view, you should also report the root-mean-square

and the standard deviation of the differences (as in: rms**2 = bias**2 + std**2). This gives a more complete estimate and breakdown of the ALH error as derived from the evaluation with LIVAS ALHs. In addition, a scatter plot of OMI ALH vs LIVAS ALH would definitely help the reader a lot here. As a final note, I would need to do some more thinking on the implication of assuming climatological (i.e. averaged values) for your analysis. You make a strong point of ALH retrieval to support scatter corrections in trace gas retrievals. What would be needed then are ALH retrievals that capture exceptional events not represented by climatological averages, right? See point below.

-It is a pity that you only show only seasonal three-year averages. The temporal variability in such an average is small (discussed in a previous point). Figure 14 suggest that the std in ALP across pixels can easily be 200 hPa or higher. It would help if you could provide ALP maps for some selected scenes that show that within a particular scene there is spatial consistency but that the variability emerges when comparing different scenes. Perhaps together with some MODIS RGB images or other info..? I have to admit that I am not convinced at this stage that your ALP retrievals have indeed sufficient sensitivity to show geophysical variability - then again, it is an exploratory study. Such an addition would really strengthen the paper, but I realise that this requires substantial work. This is a strong recommendation but I will eventually leave it to the authors to decide whether or not they follow it up.

-In the introduction you quite extensively discuss the uncertainty in DOAS trace gas retrievals due to aerosols. This is fine with me. But in the abstract you clearly state that the 'main motivation of this study is to evaluate the possibility of retrieving ALH for potential future improvements of trace gas retrievals' (p.1.,l.4-5; repeated at the end of the intro and in conclusion). This is not what you do. Either statements concerning this claim should be softened (throughout) or a thorough and critical discussion of your results from the perspective of trace gas retrievals should be provided. The reason I make a point of this is the following: Figure 16 suggests that at least your seasonal average ALH is probably something like a boundary layer top height that perhaps follows

the expected seasonal dependence. But for trace gas corrections you need ALHs on a pixel level. Given the large uncertainties on individual retrievals I doubt whether the OMI ALH will then be useful: 0.9 km +/- 1 km in winter and 1.4 km +/- 1 km in summer does not appear to me as a tighter constraint for trace gas retrievals than simply assuming climatological boundary layer heights or something similar. Of course, there is Figure 14 showing that individual ALHs show quite some variability, but the question is whether the ALH variability for AOT below about one is really geophysical variability or indicates a large retrieval error. And of course, you also retrieve AOT itself, which may be useful for the trace gas retrievals, but also the AOT seems to be very sensitive to the aerosol model you assume.

-Finally, you state in the conclusion that accurate knowledge of AOT is needed for a good ALH retrieval. I am not yet convinced, because I think the comparison with LIVAS ALH is not conclusive in this respect (see comment above). The simulations however do seem to point in this direction (Figure 6). However, also MODIS AOT has an associated error which is well documented in the literature but not discussed at all. If you want to retain the conclusion that a retrieval setup with external AOT input is the way forward, then a discussion of the uncertainty in MODIS AOT should definitely be provided and also a small test on the sensitivity of your retrieved ALH to this AOT uncertainty should be added to section 4 (can be done fairly quickly I think).

Minor comments:

-The use of a neural network trained by a look-up table has been done before. I am aware of the ROCINN cloud algorithm, but I guess there must be more references. Please add some references.

-Can you provide quantitative estimates of the increase in speed and the reduced memory needs compared to the original look-up table?

-p.2,l.1: source (typo)

-p.2,l.5: clouds –> cloud

-p.2,l.5: Throughout the paper you often talk about fine particles, and it is not clear to me what you mean. Do you mean fine-mode particles? But also coarse-mode particles can act as CCNs? (throughout paper)

-p.2,l.8: contributes to –> contributes

-p.2,l.18: vertical distribution –> the vertical distribution

-p.2,l.19: affects the computation of Air Mass Factor –> affects trace gas Air Mass Factors

-p.2,l.22-23: uncertainties in the computed tropospheric NO2 AMF for OMI are the dominant source of errors –> you mean: uncertainties associated with aerosols?

-p.2,l.27: depending if –> depending on whether

-p.2,l.30: PBL –> throughout paper abbreviations are introduced several times (should be only first time), or abbreviations are introduced that are not further used (distracting); please check paper

-p.2,l.35: O2-O2 spectral band –> O2-O2 absorption band?

-p.3,l.1: what are effective cloud parameters?

-p.3,l.26-27: The 477 nm ... individual band. –> How many DFS does the O4 band add compared to continuum reflectances according to this study? That's the interesting number here.

-p.4,l.5: area –> areas

-p.4,l.7: emphasize –> emphasis

-p.4,l.11: the North –> North

-p.4,l.3: expectation –> benefit?

-p.4,l.19-20: during daylight –> can be left out

-p.4,l.20: two dimensional –> two-dimensional

-p.4,l.24: usually –> but is this also how refl. is defined in this paper?

-p.4,l.27-30: explanation of row numbers, row-anomaly and its progression can be completely left out, because it is not relevant; just say that all OMI data used in this study are from before the row-anomaly

-p.5,l.1-20: Discussion of OMAERUV out of place; should be completely moved to introduction. But why discuss it so extensively? Are you using the data in your study?

-p.5,l.25: The initial purpose of this [sic] algorithm... –> You should describe your algorithm here. Later I understood that you use slant columns and continuum reflectances from the cloud product, but that is not the algorithm that you use for the sensitivity studies, right? Just describe what you are doing and say that you follow the same approach as the OMI cloud product.

-p.5,l.22: extraction –> fit?

-p.5,l.23: fine spectral features –> add: due to absorption

-p.5,l.24: 460-490 nm –> is this the fit window?

-p.5,l.5: initial purpose –> what is the main purpose then? what is the purpose now?

-p.5,eq.2: Please give references for xsecs. I guess you are fitting the square of the oxygen slant column: I only know of O4 xsec reference spectra that have the equilibrium constant between [O2] and [O4] included. Why define reflectance explicitly in Eq. 1 when R in Eq. 2 is basically a sun-normalized radiance? I know that I am being nitpicky here (pi/mu_o disappears in the polynomial) but I prefer not to read unnecessary info.

-p.6,l.10-15: This entire para can be left out. You mention MAX-DOAS measurements,

the Ring effect, radiance measurements without further explanation so this doesn't add anything and only distracts the reader.

-p.6,l.18: Eq. (3) and Eq. (4) –> eqs 2 and 3 ! Please check entire manuscript on references to equations and figures in text (preferably before submission): there are more examples of wrong references, for example on p.14 and p.17

-p.6,l.20: what do mean with homogeneous and finite? An infinite(ly thick?) layer doesn't seem an option.

-p.6-7,sect.2.3: I have difficulty with this section and with figure 1. I think it is good to briefly describe the effect of aerosol parameters on slant columns and continuum reflectances. But the text and the figure are confusing. First, you are making forward references a couple of times. Please discus all relevant effects here, including effects of aerosol optical properties, which I think is important. Second, I cannot quite follow your use of the terms shielding and enhancement (see for example p.6,l.30-31, which I just don't understand) but this may be because I am not too familiar with these terms. Third, you are discussing slant column and continuum reflectance as a function of AOT, ALH, and aerosol optical properties. But these dependencies are difficult to recognize in figure 1 because you put the independent variable into the color map (and sometimes you again don't). Can you please make an alternative figure 1 that clearly illustrates the observations in the text.

-p.7,l.8: 2 –> two. Please check the AMT manuscript preparation guidelines before submitting, not only here, but throughout the manuscript (I also read 'Tab. 2'). And check for typos. I see various typos which I have no time to correct. I have seen three different ways of describing ranges ('2005-2007', '[260:800]', '[0.03-0.05-0.07]'). Also I read in several captions '(2005, 2006,2007)'. In my pdf, the greek symbol for tau in the plots is not recognizable as tau!

-p.7,l.23: from –> on

-p.8,l.8: Networks –> networks

-p.8,l.12-13: Don't agree: The dependence of slant columns and continuum reflectances on aerosol parameters can be accurately simulated with an RT model.

-p.8,l.17,l.22: use non-linear consistently (not: non linear) throughout manuscript

-p.9,l.14: The MLP ... data set –> don't understand this sentence

-p.9,l.30: referent –> reference (throughout manuscript)

-p.10,l.1: you have been referring several times now to average light path distributions, maybe you should define it because I don't quite know how exactly ALH affects this distribution and what this means for your retrieval

-p.10,l.2-3: The second ... quality. –> I am very confused: Are you following a two step approach for the NN_Rc_Ns retrieval? Fit AOT first which you then use as a fixed input for the ALP retrieval?

-p.10,l.15: what is the geometric thickness of your aerosol layer?

-p.11,l.1: most relevant –> optimal?

-p.11,l.4: positive-definite –> What does this mean? Your error function is just a plain sum-of-squared-residuals, right? Nothing special here?

-p.11,eq.7: why factor 1/2?

-p.11,l.18: you mean the last 15 iteration of 70% * 460000 iterations in total? What if overfitting is already present before the last 15 points?

-p.11,l.22: best evaluation score –> you mean for the evaluation set?

-p.11,l.28: The ALP retrieval scores are significantly larger. –> But this makes sense because pressures are two orders of magnitude larger than AOTs? So ALP performs even better (by a factor of hundred)?

-p.12,l.28-30: An overestimation ... surface. –> I am confused: a higher asymmetry parameter means more forward scattering (not: backward)...?

-p.13,l.3-17: I find it confusing that the way of analysing and presenting results is different when discussing the sensitivity to the error in the surface albedo. I would have preferred a plot in Figure 5 similar to Figures 4 and 6. Also, I don't understand the advantage of a box-whisker plot here: a single box-whisker is based on only three data points, right (three angles)? In addition, I would definitely like to know the direction of effects, particularly the effect of over- and underestimations of the surface albedo on ALP. If you want to retain the box-whisker plot (not preferred by me), then at least mention direction of effects in the text.

-p.13,l.19: You haven't discussed figures 5b and 5c yet!

-p.13,l.20-21: For scenes ... 100 hPa. –> But in this AOT range I see biases up to 300 hPa?

-p.13,l.25: entire learning database –> you mean: the independent data set?

-p.13,fig.6: Why are there large biases when there are no forward model errors (no error in SSA or g)? These are closed-loop, noise-free simulations so the truth should be retrieved?

-p.14,l.3: on –> in

-p.14,l.8-16: I don't really see a systematic improvement in ALP retrievals when using a priori AOT in case the aerosol model has a bias (for example, when g=0.8 yes but when g=0.6 no).

-p.14,l.18: eq. 11? Please check all figure ad equation numbers, there are more wrong references.

-p.14,l.22-26: This is an example of repetition; I would leave sentences out.

-p.14,l.26-28: Where do these SC precision values come from? Give ref. Are these

values typical for OMI instrument noise? But then for each scenario there is only one typical SC error as it only depends on the radiance and no range should be considered, right? How can a random error in SCs lead to ALP biases? Why don't you take into account the noise error in your other input parameter? Your remark on the temperature correction seems out of place.

-p.15,l.4: Can you give some more details here about the area that you selected (latitude and longitude? How many pixels passed selection? How many ALP data points went into the seasonal averages.

-p.15,l.19: Applying ... scenes. –> repetition

-p.15,l.21-22: However... as well. –> then why do you apply the OMI cloud fraction threshold...?

-p.15,l.25-: I find your discussion of the temperature correction and later sensitivity tests completely out-of-place (particularly as you show results of it in the final most important figure). Also, I don't understand how you do the temperature correction. Is Ns_O4-meas the slant column you get from the OMI cloud product? How do you use NCEP temperature profiles to calculate gamma? This factor clearly depends on the aerosol conditions, which you are trying to retrieve? I guess the physics of this temperature correction is well understood, apparently not part of the OMI cloud product and Veefkind et al. 2016 describe that users should apply it. So applying the temperature correction is always better and doesn't present a source of uncertainty for your ALH retrieval.

-p.15,l.8: geophysical location –> grid box?

-p.15,l.15: Note that ... –> So what does this imply?

-p.15-16,sect.5.2: I stopped reading this section in detail. A few general remarks: There are several forward references to figures hat have not yet been explained. Some figure numbers are incorrect. When investigating the effect of the surface albedo input

on the retrieved AOT, I don't see the reason why you should split the analysis per season (perhaps personal taste). Are MODIS AOT and MODIS BSA truly independent?

-p.17,l.27: polarization –> Your RT calculations for the LUT excluded polarization effects?

-p.17,l.31-32: MODIS tau ... OMI tau. –> Can you provide refs?

-p.18,l.4-9: You discuss the variability in your set of ALH pixels, but what does this tell you? Isn't it obvious that the variability decreases when you use the setup with fixed AOT: there is one free parameter less? This doesn't show that those retrievals are more accurate. Isn't it obvious that the variability decreases with higher AOT as the aerosol signal is more stronger...? But there is an interfering effect of the geophysical variability! Perhaps low-AOT aerosols are typically closer to the ground (BL pollution) while high-AOT aerosols are extreme events that reach into the free troposphere? Don't know.

-p.18,l.20: Spatial average ... performed. –> How do you compute ALH from extinction profiles? How do you compute the spatial average? Do you take into account the sampling of your OMI pixels? I mean, if for a given location you have, say, twice as many OMI pixels in summer than in winter, do you take this into account when calculating the LIVAS average. Please at least explain exactly what you did.

-p.20,l.14-15: You state that the NN with AOT as a fixed input performs better. But comparing figure 16b and 16d, this holds for the red and purple lines (SSA 0.95), but the opposite is true for the green line (SSA 0.9) and the blue line is undecided. Yet, previously you argued based on the comparison of AOTS that an SSA of 0.90 is probably the more accurate value in three seasons. So I think this conclusion cannot yet be drawn.

-p.29,fig.1: What is azimuth difference (throughout)? Does H2O absorb in your fit window? Then why isn't it included in the DOAS fit? At what wvl should I interpret TAU?

-p.30,table2: I don't understand this figure. For example, the ALP test error for a given NN configuration is the sum-of-squared differences (eq. 7) for all the scenarios in the test set (the 15%). This is one number. But then you repeat the training three times. So you have three numbers. Did you calculate box-whiskers from only three points? That doesn't seem right.

-p.33,fig.4: In left panel, can you add also SSA = 1.0 simulation? You have tested positive and negative errors in the asymmetry parameter as well? Can you make axes in the right panel such that the 1-1 line corresponds to the diagonal?

-p.39,fig.10: I see many more lines in the plot than in the legend? Can you make the plot line colors the same for the same surface albedos in the top and bottom row? I don't think the differences between the OMLER and MODIS BSA are significant, do you?

-p.40-41,fig.12-13: It took me some time to realise that -for unclear reasons- these figures are split whereas they should be merged? The color map labels are not visible? The tau symbol is not a tau symbol?

---

## Referee Comment (RC2) · Anonymous Referee #2 · 10 Jan 2017

The paper examines retrievals of aerosol optical thickness and layer pressure (height) from OMI spectral measurements (or as applicable to other instruments) of the O2-O2 absorption band near 477 nm using a neural network approach. A detailed analysis is carried out using simulated data. The approach is then applied in different ways using OMI and MODIS data over land areas of Asia with relatively high aerosol loading and compared with a lidar-based data set (LIVAS).

This is a detailed paper that should be published in AMT. The paper is in general clearly written though there are a number of typos and grammatical issues that it is assumed

will be caught during the copy-editing of the manuscript. Only a very few are listed below. I agree with the comments of reviewer 1 and add some additional comments for minor revisions below (some of these may be duplicates).

The last sentence of the abstract - "This study shows the first encouraging aerosol layer height retrieval results over land from satellite observations of the 477 nm O2-O2 spectral band." - is correct as written. However, the authors may mention here that a previous study examined case study retrievals over ocean. This sentence may stick in the reader's head as this is a "first" implementation with real data (references are later given and it becomes more obvious that these are the first results shown with data over land). I had to go back and reread the sentence to find the over land part, which makes it correct.

It might be better to include up front a Data section with the various satellite data sets used (OMI, MODIS, LIVAS) rather than to mention them in different places (and not referenced at the first mention - LIVAS). As it is OMI is mentioned in its own section section with MODIS mentioned below in the Methodology section. It seems that MODIS is an important part of this study as it is important to get accurate ALH and perhaps it deserves more attention.

P. 2, L. 34, here and also elsewhere suggest to add e.g. before references as there are others not in this list.

P. 3., L. 20, please add Torres et al., 1998 before de Graaf reference, also suggest to add Torres et al., 2013 reference here and explain that monthly climatology of CALIOP aerosol heights are currently in use for determination of aerosol parameters from OMI UV measurements.

P.4, L. 2, please add appropriate references here (altogether, though they are listed above).

Section 2.3, 1st par., This information may go better in the introduction. It's not clear

why MAX-DOAS is mentioned specifically here (this sentence seems out of place and not necessary). May be useful also to mention the work of using O2 A-band to retrieve aerosol height (over ocean by e.g., Dubuisson et al. 2009) and discuss possible advantages of the O2-O2 band (lower surface albedos over land?) and also discuss availability of these bands on various sensors such as OMI, OMPS, GOME-2, TROPOMI.

P. 9, L. 6, I think "either" should be removed as it is confusing.

Sect. 5.1, Perhaps I missed it but I do not see where the area of North-East Asia is defined. Please give the latitude-longitude of the area studied and/or show it on a map. I believe the highly industrialized areas (where there is heavy aerosol loading) used in this study may also be referred to as South-East Asia, thus it can be confusing.

P. 15, The discussion of the pairing of OMI and MODIS is confusing. It should be made clear that the resolution of MODIS is 1 km or better, but that collection 6 aerosol products are available at either 3 or 10 km resolution and that you are using 10 km. Please mention that you are using dark target only if this is the case and mention the specific product name, e.g., MYD04_L2. I believe that data are provided within the 10 km grids if they have some amount of cloud free pixels, so there may still be clouds present within the given area even when MODIS data are reported. Also, the 10 km x 10 km areas should not be referred to as a pixel as this can be confused with native MODIS pixels. Does the MODIS geometric cloud fraction used come from the aerosol product? This should be clarified as there are multiple MODIS cloud detection algorithms.

P. 15, L 24, change of to or.

Please check the English meaning of referent. It is defined in most English dictionaries as a noun but used in the paper primarily as an adjective. I believe the word reference may serve better in most instances and also perhaps the word default.

P. 17, L. 19, accuracy of OMI tau retrievals with respect to MODIS.

[Figure]

P. 18, L. 29, remarkable agreement with respect to seasonal mean values

Table 1: What happens for surface pressures < 963 hPa. The neural net will be extrapolating. How well will it do this? Does this occur with the real data? Likewise, why not add a node with single scattering albedo of unity? Also, the maximum value of VZA for OMI is > 44.2 degrees, so why not include the full range?

Table 2: When providing values for delta NsO2-O2 it would be good to provide a percentage error for these for a given scenario (readers will not have a good idea as to how large these values are). Perhaps I missed it but doing a search of "table" doesn't turn up a reference to Table 2 from the text. The aerosol optical thickness error is quite large for a change in surface albedo of 0.05. If surface albedo errors are more of the order of 0.02 (as stated in the text) then perhaps this would be a more appropriate value to use.

All figures in general would benefit from larger fonts. Also the tau in the figures looks different enough from the tau in the text to be somewhat confusing.

Figure 3 caption is confusing. What exactly is the supervised data set (training-validation-test)?

Figure 4, again lines and symbols hard to distinguish. It would be helpful to mention in the caption that the scenarios for the lines with the dot symbols tend to tall on top of one another.

Figure 6: Something should be mentioned in the caption about the range of surface albedos (Alb) used (same for several other figures).

Figure 9: I don't see where it is stated that the dotted line is the 1:1 line.

Fig. 10: There are many lines on these plots. It would help the reader if the 1:1 lines were made thicker to distinguish them. There are backward brackets in the legends at the ends of lines.

---

## Author Comment (AC2) · 7 Feb 2017

The comment was uploaded in the form of a supplement:
http://www.atmos-meas-tech-discuss.net/amt-2016-352/amt-2016-352-AC2-supplement.pdf

---

## Author Response (AR1)

Review of An exploratory study on the aerosol height retrieval from OMI measurements of the 477 nm  $O_2$ - $O_2$  spectral band, using a neural network approach

The authors present a study exploring the retrieval of an aerosol layer height parameter from OMI measurements of the  $O_2$ - $O_2$  band in the visual wavelength range. They follow basically the same approach as the OMI  $O_2$ - $O_2$  cloud algorithm but use a neural network to replace the traditional look-up table. Sensitivities of the retrieval to assumptions related to the aerosol optical properties and the surface albedo are investigated. The algorithm setup has been applied to three years of OMI data and seasonal averages are compared with MODIS aerosol optical thickness and climatological aerosol heights from the LIVAS CALIPSO climatology.

In my opinion, the paper presents interesting and substantial work that in principle warrants publication in AMT. However, I feel that there are a number of major issues that should be addressed before I can commend publication. Major and minor comments are listed below. I would appreciate if these are addressed point-by-point and a diffversion of the manuscript is provided together with the replies. In addition, I struggled with the style and the structuring of the manuscript, which made it hard at times for me to quickly understand what the authors are trying to say. I give examples of this below. I realise that this is partly a matter of personal taste, but I strongly encourage the authors to use these comments to critically look at the entire manuscript. It would help the reader to better understand and appreciate the authors' valuable work.

We took into account all the comments and questions asked by Referee #1 below. We reformulated where necessary according to the remarks and question, in order to ensure a better clarification. More details on these reformulations are given below where appropriate.

**Major comments**

-A look-up table is generated and used to train a neural network, which is in turn used in the retrieval instead of the original look-up table. This lowers memory demands and increases computational speed. In my opinion this is a very good application of a neural network. However, in your extensive general discussion of neural networks you suggest that you use a neural network to resolve the ill-conditioning of the inverse problem. For example, p.7,l.30-p.8,l.2., particularly in p.8,l.16 and again in the conclusion on p.21,l.6. This is clearly not what you do. Your neural network is trained with simulated data, so all the

physical relations between input and output contained in your neural network are already explicitly described in the RT model, and strong assumptions on the aerosol model, the vertical distribution of the aerosols and the ground surface are still a priori inputs for your retrieval. I think the text should be changed to really avoid giving the reader this wrong impression. Also, I don't quite see why interpolation in a look-up table should be less accurate than 'interpolation' in a look-up table with a neural network even in the case of non-linear data.

We thank the reviewer for this comment! We verified what we wrote in these mentioned pages. And we only found the reference to "ill-conditioning of the inverse problem" on p.7, l.30 "The Multilayer Perceptron (MLP) neural networks have been widely used and acknowledged for decades in the field of remote sensing (Atkinson and Tatnall, 1997). Indeed, most retrieval problems in this field are ill-posed and nonlinear". On p.8,l.16 and p.21,l.6 we do not address the ill-conditioning problems through the NNs but compare the advantages of Neural networks (NN) with respect (w.r.t.) Optimal Estimation Method (OEM) in terms of high speed processing and optimized interpolation techniques:

- "The choice of a NN approach relies on their advantages compared to more conventional methods such as linear regression, linear interpolation in a LUT or the Optimal Estimation Method (OEM). In particular, it enables 1) very fast computations with modern computers in spite of the number of required parameters, 2) optimized interpolation technique even in case of non linear problems as the NNs are better able to reproduce the curvature between the LUT nodes therefore lead to lower systematic biases compared to a linear interpolation, 3) reduced memory use compared to a LUT with a very high sampling" (cf. p.8, l.14-19).
- "In spite of the high computing time due to the learning database creation and the training of these algorithms, very fast operational processing is allowed. Such processing is much faster than approaches relying on the Optimal Estimation Method and employs more optimized interpolation techniques than a classical linear interpolation within a LUT" (*cf.* p.21, l.6).

Based on the many references available and cited here, these aspects are correct.

We do not mean that the use of a NN, like we did, explicitly resolve the ill-conditioning aspect of the inverse problem: "The Multilayer Perceptron (MLP) neural networks have been widely used and acknowledged for decades in the field of remote sensing (Atkinson and Tatnall, 1997). Indeed, most retrieval problems in this field are ill-posed and nonlinear, in particular the atmospheric ones. Thus, the associated inverse problems can only be addressed by including a priori information and relying on statistical analysis. Since aerosol retrieval from passive spectral measurements is well-known as a nonlinear inverse problem, the MLP technique represents then a powerful approach to design a retrieval algorithm in a fast and robust way." (cf. from p.7, l.29 to p.8, l.2). However, at a high level, similarly to the OEM approach, it deals (or can deal) with this aspect by relying on prior information, either explicitly (i.e. prior input parameter given to the retrieval algorithm) or implicitly by assuming some assumptions: typically the choice of our NNs rely on our assumed Single Scattering Albedo (SSA) value, either 0.9 or 0.95. Nevertheless, the way of how to deal with all these prior information differs depending on the numerical approach which is employed: For example, in the OEM, prior parameters are generally given with associated prior errors while these errors are not used by NN or Look-Up-Table (LUT) approaches.

To be sure to avoid any confusion, we clarified this issue in our manuscript.

A conventional interpolation within a LUT is strongly sensitive to: 1) the interpolation technique (e.g. linear interpolation, spline cubic interpolation), 2) the resolution or sampling of your LUT. Regarding 1), there is in general very little arguments to support the interpolation technique selection. In general, the linear interpolation is likely the fastest interpolation technique and least dependent on (arbitrary) specifications such as which dimensions are interpolated first and which last. Regarding point 2), for a given problem, a very coarse LUT can lead to substantial numerical biases independent of your

interpolation technique, like with the previous version of the OMI cloud LUT [Chimot *et al.*, 2016; Veefkind *et al.*, 2016]. In principle, we would say that an infinite high resolution LUT could do as good as a Neural Network. But, such a LUT is impractical. If 1) the training database includes a high enough number of cases to learn (here OMI  $O_2$ - $O_2$  and aerosol spectra simulations), 2) it is representative enough of the most likely cases to be encountered by the retrieval system, 3) the number of cases is greater than the number of degrees of freedom of the problem and the variability characterized by the combination of all the cases, then a NN is expected to be more accurate than a coarse (or too limited resolution) LUT.

Inverse problems with a LUT should, in general, demonstrate that the resolution specified in all the dimensions (and the combined dimensions altogether) is high enough for a given problem. This is, in general, very difficult to establish.

- An important figure is Figure 16. You are showing the seasonal dependence of ALH for your OMI retrievals and as derived from many years of CALIPSO retrievals. The amplitude of the seasonal variability in the LIVAS ALHs is only about 0.5 km and I am wondering whether that seasonal variability is picked up by your OMI retrievals. After all, biases due to wrong assumptions on the aerosol model are of the same order of magnitude or even larger. What makes Figure 16 a bit deceptive in my view is that you plot four lines showing OMI ALP retrievals for slightly different settings. These lines are obviously dependent and therefore correlate. At first sight, the overall consistency between the collection of plot lines suggested to me that you are indeed picking up the seasonal variability. But when having a closer look, I am not sure... the apparent consistency may be visually driven by these OMI ALP lines. In practice, you would pick some optimal retrieval setting based on sensitivity studies, literature etc. and do the retrieval, or you would perhaps run the retrieval for several retrieval settings and then take the average. I really think that for a proper comparison of LIVAS ALHs with your OMI ALHs, a figure like Figure 16 should therefore basically contain one panel with only two lines: one for LIVAS ALP and one for your best OMI ALP. Can you change Figure 16 (the effect of temperature corrections is not a key point of your paper and the effect of surface albedo and SSA could then be moved to a separate plot)? I fear that the agreement between OMI ALH and LIVAS ALH will look less convincing with only four pairs of points, but it is more realistic.

Thanks for your good remark here! This allows us to emphasize better the key points messages of this figure.

We modified Figure 16 in such a way that the impact of the mentioned parameters (detailed below) are now more explicitly and easily visible. We only have 4 panels, each panel showing results:

- for 1 specific NN retrieval algorithm, either based on the OMI continuum reflectance (475 nm) Rc (cf. NN\_Rc\_NsO2-O2), or based on MODIS aerosol optical thickness, or  $\zeta$  (550 nm), Dark Target Land as a prior information (cf. NN\_ $\zeta$ -NsO2-O2).
- for 1 specific prior surface albedo database: either OMLER, or the MODIS Black Albedo.

Each of these panels only represents 1 line with 4 points associated with the Aerosol Layer Height (ALH) of LIVAS per season and, over it, the spread of our spatial-seasonal averaged OMI ALH retrievals between our 2 aerosol Single Scattering Albedo (SSA or  $\omega$ 0) assumptions: *i.e* a lower limit of 0.9, an upper limit of 0.95. This spread is coloured in grey, with the upper limit depicted in red while the lower

limit is depicted in blue. All these panels depict retrievals where a temperature correction has been applied to the OMI  $O_2$ - $O_2$  slant column densities (SCD), named here after NsO2-O2 prior to the retrievals.

To set up the best retrieval setting for each individual OMI  $O_2$ - $O_2$  observation is, at this stage, impossible as it requests to use an accurate aerosol model. By an accurate aerosol, we mean here the one describing accurately (in particular) the optical properties of the particles (*e.g.*  $\omega$ 0). However, 1) no accurate  $\omega$ 0 information is available for each OMI individual observation pixel, and 2)  $\omega$ 0 likely significantly varies spatially and temporally. Instead, we designed and trained one NN algorithm for each assumed  $\omega$ 0 value. And then, these algorithms are applied directly over all the OMI pixels. This is equivalent of assuming a constant  $\omega$ 0 value for the ensemble of all OMI  $O_2$ - $O_2$  observations. Note that the chosen  $\omega$ 0 values are based on literature mentioned in our manuscript.

Assuming "only" two  $\omega 0$  values is not representative enough to perform an average. Therefore, we think it is more appropriate to depict the gap of the spatial-seasonal OMI ALH average due to the 2 (lower-upper) assumed  $\omega 0$ , and the location of the LIVAS black line within this gap.

For these 4 panels, we can confirm all the following elements:

- assumptions on the forward aerosol model (for the creation of the supervision database) lead to the highest impacts on the ALH retrievals. This is mostly related to the ability of the corresponding NN to interpret the combined scattering and absorption of aerosols affecting the NsO2-O2. Assuming OMLER surface albedo, differences between average ALH retrievals with  $\omega 0=0.95$  and  $\omega 0=0.9$  are in the range of 540-1200 m with NN\_Rc\_NsO2-O2, and 560-660 m with NN\_  $\zeta$ \_NsO2-O2.
- assumptions on chosen surface albedo (OMLER  $\nu s$ . MODIS Black Sky Albedo) also change the spatial-seasonal averaged ALH retrievals. Related differences in the retrievals are in the range of 0-730 m with NN\_Rc\_NsO2-O2, and 0-180 m with NN\_ $\zeta$ NsO2-O2.
- Although not shown here anymore, we remarked that applying the temperature correction on  $NsO_2-O_2$  is crucial as it corrects the retrievals between 50 m and 300 m.

In general, largest differences between the spatial-seasonal averaged ALH and LIVAS are found with  $\omega 0$  = 0.95, in the range of 650-1140 m with NN\_Rc\_NsO2-O2, and 260-800 m with NN\_  $\zeta$ \_NsO2-O2. With  $\omega 0$  = 0.9, differences are reduced to 180-310 m regardless the NN algorithm.

Independently of the chosen NN configuration, the impacts due to assumptions on  $\omega 0$  (and in general, the aerosol model) are the most crucial ones. They lead to the highest biases on the ALH retrievals. The use of two different surface albedo dataset (cf. OMLER and MODIS Black Sky), although not negligible, has somehow reduced impacts.

Because of the reduced gaps between the ALH retrievals with  $\omega 0 = 0.9$  or 0.95, the more reduced impacts related to the chosen surface albedo database, and the reduction of the encountered maximum differences with respect to (w.r.t.) LIVAS ALH, NN\_  $\zeta$ \_NsO2-O2 is actually the algorithm depicting the best ALH retrieval results. This is likely the case because this configuration uses more accurate prior aerosol  $\zeta$ (550 nm) information (coming from MODIS  $\zeta$  Dark Target land product). Please see more details about this specific point below (our answers to your point number 4).

Figure 17: Comparison of the average of the OMI aerosol layer height (ALH) retrievals obtained over scenes with collocated MODIS ζ(550 nm) > 1.0, with the LIVAS ALH climatology database. The retrievals are achieved over North-East Asia for cloud-free scenes, over 3 years 2005-2007, and for the 4 seasons. OMI and MODIS Black Sky surface albedos are considered alternatively. The 2 NN OMI ALH algorithms are used (see Sect. 5.1): NN\_Rc\_NsO2-O2 based on OMI Rc(475 nm) and ) NN\_ ζ\_NsO2-O2 based on MODIS ζ (550 nm). A temperature correction is applied to the OMI NsO2-O2 prior to the retrievals (*cf.* Sect 6.1): (a) NN\_Rc\_NsO2-O2 algorithm and OMLER surface albedo, (b) NN\_Rc\_NsO2-O2 algorithm and MODIS Black Sky surface albedo, (c) NN\_ ζ\_NsO2-O2 algorithm, and OMLER surface albedo.

-In your analysis you assume that climatological CALIPSO profiles are the truth and you focus then on biases. I am willing to follow your assumption for the moment but I think presenting the story like this is too optimistic on the error in retrieved ALH for an individual pixel (which is what you would need for scatter corrections). (Again, I am making a point of this because you repeat bias values in the abstract, p.1.,I.9-10.) In the sensitivity analysis you identify several error sources that affect retrieved ALP (aerosol optical properties, surface albedo, neural network implementation). Together they form something more like a random error. When calculating biases as you do these random errors are averaged out. In my view, you should also report the root-mean-square and the standard deviation of the differences (as in: rms\*\*2 = bias\*\*2 + std\*\*2). This gives a more complete estimate and breakdown of the ALH error as derived from the evaluation with LIVAS ALHs. In addition, a scatter plot of OMI ALH vs. LIVAS ALH would definitely help the reader a lot here. As a final note, I would need to do some more thinking on the implication of assuming climatological (i.e. averaged values) for your analysis. You make a strong point of ALH retrieval to support scatter corrections in trace gas retrievals. What would be needed then are ALH retrievals that capture exceptional events not represented by climatological averages, right? See point below.

Because of the climatology nature of the LIVAS database and of our OMI ALH retrievals, their difference in terms of sampling (which leads then to additional representation errors) and their associated spatial-seasonal average, we do not think it is very appropriate to depict a RMS as you are proposing. One of the reasons, as mentioned above, is that applying a constant  $\omega 0$  value to the whole ensemble of OMI observations may create specific spatio-temporal patterns in the retrievals. Instead, by emphasizing more the spreads in our averaged ALH retrievals (as done above, *cf.* Figure 17), are likely more representative of the stability and overall quality of the ensemble of our retrievals.

Furthermore, incorrect physical assumptions on surface albedo or aerosol properties would naturally lead to a systematic on an individual ALH retrieval (*i.e.* contrary to a random noise, the ALH error is reproducible). The integrated uncertainty related to the physical error sources and the instrument noise cannot be accurately estimated for each individual OMI pixel. Indeed, it depends on the cases, the real aerosol amounts and properties, their spatial patterns and temporal variability. The different academic cases done in Sec.5 allow to identify and quantify how each error source theoretically contributes to the retrieval error. The averaged real retrievals in Sec.6 and their comparison with LIVAS gives a first global idea of their consistency assuming then LIVAS is a reference. Overall, by combining the comparisons shown in Fig.17 above and the associated standard deviations in Fig.16 (below) allow to state that maximum differences are found on the seasonal and spatial averaged ALH with  $\omega 0 = 0.95$ , in the range of 260-800 m  $\pm$  1 km. Lower differences are in the range of 180-310 m, and similar variability, with  $\omega 0 = 0.9$ . This is a global estimate over the considered region as we may not have enough knowledge, at this stage, an adequate estimate of the retrieval error of each individual OMI pixel.

Figure 1 below shows, just as illustration here, how our OMI ALH retrievals spatially vary along the latitude for one day over North East China on 2006.10.02 while a strong aerosol pollution plume was present. We can see that the associated spatial variability is about 500 m on that day, with a maximum peak-to-peak of  $\sim$ 1 km. The accuracy of this synoptic variability is currently under analyses for another study.

Since the mentioned differences can be due to retrieval errors on both sides as well as to spatial representativity errors, we realize that using the term bias is not exact. We therefore removed in our updated manuscript this term. We mentioned now only the "differences" and the diverse error sources that contribute.

Please see our answers to your remarks on climatology vs. exceptional events below.

Figure 1: OMI Aerosol layer Height (ALH) retrieved on 2006.10.02 over North-East Asia along a CALIPSO track. These retrievals are obtained from OMI cloud-free scenes, with collocated MODIS  $\zeta(550 \text{ nm}) > 0.75$  and within a distance of 50 km of CALIPSO track.

-It is a pity that you only show only seasonal three-year averages. The temporal variability in such an average is small (discussed in a previous point). Figure 14 suggest that the std in ALP across pixels can easily be 200 hPa or higher. It would help if you could provide ALP maps for some selected scenes that show that within a particular scene there is spatial consistency but that the variability emerges when comparing different scenes. Perhaps together with some MODIS RGB images or other info..? I have to admit that I am not convinced at this stage that your ALP retrievals have indeed sufficient sensitivity to show geophysical variability - then again, it is an exploratory study. Such an addition would really strengthen the paper, but I realise that this requires substantial work. This is a strong recommendation but I will eventually leave it to the authors to decide whether or not they follow it up.

As extensively discussed in our manuscript, the possibility to derive an aerosol height from a satellite OMI-like  $O_2$ - $O_2$  visible absorption band is a quite challenging and important topic. In particular for the air quality retrieval community which has discussed and thought about it during the last 15 years (and perhaps even longer). If we leave aside the effective cloud parameters and their more or less sensitivity to the aerosol scattering effects and their altitude, at our knowledge, only Park *et al.*, (2016) with a Look-Up-Table (LUT) with about 10 case studies over ocean, and our manuscript with NNs over land in North-East Asia have retrieved an explicit aerosol height parameter from the OMI  $O_2$ - $O_2$  477 nm band.

Comprehensive and detailed investigations of the performances of our retrievals will require in-depth analyses combining different scales and strategies to fully understand our potential new OMI product. They would mix: global scale, long-time observation periods, focus on regional and/or local areas, day-to-day or pixel-by-pixel variability, statistical analysis to characterize the general behaviour of the algorithms over an ensemble of data, careful investigations of each single observation over a well-know and characterized case study.

In our view, addressing all these strategies together is out of the scope of this very first study and will require more than one manuscript. Instead, for our first paper presenting our algorithms, we chose to address an ensemble of 3-year OMI observations covering a somehow large industrialized land region (North-East Asia). This ensemble is analysed through a statistic strategy in order to characterize the general performance and main sensitivities of our product. Our general idea is that if a statistic analysis

already shows a good global performance and seasonal patterns, then this would encourage to continue this work by focusing on smaller scales (regional, local, reference daily case studies) and further investigating the behaviour of individual OMI retrievals on a pixel basis.

Overall, our manuscript on this topic shows first encouragements about the feasibility of retrieving aerosol height from OMI. Therefore, we are planning to publish follow-up studies on the same topic, but with complementary strategies. In fact, we can already say that we have started a study comparing OMI aerosol height retrievals over land, on some specific case studies (*e.g.* strong urban pollution & biomass burning events) with CALIPSO aerosol measurements along-track. We are planning to publish these results in the forthcoming months in a new journal paper.

Regarding your remark about MODIS RGB images, we do not think this is applicable. These images can tell us where are aerosol plumes and, potentially, how thick they are. But they do not give any information about aerosol altitudes.

-In the introduction you guite extensively discuss the uncertainty in DOAS trace gas retrievals due to aerosols. This is fine with me. But in the abstract you clearly state that the 'main motivation of this study is to evaluate the possibility of retrieving ALH for potential future improvements of trace gas retrievals' (p.1.,l.4-5; repeated at the end of the intro and in conclusion). This is not what you do. Either statements concerning this claim should be softened (throughout) or a thorough and critical discussion of your results from the perspective of trace gas retrievals should be provided. The reason I make a point of this is the following: Figure 16 suggests that at least your seasonal average ALH is probably something like a boundary layer top height that perhaps follows the expected seasonal dependence. But for trace gas corrections you need ALHs on a pixel level. Given the large uncertainties on individual retrievals I doubt whether the OMI ALH will then be useful: 0.9 km +/- 1 km in winter and 1.4 km +/- 1 km in summer does not appear to me as a tighter constraint for trace gas retrievals than simply assuming climatological boundary layer heights or something similar. Of course, there is Figure 14 showing that individual ALHs show quite some variability, but the question is whether the ALH variability for AOT below about one is really geophysical variability or indicates a large retrieval error. And of course, you also retrieve AOT itself, which may be useful for the trace gas retrievals, but also the AOT seems to be very sensitive to the aerosol model you assume.

Your remark about the need for an aerosol scattering correction in DOAS trace gas retrievals is very relevant. Analysing what we really need for that purpose (*i.e.* a climatology database of a daily pixel-by-pixel ALH variability) and whether we can use the current ALH retrievals are clearly another study in itself, and therefore beyond the scope of this manuscript.

We realize then, although our motivation statements are clear and correct, they are possibly more related to our long-term objectives than to the present manuscript. Therefore, we softened our statements as follow:

**In the abstract:**

"[...] This study presents a first step of this long-term objective which evaluates, from a statistic point-of-view, an ensemble of OMI ALH retrievals over a long-time period of 3 years covering a large industrialized continental region. [...] "

**In the introduction:**

" [...] Since aerosol altitude, in addition to aerosol optical thickness  $\zeta$ , is one of the key parameters affecting the computation of AMF for trace gases retrievals such as NO2 (Leitão *et al.*, 2010; Chimot *et al.*, 2016), our long-term motivation is to evaluate the capability of retrieving it from the satellite O2 -O2 absorption band at 477 nm. This exploratory study is the first step and statistically analyses an ensemble of OMI observations over 3 year period (2005-2007) and covering a large industrialized continental region (*i.e.* North-East Asia). [...] "

**In the conclusion:**

"[...] The main objective of this work is first to evaluate the feasibility of a direct retrieval of this key aerosol parameter from a statistical point of view: *i.e.* over a long-time period and large industrialized continental area, and therefore a high number of observations. [...] Our study indicates that it is worthwhile to design and evaluate aerosol height retrieval algorithm exploiting the satellites 477 nm  $O_2$  – $O_2$  absorption band. Our long-term motivation is to evaluate the feasibility of replacing the effective clouds by more explicit aerosol parameters in the computation of trace gas AMF. This is relevant not only for OMI but for most of the UV-Vis satellite missions devoted to air quality monitoring. For that purpose, further analyses must be performed by focusing on significant geophysical variability cases: *e.g.* pixel-by-pixel variability over smaller regions. Furthermore, single OMI ALH retrievals should be compared with reference aerosol vertical profile measurements (ground-based and/or satellites) over some remarkable case studies."

Regarding the reviewer's remark about the ALH variability, we added in the updated Figure 16 (ex Figure 15) of our manuscript (see also below) the variability as extracted from the LIVAS database. The methodology for deriving LIVAS ALH and its variability are described further in this document, as response to one of the technical questions. On average, the spatial and climatology variability over North-East Asia is about 700 m. Our spatially and seasonally averaged OMI ALH shows similar variability in winter and autumn, but more variability and hence higher standard deviation in spring and summer (from 1100 to 1800 m). The OMI ALH variability is lower, and closer to LIVAS, when using the NN\_  $\zeta$ \_NsO2-O2 algorithm with the MODIS  $\zeta$ (550 nm) Dark target product.

Figure 16: Standard deviation of the OMI aerosol layer height (ALH) retrievals obtained for the cloud-free scenes with MODIS  $\tau$  (550nm)  $\geq$  1.0 over North-East Asia. The retrievals are done for cloud-free scene, over 3 years 2005-2007. Aerosol single scattering albedo  $\omega 0 = 0.95$  is assumed. OMLER and MODIS black sky surface albedo, and the NN configurations (N NRc,NsO2 -O2 and NNT,NsO2 -O2 with MODIS  $\tau$  (550 nm) as input) are alternatively considered (*cf.* Sect.6.1).

-Finally, you state in the conclusion that accurate knowledge of AOT is needed for a good ALH retrieval. I am not yet convinced, because I think the comparison with LIVAS ALH is not conclusive in this respect (see comment above). The simulations however do seem to point in this direction (Figure 6). However, also MODIS AOT has an associated error which is well documented in the literature but not discussed at all. If you want to retain the conclusion that a retrieval setup with external AOT input is the way forward, then a discussion of the uncertainty in MODIS AOT should definitely be provided and also a small test on the sensitivity of your retrieved ALH to this AOT uncertainty should be added to section 4 (can be done fairly quickly I think).

As discussed above and with our upgraded Figure 16, we confirm that we think our results derived with the NN\_  $\zeta$ \_NsO2-O2 algorithm using MODIS  $\zeta$ (550 nm) Dark target land present overall the best quality.

Because of all the uncertainties mentioned in our answers to your major comment #2, we do not think it is fully appropriate to look only at individual lines in Figure 16 (now Figure 17!). The apparent minor changes of the line associated with  $\omega 0 = 0.9$  and OMLER surface albedo from NN\_Rc\_NsO2-O2 and to NN\_ $\zeta$ \_NsO2-O2 cannot indicate a degradation. Instead, the spreads between the ALH retrievals due to aerosols and surface albedo assumptions and from a statistic point-of-view (*i.e.* spatial-seasonal average) are the main criteria evaluating the robustness of each NN algorithm. A second criterion is the comparison of the OMI ALH retrievals w.r.t. LIVAS ALH over the same region (see updated Figure 15 above). Because these gaps strongly reduce and the OMI & LIVAS ALH variability are closer by employing NN\_ $\zeta$ \_NsO2-O2 with MODIS  $\zeta$ (550 nm), they confirm that an accurate knowledge of  $\zeta$  is then necessary.

Such infomration reduces the freedom of the system and thus the impact of inaccurate assumptions driving the OMI  $O_2$ - $O_2$  observations.

In addition, we would like to highlight the following points:

- The ALH retrieval results from the NN\_  $\zeta$ \_NsO2-O2 algorithm statistically show a higher quality only because the MODIS  $\zeta$ (550 nm) product is used as input. Indeed, as explained in Section 2.3, both  $\zeta$  and ALH drive the NsO2-O2 magnitude. And these effects need to be distinguished. The use of the NN\_  $\zeta$ \_NsO2-O2 algorithm in itself does not guarantee a better performance than the NN\_ Rc\_NsO2-O2 algorithm.
- Since Rc is mainly driven by  $\zeta$ , using NN\_ Rc\_NsO2-O2 is at a first approximation equivalent to using NN\_  $\zeta$ \_NsO2-O2 with the OMI  $\zeta$ (550 nm) product (instead of MODIS) retrieved prior to ALH retrieval. However, as shown in our study, large uncertainties remain on our OMI  $\zeta$ (550 nm) due to incorrect assumptions on aerosol model and surface albedo. Since these uncertainties are larger than MODIS  $\zeta$ (550 nm) Dark target uncertainties, they impact the quality of our OMI ALH retrievals.
- Because uncertainties for our OMI  $\zeta(550 \text{ nm})$  product are larger than for MODIS  $\zeta(550 \text{ nm})$  Dark target product, performance of the NN\_ Rc\_NsO2-O2 algorithm is more limited. To increase this performance, our capability to interpret Rc(475 nm) must be improved by separating the effects of aerosol  $\zeta(550 \text{ nm})$ ,  $\omega 0$  and surface albedo. In summary, OMI  $\zeta(550 \text{ nm})$  retrieval from the O2-O2 visible continuum must be further developed.

We emphasized all these points in our manuscript where necessary.

The MODIS  $\zeta$  uncertainties are now discussed in the new Section 2.2 as follow:

"The expected error of MODIS DT  $\zeta$  is about +/- 0.05+15% over land (Levy *et al.*, 2013). The "Deep Blue" retrieval algorithm has been developed to complement the DT algorithm by retrieving  $\zeta$  over bright arid land surfaces (*e.g.* deserts). The typical associated uncertainties are about +/-0.03 on average (Sayer *et al.*, 2013)."

**Minor comments:**

-The use of a neural network trained by a look-up table has been done before. I am aware of the ROCINN cloud algorithm, but I guess there must be more references. Please add some references.

By default, as mentioned by our references, a NN is always created based on an ensemble of training or supervision dataset. The term LUT is not really used on this topic for this specific need. We added some recent references related to the work of Crevoisier  $et\ al.$  on  $CH_4$  and  $CO_2$  retrievals from IASI using a NN approach, and Di Noia  $et\ al.$  for Ozone and aerosol retrievals, and the work of cloud retrievals with ROCINN.

-Can you provide quantitative estimates of the increase in speed and the reduced memory needs compared to the original look-up table?

An exercise comparison with a LUT would be sensitive to the LUT settings, in particular its resolution and sampling. We can state that a NN is a bit faster since using a LUT requires to identify the nodes, in each

dimension, where to perform the interpolation for each single observation. On the contrary, a NN directly "interpolates" or perform its regularization on the given observation.

Similarly, the memory consumption of a LUT depends on the number of dimensions and nodes within the LUT. Therefore, this is difficult to give a full estimate of increase speed and reduced memory.

What we can say is that performing 3-year ALH +  $\zeta$  retrievals over the North East Asia takes about 20 min per season, by using only 1 single CPU. The memory is only consumed by the synaptic weights between each neuron (input - hidden layer number 1, hidden layer number 1 – hidden layer number 2, hidden layer number 2 – output layer). This takes between 100 and 300 Mo (a simple and rough estimate) when using 1 single NN algorithm.

However, training times are more expensive, but this is of less importance for an operational use.

-p.2,l.1: source (typo)

Ok corrected

-p.2,l.5: clouds -> cloud

Done

-p.2,l.5: Throughout the paper you often talk about fine particles, and it is not clear to me what you mean. Do you mean fine-mode particles? But also coarse-mode particles can act as CCNs? (throughout paper)

In this paper, we numerically define the size of particles through the Angstrom parameter  $\alpha$ . This parameter describes the spectral variation of the Aerosol Optical Thickness (AOT)  $\zeta$ . The higher this value, the higher the dependence on the wavelength is the  $\zeta$ . Typically,  $\alpha < 1$  suggests an optical of coarse particle type (such as dust).  $\alpha > 1$  suggests an optical dominance of fine particles (*e.g.* smoke). In our study, the entire supervision dataset used for training the neural networks contains simulations with  $\alpha = 1.5$  ( fine mode particles). The main reason is because of the assumed dominant presence of aerosols released by anthropogenic fossil-fuel based activities such as industries, power plants and traffic. Also, such a value is dominant in the MODIS aerosol product over the region of North-East Asia.

```
-p.2,l.8: contributes to -> contributes
```

We changed it as follows: "... are still the largest uncertainties of the total RF estimate ..." ("are" instead of "contributes").

-p.2,l.18: vertical distribution -> the vertical distribution

Ok

-p.2,l.19: affects the computation of Air Mass Factor -> affects trace gas Air Mass Factors

Ok

-p.2,l.22-23: uncertainties in the computed tropospheric NO2 AMF for OMI are the dominant source of errors -> you mean: uncertainties associated with aerosols?

Here, I mean that all the uncertainties in general included in the calculation of the tropospheric  $NO_2$  AMF are the dominant source of errors in the retrieval of the tropospheric  $NO_2$  Vertical Column Density. In addition, uncertainties related to the spectral fit of the slant column density also contribute to these retrieval errors. However, according to the mentioned literature, they are of lower magnitude than the errors associated with the AMF.

Of course, aerosols play a big part in the tropospheric  $NO_2$  AMF errors. But, uncertainties related to the shape of the  $NO_2$  vertical profile, clouds and surface albedo are also significant.

Overall, the magnitude of the error on the tropospheric  $NO_2$  column retrieval is over polluted areas mostly determined by the uncertainty of the air mass factor, and not by the uncertainty of the slant column fit (Boersma *et al.*, 2007). We reformulated accordingly.

```
-p.2,l.27: depending if -> depending on whether
```

Ok

-p.2,l.30: PBL -> throughout paper abbreviations are introduced several times (should be only first time), or abbreviations are introduced that are not further used (distracting); please check paper

We carefully checked that all abbreviations are only introduced once through the whole paper.

```
-p.2,l.35: O_2-O_2 spectral band -> O_2-O_2 absorption band?
```

Ok

-p.3,l.1: what are effective cloud parameters?

The OMI effective cloud parameters are the OMI cloud products retrieved from the same  $O_2$ -  $O_2$  visible absorption band assuming a simple opaque and Lambertian cloud model (single albedo = 0.8). They give the effective cloud fraction and effective cloud pressure. More can be read in Acarreta *et al.*, 2004; Boersma *et al.*, 2011; Veefkind *et al.*, 2016; and Chimot *et al.*, 2016.

-p.3,l.26-27: The 477 nm ... individual band.  $\rightarrow$  How many DFS does the  $O_4$  band add compared to continuum reflectances according to this study? That's the interesting number here.

The statement just before, in the manuscript, mentions that the  $O_2$  band at 477 nm adds between about 1 DFS according to the study of Veihelmann *et al.* (2007).

```
-p.4,l.5: area -> areas
Ok, done.
-p.4,l.7: emphasize -> emphasis
Ok, this is corrected.
-p.4,l.11: the North -> North
Ok
-p.4,l.3: expectation -> benefit?
```

Ok (I assume that I.13 was meant here, not I.3).

-p.4,l.19-20: during daylight -> can be left out

Ok

-p.4,l.20: two dimensional -> two-dimensional

Ok

-p.4,l.24: usually -> but is this also how refl. is defined in this paper?

For avoiding any confusion, in particular with the continuum reflectance (475 nm), we removed this equation which not used in the next parts of this study.

-p.4,l.27-30: explanation of row numbers, row-anomaly and its progression can be completely left out, because it is not relevant; just say that all OMI data used in this study are from before the row-anomaly

Ok

-p.5,l.1-20: Discussion of OMAERUV out of place; should be completely moved to introduction. But why discuss it so extensively? Are you using the data in your study?

We removed all the details about OMAERUV that are not used or helpful for this study. We left only the mention of the existence of this algorithm and its general performance (i.e. for  $\zeta$  retrieval).

-p.5,l.25: The initial purpose of this [sic] algorithm... -> You should describe your algorithm here. Later I understood that you use slant columns and continuum reflectances from the cloud product, but that is not the algorithm that you use for the sensitivity studies, right? Just describe what you are doing and say that you follow the same approach as the OMI cloud product.

In order to make it more clear, we reformulated the associated (new) Sect.3.1 as follow:

« In this paper, the aerosol Neural Network (NN) retrieval algorithms allow the conversion of the continuum reflectance Rc(475nm) and the  $O_2-O_2$  SCD Ns $O_2-O_2$  into  $\zeta$  and ALP (in hPa). As a consequence, the NN retrievals rely on the way how the aerosol parameters modify these two variables and thus the photons average light path.

Prior to this conversion, a spectral DOAS fit must be performed to derive Rc(475nm) and  $NsO_2-O_2$  from the OMI  $O_2-O_2$  477 nm absorption band. The various DOAS techniques rely on the same key concept: a simultaneous fit of several trace gas slant column densities from the fine spectral features due to their absorption (*i.e.* the high frequency part) present in passive UV-Vis spectral measurements of atmospheric radiation (Platt and Stutz, 2008). Here, the DOAS fit follows the same approach as in the OMI  $O_2-O_2$  cloud algorithm (Acarreta *et al.*, 2004; Veefkind *et al.*, 2016): i.e. the absorption cross-section spectrum of  $O_2-O_2$  is fitted together with a first order polynomial:

```
-\ln(R(\lambda)) = \gamma 1 + \gamma 2 \cdot \lambda + \text{NsO}_2 - \text{O}_2(\lambda) \cdot \sigma \text{O}_2 - \text{O}_2 + \text{NsO}_3(\lambda) \cdot \sigma \text{O}_3(\lambda)
```

where  $\gamma 1 + \gamma 2$  ·  $\lambda$  defines the first order polynomial,  $\sigma O_2 - O_2$  and  $\sigma O_3$  are the  $O_2 - O_2$  absorption cross-section spectrum and the  $O_3$  absorption cross section spectrum respectively, convoluted with the OMI slit function, and NsO3 is the O3 slant column density.  $\sigma O_2 - O_2$  is based on measurements of the cross

section made by C. Hermans (see http://www.aeronomie.be/spectrolab/o2.htm - file O4.txt). The  $O_3$  cross section spectrum is included because it overlaps with the  $O_2-O_2$  spectrum. The fitted parameters are  $\gamma 1$ ,  $\gamma 2$ ,  $NsO_2-O_2$ , and  $NsO_3$ . In the absence of absorbers, one may define the continuum reflectance Rc at the reference wavelength  $\lambda 0$ :

Rc =
$$\exp(-\gamma 1 - \gamma 2 \cdot \lambda 0)$$
. (2)

The reference wavelength is specified as the middle of the DOAS fit window at  $\lambda 0 = 475$  nm."

-p.5,l.22: extraction -> fit?

Ok

-p.5,l.23: fine spectral features -> add: due to absorption

Added

-p.5,l.24: 460-490 nm -> is this the fit window?

Yes, this is. We added this clarification.

-p.5,l.5: initial purpose -> what is the main purpose then? what is the purpose now?

Since the reformulation of this section (see above), these two words are now removed.

-p.5,eq.2: Please give references for xsecs. I guess you are fitting the square of the oxygen slant column: I only know of  $O_4$  xsec reference spectra that have the equilibrium constant between [O2] and [O4] included. Why define reflectance explicitly in Eq. 1 when R in Eq. 2 is basically a sun-normalized radiance? I know that I am being nitpicky here (pi/mu\_o disappears in the polynomial) but I prefer not to read unnecessary info.

The  $O_2$ - $O_2$  xsec that we have used is based on measurements of the cross section by Christian Hermans (http://www.aeronomie.be/spectrolab/o2.htm - file O4.txt).

According to your previous comments, the here mentioned Eq. 1 is now removed to avoid any confusion.

Yes indeed, we indeed fit the square of the oxygen slant column density because of the use of the  $O_4$  xsec reference spectra.

-p.6,l.10-15: This entire para can be left out. You mention MAX-DOAS measurements, the Ring effect, radiance measurements without further explanation so this doesn't add anything and only distracts the reader.

Ok, this is removed.

-p.6,l.18: Eq. (3) and Eq. (4)  $\rightarrow$  eqs 2 and 3! Please check entire manuscript on references to equations and figures in text (preferably before submission): there are more examples of wrong references, for example on p.14 and p.17

We carefully verified the equation label numbers and their use / mention though the entire manuscript. Thanks for your remark!

**-p.6,l.20: what do mean with homogeneous and finite? An infinite(ly thick?) layer doesn't seem an option.**

By "homogeneous", we mean that all the particles present in the scattering layer have exactly same properties (*e.g.* single scattering albedo, size through the Angstrom parameter etc..).

By "finite", we want to emphasize that our aerosol layer is approximated by a single box layer which has then, by definition, a finite geometric thickness of about 1 km. Therefore, we assume that no particles are present above and below this box layer. This is opposite to a full vertical profile, described over the entire atmospheric column, which could be visualized then as multiple box aerosol layers with diverse thickness and optical properties.

For more clarity, we reformulated this part as follow:

"In this paper, the aerosol layer is assumed to be one single scattering layer (*i.e.* "box layer") with a constant geometric thickness (about 1 km). All the particles included in this layer are supposed to be homogeneous (*i.e.* same size and optical properties). The aerosol layer height (ALH) is then expressed by the aerosol layer pressure (ALP), in (hPa), defined as the mid-pressure of this scattering layer."

-p.6-7,sect.2.3: I have difficulty with this section and with figure 1. I think it is good to briefly describe the effect of aerosol parameters on slant columns and continuum reflectances. But the text and the figure are confusing. First, you are making forward references a couple of times. Please discus all relevant effects here, including effects of aerosol optical properties, which I think is important. Second, I cannot quite follow your use of the terms shielding and enhancement (see for example p.6,l.30-31, which I just don't understand) but this may be because I am not too familiar with these terms. Third, you are discussing slant column and continuum reflectance as a function of AOT, ALH, and aerosol optical properties. But these dependencies are difficult to recognize in figure 1 because you put the independent variable into the color map (and sometimes you again don't). Can you please make an alternative figure 1 that clearly illustrates the observations in the text.

Following your suggestion, we changed the figure 1 as follow (see below as well the new figure 1 please): we put the measurement variables into the color maps, while the physical variables (aerosol optical thickness and aerosol layer pressure - ALP) are out of them. In that way, the reader can better see how aerosols (both  $\zeta$  and their altitude expressed in pressure) directly drive the OMI  $O_2$ - $O_2$  measurements. Furthermore, we added 2 panels with different aerosol SSA and surface albedo properties. Then, it becomes more clear how these properties, in addition of  $\zeta$  and ALP, affect these measurements.

Finally, we reformulated in part the referred section to remove all the confusions as follow. The new section is now written as follows:

"Figure 1 illustrates how aerosol particles drive the OMI  $O_2-O_2$  DOAS parameters at 477 nm assuming cloud-free space-borne observations. These effects are obtained from radiative transfer simulations including aerosols and no clouds. The detailed generation of such simulations is given in Sect. 3.2. The DOAS fit equations following Eq. (2) and Eq. (3) are then applied on these simulated spectra. In this paper, the aerosol layer is assumed to be one single scattering layer (*i.e.* "box layer") with a constant geometric thickness (about 1 km). All the particles included in this layer are supposed to be

homogeneous (*i.e.* same size and optical properties). The aerosol layer height (ALH) is then expressed by the aerosol layer pressure (ALP), in hPa, defined as the mid-pressure of this scattering layer.

Qualitatively, aerosols have two separate effects on the average light path, and therefore on the  $O_2-O_2$  absorption signal at the top of the atmosphere (TOA). These two effects are similar as aerosols and clouds have on  $NO_2$  absorption signal (Leitão *et al.*, 2010; Chimot *et al.*, 2016): 1) a shielding effect, *i.e.* a decreased sensitivity within and below the aerosol layer due to a reduced amount of photons coming from the TOA and reaching the lowest part of the atmosphere compared to an aerosol-free scene, 2) an enhancement (albedo) effect, *i.e.* an increased sensitivity within and above the aerosol layer as more photons are scattered back towards the sensors, the part of the atmosphere above the aerosol layer is then sampled by a larger fraction of detected photons. Shielding then leads to a reduced  $O_2-O_2$  absorption while enhancement may increase the  $O_2-O_2$  absorption especially for low cloud or aerosol layers. The overall effect (enhancement  $\nu s$ . shielding) depends on the aerosol optical properties, the total column  $\zeta$  and ALP.

The OMI Rc( 475 nm) is directly and primarily affected by the total column  $\zeta$  of particles present in the observed scene. Indeed, Rc increases with increasing  $\zeta$  independently of the ALP (cf. Fig. 1a). This mostly results from the influence of aerosols on the number of detected photons and on the additional scattering effects observed in the scene compared to an aerosol-free scene. However, the magnitude of this increase relies on aerosol optical properties and the surface brightness. As a consequence, Rc is also affected by aerosol  $\omega$ 0 , phase function, and the surface albedo A. Indeed, Rc decreases with decreasing  $\omega$ 0 and over a darker surface (*i.e.* smaller A value) for all the  $\zeta$  values (cf. Fig. 1c and Fig. 1e). The importance of these parameters is further discussed in Sect. 4 and Sect. 5. Note that, in addition, the reflectance is also driven by the geometry angles: *i.e.* viewing zenith angles  $\theta$ , solar zenith angles  $\theta$ 0 and relative azimuth angle  $\phi$  –  $\phi$ 0. An increase of  $\theta$ 0 or  $\theta$ 0 will lead to longer average light path, and thus will amplify aerosol related additional scattering effects (for a given  $\tau$ 1).

OMI NsO2–O2 relies on the O2–O2 absorption magnitude along the average light path in the whole atmosphere. It is driven by the overall shielding or enhancement effect of photons by the O2–O2 complex in the visible spectral range, due to the presence of particles. As depicted in Fig. 1b, NsO2–O2 decreases with decreasing ALP. This is a direct consequence of a larger shielding effect applied by aerosols located at higher altitudes (*i.e.* part of the O2–O2 complex located below the aerosol layers are shielded). Nevertheless, in case of low  $\zeta$  values (*i.e.*  $\leq$  0.5), NsO2–O2 does not significantly vary with respect to ALP. This shows that a low amount of aerosols has very little impacts on O2–O2 absorption measurements.

However, as depicted in Fig. 1b, 1d and 1f, not only ALP but also  $\tau$  directly influences the slant  $O_2-O_2$  absorption since both parameters simultaneously affect the photon path distribution, and therefore the overall shielding or enhancement effect. An increase of  $\zeta$  leads to a decrease of NsO2-O2. The slope of this decrease depends on the aerosol altitude (*i.e.* higher for particles at high altitude). Note that both  $\omega$ 0 and A also affect NsO2-O2, but this effect is smaller than  $\zeta$ . For example, a reduced  $\omega$ 0 and A lead to a small decrease of NsO2-O2 (*cf.* Fig. 1d and Fig. 1f).

**As a consequence:**

– the single parameter  $NsO_2-O_2$  contains information on both  $\zeta$  and ALP. These parameters cannot be separated from this unique variable alone. As a consequence, if  $\zeta$  is not accurately known, there will likely be an ambiguity when analysing  $NsO_2-O_2$  to retrieve ALP.

- if an external or prior  $\zeta$  estimate is not available, then the two parameters  $NsO_2-O_2$  and Rc could be simultaneously combined to retrieve ALP provided that one can accurately and independently retrieve  $\zeta$  from Rc. Then, in that condition, Rc may help to distinguish both  $\zeta$  and ALP contributions in  $NsO_2-O_2$ . However, the simultaneous effects of aerosol  $\omega 0$  and A on Rc (as discussed above), and therefore their associated uncertainties, will impact the feasibility of retrieving  $\zeta$  from OMI measurements. It may then degrade the retrieved ALP performances.
- $-\zeta$ , Rc and NsO2-O2 have a non negligible correlation. Indeed, an increase of  $\zeta$  results in a simultaneous increase of Rc and NsO2-O2. Therefore, it has to be noted that these two last parameters are not independent and combining them does not provide with two independent pieces of information.

Overall, the impact of aerosol particles on the OMI  $O_2-O_2$  spectral band is similar to cloud particles. This explains in part the difficulty to distinguish aerosols from clouds. In cases with a mix of aerosols and clouds, there is an ambiguity between Rc ,  $\zeta$  and the OMI effective cloud fraction on the one hand, and NsO2 $-O_2$ , the aerosol layer pressure,  $\zeta$ , the OMI effective cloud pressure and fraction on the other hand (Boersma *et al.*, 2011; Castellanos *et al.*, 2015; Chimot *et al.*, 2016). Therefore, this study only focuses on cloud-free reflectance to avoid this complexity."

Figure 1: Continuum reflectance Rc (475nm) and O2 -O2 slant column density NsO2 -O2 as a function of  $\tau$  and aerosol layer pressure for the following conditions: climatology mid-latitude summer temperature, NO2, O3 and H2 O profiles,  $\theta$ 0 = 32 $^{\circ}$ ,  $\theta$  = 32 $^{\circ}$ , surface pressure = 1010 hPa and fine aerosol particles ( $\alpha$  = 1.5, g = 0.7): (a) and (b) surface albedo = 0.07 and aerosol  $\omega$ 0 = 0.95, (c) and (d) surface albedo = 0.03 and aerosol  $\omega$ 0 = 0.95, (e) and (f) surface albedo = 0.07 and aerosol  $\omega$ 0 = 0.9.

-p.7,l.8: 2 -> two. Please check the AMT manuscript preparation guidelines before submitting, not only here, but throughout the manuscript (I also read 'Tab. 2'). And check for typos. I see various typos which I have no time to correct. I have seen three different ways of describing ranges ('2005-2007', '[260:800]', '[0.03-0.05-0.07]'). Also I read in several captions '(2005, 2006,2007)'. In my pdf, the greek symbol for tau in the plots is not recognizable as tau!

We corrected the writing of numbers where needed. We checked the recent guidelines. We acknowledged that the word "Table" should never abbreviated. We corrected accordingly (e.g. Table 2).

We homogenized the way of describing ranges of numbers in the entire manuscript as follow: 2005-2007, or 0.03-0.05-0.07.

-p.7,l.23: from -> on

Corrected.

-p.8,l.8: Networks -> networks

Ok

-p.8,l.12-13: Don't agree: The dependence of slant columns and continuum reflectances on aerosol parameters can be accurately simulated with an RT model.

In our sentence, "with accurate functions" we mean that no invertible analytical function exist that describes the dependence of slant columns and continuum reflectances on aerosols.

Instead, we can accurately simulate it through the processing chain radiative transfer model + spectral DOAS fit. But for the conversion to vertical columns one then employs an approximate inverse model such a conventional Look--Up-Table (LUT) or NNs (as we did).

In our sentence, we therefore replace "the link between the OMI  $O_2$ - $O_2$  measurements and aerosol parameters cannot be exactly described with accurate functions" by "no invertible analytical function exist that describes the dependence of slant columns and continuum reflectances on aerosols."

-p.8,l.17,l.22: use non-linear consistently (not: non linear) throughout manuscript

We corrected this in the whole manuscript.

```
-p.9,l.14: The MLP ... data set -> don't understand this sentence
```

This sentence means that a MLP NN algorithm needs a well-known ensemble of dataset in order to perform an accurate training. We reformulated this sentence.

-p.9,l.30: referent -> reference (throughout manuscript)

We corrected this in the whole manuscript.

-p.10,l.1: you have been referring several times now to average light path distributions, maybe you should define it because I don't quite know how exactly ALH affects this distribution and what this means for your retrieval

The notion of average light path, and how aerosols impact it, are detailed in the previous Section 2.3 through the so-called shielding and enhancement effects. Aerosols impact the length of the average light path depending on the altitude of atmospheric layers that are shielded and enhanced. Since this is clarified there, we don't think we need to repeat it here. We removed the word "distributions".

-p.10,l.2-3: The second ... quality. -> I am very confused: Are you following a two step approach for the NN\_Rc\_Ns retrieval? Fit AOT first which you then use as a fixed input for the ALP retrieval?

No. As described on p. 9 I5-6, while the NN\_  $\zeta$ ,\_Ns configuration uses a  $\zeta$  information as input for retrieving ALP, the NN\_Rc\_Ns uses the OMI continuum reflectance at 475 nm as input information, instead of  $\zeta$ . Therefore, no  $\zeta$  information is used with this algorithm.

Only the NN\_  $\zeta$  \_Ns algorithm uses an AOT information then. The choice of which AOT information use here is up to the user. Here, we made the choice of using MODIS AOT, as this provides the best information available. But, of course, it would be very possible to use the retrieved OMI AOTs as input instead.

One could argue that the NN\_Rc\_Ns implicitly uses the OMI  $\zeta$  information to retrieve ALP. Indeed, as explained in Section 2.3, when analyzing the O2-O2 SCD, it is crucial to be able to separate the effects due to aerosol height (or ALP) on the one hand, and those due to  $\zeta$  on the other hand. If a prior  $\zeta$  information from an independent instrument is not available, then the OMI continuum reflectance (475 nm) may be used to retrieve  $\zeta$ . The accuracy however strongly relies on assumptions about surface albedo and aerosol optical properties which drive as well the magnitude of Rc. If these assumptions are accurate enough, then one can retrieve a good OMI  $\zeta$  which could be then used by NN\_  $\zeta$ \_NsO2-O2 for an accurate ALP retrieval.

Since therefore, retrieving a very accurate OMI AOT theoretically means that one is able to precisely interpret the continuum reflectance (475 nm), one could in principle directly retrieve in one step OMI ALP by using the NN\_Rc\_Ns configuration. That would replace a 2-step approach consisting of firstly retrieving OMI AOT using the NN\_Rc\_Ns  $\zeta$  retrieval algorithm, and then secondly using the NN\_  $\zeta$  \_Ns ALP retrieval algorithm with the retrieved OMI AOT as prior information.

-p.10,l.15: what is the geometric thickness of your aerosol layer?

The thickness of the aerosol layer, considered in all the retrievals is 1 km. We described it.

-p.11,l.1: most relevant -> optimal?

This is changed.

-p.11,l.4: positive-definite -> What does this mean? Your error function is just a plain sum-of-squared-residuals, right? Nothing special here?

We just meant that computed error here has, by definition, a positive sign (because of the sum-of-squared-residuals). We removed this term to avoid any confusion.

-p.11,eq.7: why factor 1/2?

The equation given here, of the error minimization function during the training process of the Neural Network algorithm follows the more general equation as defined and critically specified by Rumelhart *et al.*, (1986) through the description of the Error back-propagation algorithm (described in the present previous sections).

The general error minimization equation given by Rumelhart et al., (1986) is the following:

$$E = \frac{1}{2} \sum_{c} \sum_{j} y_{j,c} - d_{j,c}$$

Where c is an index over the cases (input-output pairs) contained in the training database, j is an index over the output neurons, y is the retrieved value of an output neuron while d is the true expected output. This general equation differs by the one given in our paper by the number of sums, two sums here (over the output neurons and the training cases) while only one in our manuscript (over the training cases).

In theory, a Neural Network algorithm can contain more than one output neuron: *i.e.* a single NN algorithm can simultaneously retrieve more than one single parameter. In practice, because retrieving  $\zeta$  and ALP from the OMI  $O_2$ - $O_2$  absorption band does not exactly require the same input parameters, we made the choice of one unique output neuron for each of the developed NN algorithm: *i.e.* each algorithm only retrieves one parameter, either AOT or ALP. As a consequence, only one sum remains in our equation (the sum over the output neurons implicitly disappears because the upper limit of the index j is 1). Therefore, our equation keeps the sum over the training cases and the factor 1/2 from the more general error minimization equation.

Leaving out the factor 1/2 does not lead to different results, but that it (probably) originates from more general cases where the 1/2 weights this term compared to another addition to E.

-p.11,l.18: you mean the last 15 iteration of 70% \* 460000 iterations in total? What if overfitting is already present before the last 15 points?

No. The number 70% (mentioned in line 24) refers to the percentage of the initial learning dataset (i.e. 460 000 spectral simulations as described in section 3.2.) that is randomly extracted and then used for the training of each designed NN. So, about 322 000 simulations (=70% x 460 000) are then used to train the NNs. The verification and evaluation steps are achieved with the remaining learning dataset (15% each) to ensure independency. The training, step is about finding the optimal weights of the synaptic connections of each NN configuration.

As described in section 3.3., the training step is iteratively performed. Prior to the 1st iteration, the synaptic weights are randomly initialized. Then, at each iteration, they are changed through the back-propagation approach. By experience, the error function (cf. Eq. 7) is maximal at the first iteration and decreases with increasing number of iterations. Generally, through all the experiments, this error strongly decreases during the first 50 iterations. After that, the decrease magnitude is relatively low, although not negligible. Therefore, it is not expected (it is even extremely unlikely) that overfitting occurs over the first 15 iterations. Generally, the overfitting, when identified, only occurs after the first 100 iterations. Then, when the training system identified an overfitting (*i.e.* verification step), the training process is stopped.

**-p.11,l.22: best evaluation score -> you mean for the evaluation set?**

Yes. The so-called valuation score is the error as defined in the Error Equation computed over the evaluation dataset ( = 15% of the entire learning database).

-p.11,l.28: The ALP retrieval scores are significantly larger. -> But this makes sense because pressures are two orders of magnitude larger than AOTs? So ALP performs even better (by a factor of hundred)?

No. As written in the caption of the figure, the direct output (or retrieved) values from the NN algorithms are normalized between -1 and 1. Because of the nature of the sigmoid functions, used by our algorithms in the hidden layer, all the input parameters are normalized between -1 and 1 to avoid the saturation of these activation functions. Therefore, the outputs are also similarly normalized.

As a consequence, in the NN error back-propagation algorithm, all the trainings and error calculations are performed over normalized values. Then, the values of the total quadratic error E is based on same orders of magnitude. The plotted scores here do show then that ALP errors are, on average, larger than  $\zeta$  errors. This confirms than the ALP training score is, in theory, less reliable (and thus more complex) than the  $\zeta$  training score.

Note that we did not really emphasize too much the normalization step and its reason too much in the text of our manuscript, to avoid to dilute too much information. We think this is a technical detail which, except in the caption of figure 3, does not add any value in our analyses. However, if the associate editor thinks otherwise, we will add it then in Section 3.

-p.12,l.28-30: An overestimation ... surface. -> I am confused: a higher asymmetry parameter means more forward scattering (not: backward)...?

Yes, we agree. If the assumed asymmetry parameter is overestimated (*i.e.* g = 0.7, while the "true" g is 0.6), then we would expect to observe in the signal an increased forward scattering. Since, the true g is lower, there is actually less forward scattering (*i.e.* more back-scattering) than expected. Therefore, the retrieved aerosol optical thickness is biased negatively: i.e. the retrieved AOT is lower than the true AOT.

To ensure clarity, we slightly reformulated these sentences as follow:

"An overestimation of g (i.e. assumed g=0.7 while true g=0.6) leads to an increased retrieved  $\zeta$  value (*i.e.* positive bias) as a result of less photons scattered towards the surface, and therefore more photons scattered towards the satellite sensor, compared to what is theoretically assumed. Reciprocally, an underestimation of g (*i.e.* assumed g=0.7 while true g=0.8) leads to a decreased retrieved  $\zeta$  value (*i.e.* negative bias) as a result of less photons scattered towards the satellite sensor, and more towards the surface compared to the assumption."

-p.13,I.3-17: I find it confusing that the way of analysing and presenting results is different when discussing the sensitivity to the error in the surface albedo. I would have preferred a plot in Figure 5 similar to Figures 4 and 6. Also, I don't understand the advantage of a box-whisker plot here: a single box-whisker is based on only three data points, right (three angles)? In addition, I would definitely like to know the direction of effects, particularly the effect of over- and underestimations of the surface albedo on ALP. If you want to retain the box-whisker plot (not preferred by me), then at least mention direction of effects in the text.

We removed the box-plots and reproduced impacts of surface albedo on retrieved AOT and ALP similarly to the previous ones (see new Figures 4 and 8).

However, as explained now in our upgraded manuscript, we kept the approach of box-whisker plots for evaluating the impacts of  $O_2$ - $O_2$  SCD uncertainties. Indeed, these uncertainties are precision problem (*i.e.* random error by opposite to systematic error or bias). They are mostly (but not only) related to the affects of instrument noise when fitting the  $O_2$ - $O_2$  SCD. Therefore, they will lead to a random error on a the retrieval of ALP, and thus affect the ALP precision. We explained this accordingly in our text.

We added these elements in our upgraded manuscript:

**Sect. 5.1:**

"Errors in surface albedo also lead to biases in retrieved  $\tau$  (*cf.* Fig. 4c). Overall, biases are larger over scenes with small  $\tau$  values. The reason is the dominance of surface reflection in this regime. In cases of high amount of aerosols, then aerosol scattering signals are dominant and surface reflection uncertainties have less impacts. An underestimated (overestimated) surface albedo results in a negative (positive) retrieved  $\tau$  bias. This is directly related to the change in the measured OMI Rc (*cf.* Fig. 1). Surface albedo uncertainties in the range of 0.025-0.05 lead to biases, in absolute, close to 0.5 for  $\tau$  in the range of 0.0-0.5, smaller than 0.25 for  $\tau$  = 2.0. Typical differences in climatological surface albedo from the total ozone monitoring spectrometer (TOMS) and the global ozone monitoring experiement (GOME) (Koelemeijer *et al.*, 2003), or between the Lambertian equivalent reflectance (LER) of OMI and the MODIS black sky albedo (Kleipool *et al.*, 2008), are known to be up to 0.02."

**Sect.5.2:**

"Surface albedo contributes to the length of the average light path and thus affects NsO $_2$   $-O_2$ . Retrieved ALP biases are maximum (several hundreds hPa) for  $\zeta \leq 0.5$  (cf. Fig. 8a and Fig.8b). For  $\zeta$  in the range of 0.5-1.0, retrieved ALP have lower absolute values (between 100 hPa and 200 hPa on average) with NN $_\zeta$  NS, while they remain too high with NN $_\zeta$ NS. Over scenes with  $\zeta \geq 1.0$ , biases are reduced to 0-50 hPa since aerosol scattering signals dominate over surface reflection. The main cause of all these improvements is that using an accurate prior  $\zeta$  information (or at least more accurate than retrieved OMI  $\zeta$  from Rc) allows a better distinction of  $\zeta$  and ALP effects on the  $O_2-O_2$  slant column density.

An accuracy better than 0.2 must be required on prior  $\zeta$  information (*cf.* Fig. 8c). Indeed, a  $\zeta$  bias of 0.25 can impact, in absolute, the retrieved ALP up to 50 hPa for  $\tau$  in the range of 0.5-1.0. For  $\tau \geq 1.0$ , ompact on ALP almost becomes null. Therefore, using MODIS  $\tau$  as prior to NN\_  $\zeta$  \_Ns is likely expected to show retrieved ALP with a higher quality O2-O2 than with NN\_Rc\_Ns. Indeed, the current retrieved OMI  $\zeta$  from Rc are below the MODIS  $\tau$  accuracy."

Figure 4: Simulated  $\tau$  (550 nm) retrievals, based on noise-free synthetic spectra with aerosols, as a function of true  $\tau$ . The assumed geophysical conditions are: temperature, H2O, O3, and NO2 from climatology mid latitude summer,  $\theta 0 = 25^{\circ}$  and  $\theta = 25^{\circ}$ , P s = 1010 hPa. All particles are located between 800 and 900 hPa and  $\alpha = 1.5$ . Note that the scenarios with lines and similar symbols general tend to fall on top of each other. The reference aerosol scenario is plotted with continuous lines and circle symbols and includes consistent aerosol optical properties with the supervision dataset used to train the neural network algorithm: *i.e.*  $\omega 0 = 0.95$ , g = 0.7. All the retrievals are achieved with the NN algorithm trained with  $\omega 0 = 0.95$ : (a) Sensitivity of  $\tau$  (550 nm) retrievals to the aerosol single scattering albedo ( $\omega 0 = 0.95$ , 0.9 or 1.0), (b) Sensitivity of  $\tau$  (550 nm) retrievals to the aerosol asymmetry parameter (g = 0.6, 0.7 or 0.8), (c) Sensitivity of  $\tau$  (550 nm) retrievals to a surface albedo bias ( $\partial$ Alb = 0.0, -0.025, 0.05) with  $\omega 0 = 0.95$ , g = 0.7.

Figure 8: Simulated ALP retrievals, based on noise-free synthetic spectra with aerosols, as a function of true  $\tau$ . The retrievals are achieved with the NN configurations (N NRc,N s and / or N NT,N s ) (see Sect. 3.1) trained with  $\omega 0 = 0.95$ . The assumed geophysical conditions are: temperature, H2 O, O3 , and NO2 from climatology mid latitude summer,  $\theta 0 = 25^{\circ}$  and  $\theta = 25^{\circ}$ , Ps = 1010 hPa. The reference aerosol scenario assumes fine scattering particles ( $\alpha = 1.5$ ,  $\omega 0 = 0.95$ ,  $\omega = 0.7$ ) located between 800 and 900 hPa: (a) and (b) Sensitivity of ALP retrievals to a surface albedo bias ( $\partial Alb = 0.0$ , -0.025, 0.05) with N NRc ,N s and N NT,N s , (c) Sensitivity of ALP retrievals to a  $\tau$  bias ( $\partial \tau = 0.0$ , -0.025, 0.05) with NNT,NsO2 -0.025.

**-p.13,l.19: You haven't discussed figures 5b and 5c yet!**

Please note that, according to the previous remark, we changed these figures 5a, b and c.

**-p.13,l.20-21: For scenes ... 100 hPa. -> But in this AOT range I see biases up to 300 hPa?**

Here we are only referring to cases where no error on aerosol properties and surface albedo are included (cf. continuous lines with circles in the plots which are associated with reference cases). The biases up to 300 hPa in this  $\zeta$  range appear when including errors in the assumed aerosol SSA. We move this note from the end to the beginning statement.

**-p.13,l.25: entire learning database -> you mean: the independent data set?**

No. We really mean here the entire learning database (*i.e.* training-verification-evaluation, *cf.* section 3.3) which is used to train the NNs, avoid overfitting and identify the optimal NN architecture. We added the clarification.

-p.13,fig.6: Why are there large biases when there are no forward model errors (no error in SSA or g)? These are closed-loop, noise-free simulations so the truth should be retrieved?

We think that being in a "closed-loop" condition is not enough in this case. Approaches like machine Learning (e.g. neural Networks) or interpolation within a Look-Up-Table (LUT) can be seen as a retrieval method based on an off-line forward (radiative transfer) model. This means that the explicit use of a full physics radiative transfer model for each single synthetic spectrum is here replaced by an approximate model, which tries to predict the behaviour of the real model for a given set of input parameters. The retrieval approach relies then more on a regularization process which resolves the retrieval problem from a statistic point of view (i.e. the most likely output associated with the set of all combined inputs), with an approximate (or more simple) model without iteration.

Bayesian retrieval approaches, such as the Optimal Estimation Method (OEM), might be able to retrieve the truth on academic cases (noise-free spectra and very low  $\zeta$ ) as 1) they use an exact and full physics forward model directly on-line for each synthetic case study, 2) prior knowledge and parameters are weighted with their corresponding uncertainties, 3) the non-linear problem is linearized around the prior information, 4) iteration is generally performed until the residuals are low enough (below a given threshold). Contrary to approaches based on off-line model, such an on-line forward model approach tries to fit all the details / fine structure present in the combination of input set of parameters (by opposite to regularization techniques).

However, OEM approaches may encounter other problems: *e.g.* the iterative process (in cases of very low  $\zeta$ ) may not converge, and thus no solution may be delivered to the addressed retrieval problem. Such a so-called divergence problem is not accessible by regularization processes (*e.g.* NNs).

NNs can be seen, at a high level, as an optimal interpolator (compared to a more conventional interpolation within a LUT). The large (negative) biases in ALP retrievals over cases with low AOTs, can be interpreted as our inability to interpret very minor changes in  $O_2$ - $O_2$  SCD. This is very similar to the problems of OMI effective cloud pressure in cases of very low effective cloud fraction. As described in different studies, OMI effective cloud pressures, which are retrieved through a LUT, are generally largely (negatively) biased in these conditions (Acarreta *et al.*, 2004; Chimot *et al.*, 2016; Veefkind *et al.*, 2016). The negative sign of the retrieval biases in those cases may be related to the specific curvature of the solution space in those regimes.

-p.14,l.3: on -> in

Ok

-p.14,l.8-16: I don't really see a systematic improvement in ALP retrievals when using *a priori* AOT in case the aerosol model has a bias (for example, when q=0.8 yes but when q=0.6 no).

It is true that using an accurate prior  $\zeta$  value as input, there is no systematic improvement ALP retrievals. However, major improvements occur when there were already large biases (more than 50 hPa) when using the continuum reflectance (475 nm) Rc. When biases with the use of Rc are less major (*i.e.* smaller than 50 hPa), the improvements are not significant anymore.

However, because, the use of AOT already helps to significantly improve cases with large biases, we can conclude that it is better (and therefore recommended) to use AOT instead of Rc as input for ALP retrieval.

-p.14,l.18: eq. 11? Please check all figure ad equation numbers, there are more wrong references.

We carefully and corrected, where necessary, all the equation and section numbers.

-p.14,l.22-26: This is an example of repetition; I would leave sentences out.

These sentences summarize and give all the major arguments justifying the use of a prior accurate  $\zeta$  as input for a correct ALP retrieval. Because these information are not exactly given, or only partially, in the previous analyses, we would prefer to leave these final and important conclusion at this place.

-p.14,l.26-28: Where do these SC precision values come from? Give ref. Are these values typical for OMI instrument noise? But then for each scenario there is only one typical SC error as it only depends on the radiance and no range should be considered, right? How can a random error in SCs lead to ALP biases? Why don't you take into account the noise error in your other input parameter? Your remark on the temperature correction seems out of place.

The  $O_2$ - $O_2$  SCD precision numbers that we considered come from the most typical values encountered in the OMI NASA O2-O2 product (we made a basic histogram analyse to identify their most likely ranges).

The  $O_2$ - $O_2$  SCD precision is typically related to the impact of the OMI noise instrument on the  $O_2$ - $O_2$  fit. Of course, it also includes other uncertainties, such as those related to the employed DOAS approach in the  $O_2$ - $O_2$  absorption band. We replaced "ALP biases" by "ALP uncertainties". We do not see the reason to take into account a noise error in the other input parameter. The impact of surface albedo and  $\zeta$  errors are already analysed previously. For these parameters, we assumed them more as geophysical parameter errors which may have specific spatial and/or temporal patterns (contrary to a noise which, by definition, presents a random pattern). Therefore, these errors would lead directly to a bias on the ALP retrieval instead of a precision impact. An illustration is given in Section 6 with the OMI surface albedo which include some aerosol residuals. These residuals lead to some specific patterns in the ALP retrievals as observed in Fig. 11.

We removed the remark about the temperature correction, since this is mentioned later on, in the Section 5.

-p.15,l.4: Can you give some more details here about the area that you selected (latitude and longitude? How many pixels passed selection? How many ALP data points went into the seasonal averages.

The considered area is defined by the range of latitude 25-40 deg North, and longitude 110-130 deg East excluding the part over the Gobi desert which presents a too bright surface (as explained in Sect. 4.2 and 6.1).

As already mentioned in Sect. 6.3 (former Section 5.3), when comparing OMI ALH results with LIVAS, about 17% of the pixels passed the selection in summer and spring, and between 5% and 6% in winter and autumn. This leads to about 77 000 OMI observations in spring and summer, and about 20 000 in winter and spring.

```
-p.15,l.19: Applying ... scenes. -> repetition
```

Ok, this is removed.

-p.15,l.21-22: However... as well. -> then why do you apply the OMI cloud fraction threshold...?

By applying simultaneously OMI and MODIS cloud fraction threshold, we maximize the probability of filtering out OMI cloudy pixels. According to past studies [Boersma *et al.*, 2011; Castellanos *et al.*, 2015; Chimot *et al.*, 2016], we know that OMI observation pixels with an effective cloud fraction larger than 0.1 are more likely to contain clouds than aerosols. If, the MODIS cloud fraction is lower than 0.1, but OMI effective cloud fraction is larger than 0.1, we preferred to reject this pixel. Such cases might happen since OMI and MODIS pixels do not always strictly overlap, and thus apparent "inconsistencies" between OMI and MODIS cloud detection might be observed.

-p.15,l.25-: I find your discussion of the temperature correction and later sensitivity tests completely out-of-place (particularly as you show results of it in the final most important figure). Also, I don't understand how you do the temperature correction. Is Ns\_O4-meas the slant column you get from the OMI cloud product? How do you use NCEP temperature profiles to calculate gamma? This factor clearly depends on the aerosol conditions, which you are trying to retrieve? I guess the physics of this temperature correction is well understood, apparently not part of the OMI cloud product and Veefkind *et al.* 2016 describe that users should apply it. So applying the temperature correction is always better and doesn't present a source of uncertainty for your ALH retrieval.

We shortened our discussion on the temperature correction and mostly included it when analysing the spatially and seasonal ALH average retrievals. Ideally, for each single  $O_2$ - $O_2$  SCD measurements, a NCEP temperature profile (the spatially closest one) has to be used to compute the so-called gamma factro and then apply it for the temperature correction. That being said, theoretically, by definition of the gamma factor, this correction would depend on aerosol properties (typically  $\zeta$  and ALP).

However, we found out that in cases of aerosols (contrary to effective cloud retrievals as performed in the OMI  $O_2$ - $O_2$  cloud algorithm), the gamma values does not vary (or very barely) with respect to  $\zeta$  and ALP variability, at least in the  $\zeta$  range of 0.0-2.0. It is mostly changing with NCEP temperature profiles. The reasons are double:1) the magnitude of aerosol optical thickness (between 0 and 2) is way lower than typical cloud optical thickness (at least 1 order less), 2) the range of 0-2 of aerosol optical thickness somehow depicts low changes (and then more higher stability) than changes in cloud optical thickness. Therefore, for convenience and low computing time, we considered for all the OMI observations similar aerosol properties (typical optical thickness of 1 and ALP = 800 hPa).

Also, as indicated in our written text, for practical reasons (implementation and computing time), we considered a spatial-seasonal average NCEP temperature profile which is then applied for computing the gamma factor for each single ALP / ALH retrieval. This may have some impacts (order of about 50 m) on single retrieval result. But the impacts on the average ALH results are expected to be minimal.

-p.15,l.8: geophysical location -> grid box?

We assumed that you meant p.16 here. This is changed.

-p.15,l.15: Note that ... -> So what does this imply?

This is to justify why we considered here the MODIS Black Sky albedo, and not the White Sky albedo.

-p.15-16,sect.5.2: I stopped reading this section in detail. A few general remarks: There are several forward references to figures hat have not yet been explained. Some figure numbers are incorrect. When investigating the effect of the surface albedo input on the retrieved AOT, I don't see the reason why you

should split the analysis per season (perhaps personal taste). Are MODIS AOT and MODIS BSA truly independent?

We reformulated where necessary this section to make it clear. Please read the new Sect.~6.2. We only illustrate the impact of surface albedo for the autumn season. The impacts are somehow similar for all the other seasons.

As explained at the end of our Sect.6.2., OMLER and MODIS BSA can be expected to present some differences and thus independences: OMLER likely includes some aerosol and cloud residuals, MODIS BSA is the integration over the full hemisphere while OMLER includes most (and only) the viewing angles encountered by OMI, and the assumed solar reference zenith angle in MODIS BSA is likely inconsistent with the true OMI local observation time.

**-p.17,l.27: polarization -> Your RT calculations for the LUT excluded polarization effects?**

Yes, indeed it did not include polarization since the OMI measurements are depolarized (because of the use of a scrambler). However, we wanted to acknowledge that not using polarized simulations, in the learning database, may lead to some errors (likely of the order of 2% in the Visible according to reference studies). This should be investigated in a future work, since this is out the scope of the present study.

**-p.17,l.31-32: MODIS tau ... OMI tau. -> Can you provide refs?**

Yes, we added the following references about the overall quality of MODIS  $\zeta$  in the new Section 2.2 as follow:

"The expected error of MODIS DT  $\zeta$  is about  $\pm 0.05 + 15$  % over land (Levy *et al.*, 2013). The "Deep Blue" retrieval algorithm has been developed to complement the DT algorithm by retrieving  $\zeta$  over bright arid land surfaces (*e.g.* deserts). The typical associated uncertainties are about  $\pm 0.03$  on average (Sayer *et al.*, 2013)."

-p.18,l.4-9: You discuss the variability in your set of ALH pixels, but what does this tell you? Isn't it obvious that the variability decreases when you use the setup with fixed AOT: there is one free parameter less? This doesn't show that those retrievals are more accurate. Isn't it obvious that the variability decreases with higher AOT as the aerosol signal is more stronger...? But there is an interfering effect of the geophysical variability! Perhaps low-AOT aerosols are typically closer to the ground (BL pollution) while high- AOT aerosols are extreme events that reach into the free troposphere? Don't know.

I am not sure that, by just looking at this figure alone, we can say this is obvious that using MODIS  $\zeta$  as input should decrease the ALP variability. Or at least, this only becomes obvious thanks to the analyses achieved on synthetic cases as described and summarised in Sect. 5.2. Only thanks to these analyses, we know that retrievals at low  $\zeta$  presents some instabilities due to a very low  $O_2$ - $O_2$  absorption signal.

To have an idea about the true geophysical variability included in LIVAS, we added the thick black line: it shows the expected variability of ALH as seen by LIVAS over the same area. This variability seem to be independent of seasons and is about 700 m. ideally, we should expect then our ALH retrievals to show varibility as close as possible to this black line. We discussed it in our updated manuscript and as response to one of your main remarks above.

The methodology of deriving the LIVAS variability is given below (next remark / answer).

-p.18,l.20: Spatial average ... performed. -> How do you compute ALH from extinction profiles? How do you compute the spatial average? Do you take into account the sampling of your OMI pixels? I mean, if for a given location you have, say, twice as many OMI pixels in summer than in winter, do you take this into account when calculating the LIVAS average. Please at least explain exactly what you did.

The LIVAS ALH is derived from the given averaged vertical profile of aerosol extinction (532 nm)  $\sigma(I)$  over each vertical layer I defined by its altitude h(I) as follow:

$$ALH(LIVAS) = \frac{\sum\limits_{l} h(l)\sigma(l)}{\sum\limits_{l} \sigma(l)}$$

Since LIVAS also provides the standard deviation associated with each averaged vertical profile of aerosol extinction (532 nm)  $\partial \sigma(I)$ , the equivalent standard deviation of each LIVAS ALH  $\partial$ ALH is derived as follow:

$$\partial ALH(LIVAS) = \frac{\sum\limits_{l} \partial h(l) \partial \sigma(l)}{\sum\limits_{l} \sigma(l)}$$

where  $\partial z(I)$  is the geometric thickness of each vertical layer I.

We added these equations in our new Sect.2.3.

We did not take into account the potential discrepancy between the number of OMI pixels and CALIPSO pixels included in LIVAS, and therefore their specific sampling. Of course, satellite observation resolution and samplings should have an impact. However, fully addressing in this unique study the problem of representation error between OMI and LIVAS (which is derived from CALIPSO observations) is a real work in itself which is currently out the scope of this project.

-p.20,l.14-15: You state that the NN with AOT as a fixed input performs better. But comparing figure 16b and 16d, this holds for the red and purple lines (SSA 0.95), but the opposite is true for the green line (SSA 0.9) and the blue line is undecided. Yet, previously you argued based on the comparison of AOTS that an SSA of 0.90 is probably the more accurate value in three seasons. So I think this conclusion cannot yet be drawn.

I don't think that, in general we have the right to focus on 1 single colour line precisely is appropriate. What really matters here is the spread of the overall / global retrievals due to assumptions on aerosol optical properties (in particular single scattering albedo) which are the most crucial assumptions impacting the spread of our results. The chosen surface albedo being of lower order of magnitude.

Indeed, we stated that, on average, a  $\omega 0$  of 0.9 (in particular in spring, autumn and winter) is likely a more reasonable value than 0.95 (upper limit in this region). Since, there is not accurate aerosol  $\omega 0$  value available for each individual OMI observation pixel, we do not have the choice but to use a constant  $\omega 0$  value to all the OMI observations. However, using a constant  $\omega 0$  value leads to substantial errors in the ALH results (biases and precisions). Therefore, even if the green line looks to overlap with the LIVAS

line when using NN\_Rc\_Ns, I don't think we can directly conclude that the retrievals are here very good in this framework.

What mainly matters here, because again of how assumed SSA values are employed here, is the spread between the lines associated with  $\omega 0$  0.9 and 0.95 when using NN\_Rc\_Ns and NN\_  $\zeta$  \_Ns algorithms. This is because this spread significantly reduces from NN\_Rc\_Ns to NN\_ $\zeta$ \_Ns, that I think we can conclude that the NN\_ $\zeta$ \_Ns fits ALP / ALH better. But this algorithm also fits better because we use an accurate  $\zeta$  prior information, provided by MODIS. If we are able to retrieve very accurate  $\zeta$  from every OMI single pixels, the NN\_Rc\_Ns algorithm would probably correctly fit ALP / ALH as well.

-p.29,fig.1: What is azimuth difference (throughout)? Does H2O absorb in your fit window? Then why isn't it included in the DOAS fit? At what wvl should I interpret TAU?

The relative azimuth difference is the difference between the viewing azimuth (of the OMI satellite) and the solar azimuth angle. This is already defined in our paper in Sect. ~3.2.

It is known that the  $H_2O$  absorption is very low in this spectral window. Since,  $O_3$  and  $O_2$ - $O_2$  are the dominant absorption gases here, we did not consider the fit of  $H_2O$  SCD. Note that, similarly, this gas is not considered in the OMI cloud algorithm.

As indicated in the text, and in most of the captions,  $\zeta$  (*i.e.* AOT) should always be interpreted at the reference wavelength 550 nm. We verified that this information is always clarified though the entire manuscript.

-p.30,table2: I don't understand this figure. For example, the ALP test error for a given NN configuration is the sum-of-squared differences (eq. 7) for all the scenarios in the test set (the 15%). This is one number. But then you repeat the training three times. So you have three numbers. Did you calculate boxwhiskers from only three points? That doesn't seem right.

-p.33,fig.4: In left panel, can you add also SSA = 1.0 simulation? You have tested positive and negative errors in the asymmetry parameter as well? Can you make axes in the right panel such that the 1-1 line corresponds to the diagonal?

The case with  $\omega 0 = 1.0$  is now added.

By testing cases with g=0.6 and g=0.7, while the NNs are trained over a database with g=0.7, we already indeed considered positive and negative errors in the asymmetry parameter. We found out that making the axes such that the 1-1 line would correspond to the diagonal does not lead to nice plots. Indeed, depending on the magnitude of the retrieved  $\zeta$  biases, we may have parts of the plots empty / white on the right or left part of the figures which would give strange and diverging feeling to the reader.

Instead, we added the remark, in all the captions, that continuous lines with the circle symbols should be considered as the reference. Indeed, the aerosol properties are consistent between the tested cases and the NNs trainings. And they do not include any bias on the input surface albedo. Therefore, any gap between these lines and the others, refer to a (negative or positive) bias in the plotted retrievals.

-p.39,fig.10: I see many more lines in the plot than in the legend? Can you make the plot line colors the same for the same surface albedos in the top and bottom row? I don't think the differences between the OMLER and MODIS BSA are significant, do you?

We remove the two plots at the top. Showing the impact of aerosol residuals in the OMI surface albedo for only one season (*e.g.* Autumn) is enough. Similar effects are found for the other seasons.

-p.40-41,fig.12-13: It took me some time to realise that -for unclear reasons- these figures are split whereas they should be merged? The color map labels are not visible? The tau symbol is not a tau symbol?

There are 2 basic reasons why to split the figures as you mentioned:

- putting all the 4 seasons (x 4 panels) on the same page via LaTex leads to a reduction of their global size, and thus make them less visible for the reader.
- Also, since summer seem to depict better retrieved AOT results assuming SSA = 0.95, it may make sense to separate it from the other seasons where assuming SSA = 0.9 seems to be more accurate.

**Interactive comment on «An exploratory study on the aerosol height retrieval from OMI measurements of the 477nm $O_2$ – $O_2$ spectral band, using a Neural Network approach » J. Chimot *et al.**

Julien Chimot et al.

**J.J.Chimot@tudelft.nl**

We thank Referee #2 for his / her valuable comments. Below we address them one by one (Referee #2 comments in blue, author and co-authors in black).

The paper examines retrievals of aerosol optical thickness and layer pressure (height) from OMI spectral measurements (or as applicable to other instruments) of the  $O_2$ - $O_2$  absorption band near 477 nm using a neural network approach. A detailed analysis is carried out using simulated data. The approach is then applied in different ways using OMI and MODIS data over land areas of Asia with relatively high aerosol loading and compared with a lidar-based data set (LIVAS).

This is a detailed paper that should be published in AMT. The paper is in general clearly written though there are a number of typos and grammatical issues that it is assumed will be caught during the copyediting of the manuscript. Only a very few are listed below. I agree with the comments of reviewer 1 and add some additional comments for minor revisions below (some of these may be duplicates).

We thank you for your recommendations and encouragements. We took into account and answer the comments and questions of Referee #1. You can see them in our corresponding document on the AMT website, below the link to the reviews of Referee #1.

We carefully reviewed the new version of our manuscript and verified & corrected the typos and grammatical issues where necessary.

The last sentence of the abstract - "This study shows the first encouraging aerosol layer height retrieval results over land from satellite observations of the 477 nm  $O_2$ - $O_2$  spectral band." - is correct as written. However, the authors may mention here that a previous study examined case study retrievals over ocean. This sentence may stick in the reader's head as this is a "first" implementation with real data (references are later given and it becomes more obvious that these are the first results shown with data over land). I had to go back and reread the sentence to find the over land part, which makes it correct.

Following the mention of the reference of Park *et al.*, (2016) in the other parts of our manuscript, we added the following statement in the abstract: "Following the previous work of Park *et al.*, (2016) over ocean, our study shows the first ...."

It might be better to include up front a Data section with the various satellite data sets used (OMI, MODIS, LIVAS) rather than to mention them in different places (and not referenced at the first mention - LIVAS). As it is OMI is mentioned in its own section section with MODIS mentioned below in the Methodology section. It seems that MODIS is an important part of this study as it is important to get accurate ALH and perhaps it deserves more attention.

Thanks for your suggestion. We created a specific Section 2 where are gathered and described all the satellite dataset: aerosol measurements and products (OMI, MODIS AOT Collection 6, and LIVAS climatology), and the surface albedo databases (OMLER and MODIS Black Sky Albedo). These descriptions are mostly all the parts that were written in our former manuscript version (ex-Section 6.1, now renamed as section 7.1). This section is now reduced, mostly focusing on the process of collocation of OMI-MODIS, cloud filtering and the applied temperature correction on the  $O_2$ - $O_2$  slant column density (SCD).

P. 2, L. 34, here and also elsewhere suggest to add *e.g.* before references as there are others not in this list

Ok, this is done on P.2, L.34 and other pages.

P. 3., L. 20, please add Torres *et al.*, 1998 before de Graaf reference, also suggest to add Torres *et al.*, 2013 reference here and explain that monthly climatology of CALIOP aerosol heights are currently in use for determination of aerosol parameters from OMI UV measurements.

We included the reference Torres *et al.*, (1998) prior to de Graaf *et al.*, (2005). In the introduction, we added a mention about the study of Torres *et al.*, (2013) that has integrated monthly climatology CALIPSO aerosol heights to update the OMAERUV algorithm

P.4, L. 2, please add appropriate references here (altogether, though they are listed above). Section 2.3, 1st par., This information may go better in the introduction. It's not clear why MAX-DOAS is mentioned specifically here (this sentence seems out of place and not necessary). May be useful also to mention the work of using  $O_2$  A-band to retrieve aerosol height (over ocean by *e.g.*, Dubuisson et al. 2009) and discuss possible advantages of the  $O_2$ - $O_2$  band (lower surface albedos over land?) and also discuss availability of these bands on various sensors such as OMI, OMPS, GOME-2, TROPOMI.

The statement on P.4 L.2 states that no aerosol height retrieval exists (at our knowledge) from the OMI  $O_2$ - $O_2$  visible absorption band over land. Because of the nature of this statement, we do not have specific references here. Regarding the previous statements, mentioning the past works analyzing the nature of this spectral band, the rereferences are already mentioned (*e.g.* Veihelmann *et al.*, (2007), Dirksen *et al.*, (2009), Park *et al.*, (2016)).

We removed our discussion about MAX-DOAS. It was just an information about existing  $O_2$ - $O_2$  ground-based measurements. But it is true that we do not use them then further in the manuscript.

P. 9, L. 6, I think "either" should be removed as it is confusing. Sect. 5.1, Perhaps I missed it but I do not see where the area of North-East Asia is defined. Please give the latitude-longitude of the area studied and/or show it on a map. I believe the highly industrialized areas (where there is heavy aerosol loading) used in this study may also be referred to as South-East Asia, thus it can be confusing.

We removed the word "either". We added the description of the area as follow:

"The considered area is defined by the range of latitude 25-40 deg North, and longitude 110-130 deg East excluding the part over the Gobi desert which presents a too bright surface (as explained in Sect. 4.2 and 6.1)."

P. 15, The discussion of the pairing of OMI and MODIS is confusing. It should be made clear that the resolution of MODIS is 1 km or better, but that collection 6 aerosol products are available at either 3 or 10 km resolution and that you are using 10 km.

We clarified in the new Section 2.2 as follow: "The MODIS instrument, launched on the NASA EOS-Aqua platform in May 2002, is a spectrometer delivering continuous images of the Earth in the visible, solar infrared and thermal infrared approximately 15 min prior to OMI on-board EOS-Aura. The considered MODIS Aqua Level 2 (L2) aerosol product is the collection 6 of M Y D04L 2, based on the Dark Target (DT)

Land algorithm with a high enough quality flag (Xiao  $et\ al.$ , 2016). While the MODIS measurement is available at the resolution of 1 km, the MODIS aerosol product is available at both 3 km x 3 km and 10 km x 10 km. Since this last one is relatively close to the OMI nadir spatial resolution, it is then used ine the work below (cf. Sect. 7). The improved calibration of MODIS Aqua instrument is included in the reprocessing of the collection 6 aerosol product (Levy  $et\ al.$ , 2013; Lyapustin  $et\ al.$ , 2014)."

Please mention that you are using dark target only if this is the case and mention the specific product name, *e.g.*, MYD04\_L2. I believe that data are provided within the 10 km grids if they have some amount

of cloud free pixels, so there may still be clouds present within the given area even when MODIS data are reported. Also, the 10 km x 10 km areas should not be referred to as a pixel as this can be confused with native MODIS pixels. Does the MODIS geometric cloud fraction used come from the aerosol product? This should be clarified as there are multiple MODIS cloud detection algorithms.

We added the mention of the Dark Target algorithm where the MODIS aerosol product comes from over land. We clarified in the text that the MODIS measurement is acquired over a pixel observation of 1 km, but the grid cell of the used MODIS Level 2 (L2) aerosol product has a resolution of 10 km.

**P. 15, L 24, change of to or.**

Please check the English meaning of referent. It is defined in most English dictionaries as a noun but used in the paper primarily as an adjective. I believe the word reference may serve better in most instances and also perhaps the word default.

Change is done.

The word "reference" is more appropriate indeed. We modified where necessary in the text.

**P. 17, L. 19, accuracy of OMI tau retrievals with respect to MODIS.**

We added in Sect.2.2 the following:

"The expected error of MODIS DT  $\tau$  is about  $\pm 0.05 + 15$  % over land (Levy *et al.*, 2013). The "Deep Blue" retrieval algorithm has been developed to complement the DT algorithm by retrieving  $\tau$  over bright arid land surfaces (*e.g.* deserts). The typical associated uncertainties are about  $\pm 0.03$  on average (Sayer *et al.*, 2013)."

P. 18, L. 29, remarkable agreement with respect to seasonal mean values Table 1: What happens for surface pressures < 963 hPa. The neural net will be extrapolating. How well will it do this? Does this occur with the real data? Likewise, why not add a node with single scattering albedo of unity? Also, the maximum value of VZA for OMI is > 44.2 degrees, so why not include the full range?

Somehow, when the input parameters are outside of the range of the training ensemble dataset, the NN will by default extrapolation. The activation function associated with the neurons in the hidden layer is a sigmoid function. Because of the shape of this function (by opposite to a Dirac for instance), the extrapolation is possible over a limited range of the dynamic of each parameter. Therefore, we can expect the NN to correctly extrapolate. This is confirmed by the ensemble results and our statistical analyses in this manuscript. No results show strange extrapolation features or saturated values because of extrapolation beyond the shape of the sigmoid function.

But, further analyses should be done over single and specific cases to ensure the quality and characterize the degree of this extrapolation.

Because of the specified geometric thickness of our aerosol layer, and its nature, we could not define a proper simulation in the training dataset with an ALP (Aerosol Layer Pressure) below 963 hPa. The radiative transfer simulations would have been unrealistic and would have greatly affected the NN training performances.

The reason not include OMI VZA > 44.2 degrees is due to the high time consuming to increase the number of simulations in the supervision dataset. We had already 400 000 simulations which we think is already a large enough number to ensure the robustness of the training step. Furthermore, most of the OMI observation pixels have statistically a VZA lower than 44 deg. But we do recognize that we probably selected pixels with larger angles...

Table 2: When providing values for delta  $NsO_2-O_2$  it would be good to provide a percentage error for these for a given scenario (readers will not have a good idea as to how large these values are). Perhaps I missed it but doing a search of "table" doesn't turn up a reference to Table 2 from the text. The aerosol optical thickness error is quite large for a change in surface albedo of 0.05. If surface albedo errors are more of the order of 0.02 (as stated in the text) then perhaps this would be a more appropriate value to use.

The percentages may vary, but as a first order numbers in the range of 2-7% are probably a good approximation. We added them.

The impact of surface albedo uncertainties on retrieved AOT over synthetic cases is updated in Figure 4, following the comments of Referee #1. The retrieved AOT biases are shown per AOT and for 2 typical surface albedo uncertainties, with opposite signs: a high uncertainty of 0.05, and a lower of 0.025. The second one is then more in line with a more appropriate value as discussed in our manuscript.

All figures in general would benefit from larger fonts. Also the tau in the figures looks different enough from the tau in the text to be somewhat confusing. Figure 3 caption is confusing. What exactly is the supervised data set (training-validation-test)?

We have enlarged fonts where captions, legends, ax labels where not readable enough.

Figure 4, again lines and symbols hard to distinguish. It would be helpful to mention in the caption that the scenarios for the lines with the dot symbols tend to tall on top of one another. Ok, we added this remark in the caption of Figure 4.

Figure 6: Something should be mentioned in the caption about the range of surface albedos (Alb) used (same for several other figures).

The range of surface albedos is already mentioned in the captions. The Albedo parameter is defined by the notation A. We verified that this notation is already properly introduced throughout the entire manuscript.

Figure 9: I don't see where it is stated that the dotted line is the 1:1 line.

Thanks! We added this remark in the caption of Figure 9.

Fig. 10: There are many lines on these plots. It would help the reader if the 1:1 lines were made thicker to distinguish them. There are backward brackets in the legends at the ends of lines.

Following your suggestion and the one from referee #1, we changed this figure by leaving only 1 season (autumn), so then the reader has only a limited number of lines to visualise.

**An exploratory study on the aerosol height retrieval from OMI measurements of the 477 $\rm nm~O_2-O_2$ spectral band, using a Neural Network approach**

Julien Chimot1, J. Pepijn Veefkind1,2, Tim Vlemmix1, Johan F. de Haan2, Vassilis Amiridis3, Emmanouil Proestakis3,4, Eleni Marinou3,5, and Pieternel Felicitas Levelt1,2

[revised manuscript text omitted]
(475\text{nm})$  and  $N_{\text{O}_2-\text{O}_2}^{\text{s}}$  from the OMI  $\text{O}_2-\text{O}_2$ 477 nm spectral band The absorption band. The various DOAS techniques rely on the same key concept: a simultaneous extraction-fit of several trace gas slant column densities from the fine spectral features due to their absorption (i.e. the high frequency part) present in passive UV-Vis spectral measurements of atmospheric radiation (Platt and Stutz, 2008). The OMI cloud algorithm exploits the 460-490spectral band, focused on the absorption line at 477. The initial purpose of this algorithm is the retrieval of cloud parameters, namely effective cloud fraction and effective cloud pressure (Acarreta et al., 2004; Veefkind et al., 2016). The first step of this algorithm is a DOAS spectral fit in which Here, the DOAS fit follows the same approach as in the OMI O2–O2 cloud algorithm (Acarreta et al., 2004; Veefkind et al., 2016): i.e. the absorption cross-section spectrum of O2–O2 is fitted together with a first order polynomial:

$$-\ln(R(\lambda)) = \gamma_1 + \gamma_2 \cdot \lambda + N_{\mathcal{O}_2 - \mathcal{O}_2}^{\mathbf{s}}(\lambda) \cdot \sigma_{\mathcal{O}_2 - \mathcal{O}_2} + N_{\mathcal{O}_2}^{\mathbf{s}}(\lambda) \cdot \sigma_{\mathcal{O}_3}, \tag{3}$$

where  $\gamma_1 + \gamma_2 \cdot \lambda$  defines the first order polynomial,  $\sigma_{\rm O_2-O_2}$  and  $\sigma_{\rm O_3}$  are the  ${\rm O_2-O_2}$  absorption cross-section spectrum (at 253) and the  ${\rm O_3}$  absorption cross section spectrum respectively, convoluted with the OMI slit function, and  $N_{\rm O_3}^{\rm s}$  is the  ${\rm O_3}$  slant column density and  $N_{\rm O_2-O_2}^{\rm s}$  is the slant column density.  $\sigma_{\rm 
[revised manuscript text omitted]
 part of the  ${\rm O_2-O_2}$  slant column density (SCD)  $N_{\rm O_2-O_2}^{\rm s}$  values. As observed in Fig. 1 and also reported by Wagner et al. (2004); Park et al. (2016); Chimot et al. (2016) complex located below the aerosol layers are shielded). Nevertheless, in case of low  $\tau$  values (i.e.  $\leq$  0.5),  $N_{\rm O_2-O_2}^{\rm s}$  generally increases with increasing aerosol layer pressure

(or decreasing aerosol layer height), does not significantly vary with respect to ALP. This shows that a low amount of aerosols has very little impacts on  $O_2$ – $O_2$  absorption measurements.

However, as depicted in Fig. 1e, 1b, 1d and 1f, not only ALP but also τ directly influences the slant O2-O2 absorption since both parameters simultaneously affect the photon path distribution average path followed by the photons, and therefore the overall shielding or enhancement effect. An increase of τ, for particles with ω0 ≤ 0.95, leads to a decrease of NO2-O2. The slope of this decrease depends on the aerosol altitude (i.e. higher for particles at high altitude). This means that τ also drives the shielding effect, while the aerosol height does not significantly affect Rc. As a consequence: the 2 parameters Rc and Note that both ω0 and A also affect NO2-O2 have a non negligible correlation (see, but this effect is smaller than τ. For example, a reduced ω0 and A lead to a small decrease of NO2-O2 (cf. Fig. 1b) as they share, in part, same information: the length of the average light path and the aerosol optical thickness τ. They are not two independent variables and combining them does not provide with two independent pieces of aerosol information. Id and Fig. 1f).

**As a consequence:**

15

20

25

30

- the variable single parameter  $N_{\rm O_2-O_2}^{\rm s}$  contain both information, contains information on both  $\tau$  and ALPthat cannot be easily separated. As a consequence. 
[revised manuscript text omitted]

- either  $R_c$   $R_c$  (475 nm) and  $N_{\text{O}_2-\text{O}_2}^{\text{s}}$  for  $\tau$  (550 nm) and ALP retrievals: this configuration is named  $NN_{R_c,N_{\text{O}_2-\text{O}_2}^{\text{s}}}$ ,
- or  $\tau$  (550nm) and  $N_{\rm O_2-O_2}^{\rm s}$  for ALP retrieval: this configuration is named  $NN_{\tau,N_{\rm O_2-O_2}}$ .

The output layer is, for each NN retrieval algorithm, composed of only one output variable: either  $\tau$ , or aerosol layer pressure ALP. In total, three NN retrieval algorithms are then selected and used at the end: one  $NN_{R_c,N_{O_2-O_2}^s}$  for  $\tau$  (550nm) retrieval, one  $NN_{R_c,N_{O_2-O_2}^s}$  and one  $NN_{\tau,N_{O_2-O_2}^s}$  for ALP retrieval.

The choice to use either  $NN_{R_c,N_{\rm O_2-O_2}^{\rm s}}$  or  $NN_{\tau,N_{\rm O_2-O_2}^{\rm s}}$  will impact the accuracy of the ALP retrieval results (see Sect. 4.2, 5.2 and 5.3).

**4.2 Generation of the supervision database: aerosol properties and simulations**

The MLP neural networks must be trained in order to build models that can learn accurately trained from a well-known data set, and then. They are then able to generalize the inverse problem by predicting the aerosol retrievals from input observations that have never been seen before. For that purpose, a learning database must be carefully designed and generated. It must be representative of the entire distribution of (input-output) values that can likely be encountered in the OMI observations. As a consequence for the MLP algorithms, a large quantity of data is often required for the learning process. However, very large learning data set can be extremely time consuming in terms of generation and then NN training.

The NNs are here trained with simulated data Training a neural network based on a large ensemble of synthetic dataset has been widely employed in atmospheric retrieval science such as for CO2 and CH4 (??), aerosol (?) and cloud properties

(???). This study uses created our own training dataset based on simulations from the Determining Instrument Specifications and Analyzing Methods for Atmospheric Retrieval (DISAMAR) software of KNMI (de Haan, 2011). DISAMAR includes a radiative transfer model and different retrieval methods. The radiative transfer model is based on the Doubling Adding KNMI (DAK) model (de Haan et al., 1987; Stammes, 2001) and thus computes the reflectance and transmittance in the atmosphere using the adding/doubling method. This method calculates the internal radiation field in the atmosphere at levels to be specified by the user and takes into account Rayleigh, aerosol and cloud scattering and trace gas and aerosol absorption. Scattering by aerosols is simulated with a Henyey–Greenstein scattering phase function  $\Phi(\Theta)$  (Hovenier and Hage, 1989):

10

5

$$\Phi(\Theta) = \frac{1 - g^2}{(1 + g^2 - 2g\cos\Theta)^{3/2}}.$$
(7)

where  $\Theta$  is the scattering angle. The phase function is then parameterized by the asymmetry parameter g, which is the average of the cosine of the scattering angle, such its variation  $-1 \le g \le 1$  ranges from back-scattering through isotropic scattering to forward scattering. Following the DISAMAR configuration,  $\tau$  values in the simulations are specified at the referent reference wavelength of 550 nm. The Ångström exponent  $\alpha$  describes the spectral dependence of  $\tau$ .

ALP is the main target parameter since this is one of the main parameters describing the average light path distribution in the tropospheric  $NO_2$  AMF computation. The second target is  $\tau$  since this information may be requested for a good ALP retrieval quality. We thus assume we do not need at this level to define more realistic aerosol models for every aerosol scene. With a referent-reference asymmetry parameter of g = 0.7, intermediate value typically observed (Dubovik et al., 2002), the Henyey-Greenstein function is known to be smooth and reasonably well reproduce the Mie scattering functions for most of aerosol types. This approach is also used for the preparation of the operational aerosol layer height retrieval algorithm from Sentinel-5 Precursor (Sanders et al., 2015) and for explicit aerosol corrections in the AMF calculation when retrieving trace gases such as tropospheric  $NO_2$  (Spada et al., 2006; Wagner et al., 2007; Castellanos et al., 2015).

The ensemble of parameters and associated values used for generating the learning database is detailed in Table 1. About 460 000 spectral simulations, over the  $O_2-O_2$  spectral band (460-490 nm), were generated, assuming different satellite viewing and solar geometries, surface albedo, surface pressure A, Ps and aerosol pollution levels. Scenes with too large angles (i.e.  $\theta_0 \ge 65^\circ$ ) and too bright surfaces (i.e. A > 0.1) are excluded. For each of these simulations,  $R_c$   $R_c$  (475nm) and  $N_{O_2-O_2}^s$  were deduced from the DOAS fit equations Eq. (23) and Eq. (34). Aerosols are specified for a standard case, assuming fine particles with a unique value of  $\Lambda$  negation coefficient  $\alpha = 1.5$  and g = 0.7. Aerosol profiles are parameterized by scattering layers with constant aerosol volume extinction coefficient and aerosol single scattering  $\omega_0$  and with a fixed pressure thickness. The ALP corresponds to the mid-pressure of the assumed scattering aerosol layer. In order to investigate the assumptions related to the single scattering albedo properties  $\omega_0$ , two typical values are considered:  $\omega_0 = 0.95$  and 0.9. Contrary to the other variables,  $\omega_0$  is not known for each OMI pixel and thus cannot be used as an explicit input parameter by the designed NNs. Moreover, it cannot be retrieved from this band since it is supposed to affect  $R_c$  (475nm) and  $N_{O_2-O_2}^s$ , similarly as  $\tau$ . Therefore, 2 sets of NN models are developed for different purposes:

- one set of three MLP NN algorithms  $(NN_{R_c,N_{{\rm O}_2-{\rm O}_2}}^{\rm s})$  for  $\tau$  retrieval,  $NN_{R_c,N_{{\rm O}_2-{\rm O}_2}}^{\rm s}$  and  $NN_{\tau,N_{{\rm O}_2-{\rm O}_2}}^{\rm s}$  for ALP retrieval) is trained assuming  $\omega_0 = 0.95$ ;

- one set of three MLP NN algorithms ( $NN_{R_c,N_{O_2-O_2}}^s$  for  $\tau$  retrieval,  $NN_{R_c,N_{O_2-O_2}}^s$  and  $NN_{\tau,N_{O_2-O_2}}^s$  for ALP retrieval) is trained assuming  $\omega_0 = 0.9$ .

[revised manuscript text omitted]
 250:350to 150:250250-350 hPa with  $NN_{R_0,N_{\Phi_2,\Phi_2}^8}$  to 150-250 hPa with  $NN_{\tau,N_{\Phi_2,\Phi_2}^8}$  and true  $\tau(550\text{nm})$  value. For particles higher than 650 hPa (or 3.5 km), no improvements are however observed. The low sensitivity to retrieve ALP when particles are

located at a very high altitude is directly due to the  $O_2-O_2$  complex and its vertical distribution. This was demonstrated by Park et al. (2016):  $O_2-O_2$  concentration exponentially decreases with increasing atmospheric altitude.

Secondly, impacts due to uncertainties related to on the chosen surface albedo and aerosol model are reduced. Assumptions on aerosol  $\omega_0$  drive the interpretation of the shielding of the  $O_2-O_2$  dimers by aerosols. If the assumed  $\omega_0$  is too high (i.e. assumed  $\omega_0=0.95$  while  $\omega_0$  true = 0.9), can perturb ALP retrievals obtained with the  $NN_{R_e,N_{O_2-O_2}}$  configuration are perturbed by more than 100 hPa (see Fig. 6a5a). These perturbations are reduced to the range of 0.100 hPa over scenes with high  $\tau$  values (larger than 1) only for particles close to the surface, i.e. true  $ALP \geq 850 \text{hPa}$  (see Fig. 7a 6a and Fig. 7e6c). Using the  $NN_{\tau,N_{O_2-O_2}}$  configuration, assuming true  $\tau$  as input, with true  $\tau$  (550nm) value helps to mitigate theses biases. The ALP retrievals are almost superposed to those with true  $\omega_0$  assumptions and All the ALP retrievals present the same behaviors with respect to the particles altitude and  $\tau$  and biases lie in the range of 0.50 hPa(see Fig. 6b5b). Similar conclusions are observed regarding uncertainties on the asymmetry parameter. Higher g (cf. Fig. 5c and Fig. 5d). Too high g values impact the ALP retrievals from  $NN_{R_e,N_{O_2-O_2}}$  over scenes with  $\tau \leq 1.0$ . Such a bias is largely reduced with the  $NN_{\tau,N_{O_2-O_2}}$  configuration.

Surface albedo variability contributes to contributes the length of the average light path and thus leads to changes in affects  $N_{O_2-O_2}^{N}$ . Retrieved ALP bias (Eq. 11) varies between 23 biases are maximum (several hundreds hPaand 141 on average, for surface albedo uncertainties  $\partial x_{atb}$  in the range of 0.05-0.05 with the  $NN_{\tau,N_{O_2-O_2}}$  configuration (see ) for  $\tau$  (550nm)  $\leq$  0.5 (cf. Fig. 6b). To limit the values below 100 8a and Fig. 8b). For  $\tau$  (550nm) in the range of 0.5-1.0, retrieved ALP are impacted by lower absolute values (between 50 hPa on average,  $\partial x_{atb}$  should not exceed 0.025. Using the  $NN_{\tau,N_{O_2-O_2}}$  configuration with

true  $\tau$  value, not only reduces the average but also the standard deviation of the bias: i.e. for  $\partial x_{alb} = 0.05$ , the aerosol pressure bias decreases from  $141 \pm 96$  and 100 hPa on average) with  $NN_{\tau,N_{02-02}^s}$ , while they remain too high with  $NN_{R_c,N_{02-02}^s}$  to  $69 \pm 79$ . Over scenes with  $\tau(550 \text{nm}) \ge 1.0$ , biases are reduced to 0-50 hPa with  $NN_{\tau,N_{02-02}^s}$ . The reasons since aerosol scattering signals dominate over surface reflection. The main cause of all these improvements are: 1) the strong correlation between  $R_c$  and  $N_{02-02}^s$  provides with a limited number of independent pieces of information to the retrieval system (see Sect. 2.3), 2) uncertainties about the surface albedo and aerosols  $\omega_0$  impact the interpretation of both of  $R_c$  and  $N_{02-02}^s$  and mislead then the interpolation process, 3) using is that using an accurate prior  $\tau$  as input is more consistent with the definition of aerosol models in the learning database and allows a better distinction the effects of  $\tau$  and ALP on the information (or at least more accurate than retrieved OMI  $\tau(550 \text{nm})$  from Rc(475 nm)) allows a better distinction of  $\tau$  and ALP effects on the  $O_2-O_2$  slant column density.

An accuracy better than 0.2 must be required on prior  $\tau$  information (cf. Fig. 8c). Indeed, a  $\tau(550 \mathrm{nm})$  bias of 0.25 can impact, in absolute, the retrieved ALP up to 50 hPa for  $\tau(550 \mathrm{nm})$  in the range of 0.6-1.0. For  $\tau(550 \mathrm{nm}) \geq 1.0$ , impact on ALP becomes almost null. Therefore, using MODIS  $\tau$  as prior to  $NN_{\tau,N_{02-02}^s}$  is likely expected to show retrieved ALP with a higher quality than with  $NN_{R_s,N_{02-02}^s}$ . Indeed, the current retrieved OMI  $\tau(550 \mathrm{nm})$  from  $Rc(475 \mathrm{nm})$  does not present a better accuracy than MODIS  $\tau(550 \mathrm{nm})$ .

Figure 9 depicts the box-whisker distribution of ALP precision  $\epsilon(\partial N_{O_2 - O_2}^s)$  due to  $N_{O_2 - O_2}^s$  precision. Estimations are obtained for fine and scattering particles ( $\alpha = 1.5$ ,  $\omega_0 = 0.95$ , g = 0.7).  $\epsilon(\partial N_{O_2 - O_2}^s)$  is obtained from the half of ALP

differences between adding and deducting uncertainties of the variables as follows:

$$\epsilon(\partial N_{\rm O_2-O_2}^{\rm s}) = \frac{1}{2} |ALP(N_{\rm O_2-O_2}^{\rm s} + \partial N_{\rm O_2-O_2}^{\rm s}) - ALP(N_{\rm O_2-O_2}^{\rm s} - \partial N_{\rm O_2-O_2}^{\rm s})|.$$
(9)

where  $\partial N_{\rm O_2 - O_2}^{\rm s}$  is the uncertainty applied to  $N_{\rm O_2 - O_2}^{\rm s}$ .  $\epsilon(\partial N_{\rm O_2 - O_2}^{\rm s})$  values are computed for all combinations of surface albedo 0.03-0.05-0.07 and  $\theta_0$ - $\theta = [25^{\circ}-25^{\circ}, 50^{\circ}-25^{\circ}, 25^{\circ}-45^{\circ}]$ . The reason to use this approach here is that, since  $N_{\rm O_2 - O_2}^{\rm s}$  precision is a random error (opposite to systematic), it will directly impact the retrieved ALP precision instead of leading to a systematic bias. A precision of  $N_{\rm O_2 - O_2}^{\rm s}$  lying in the range of 0.05-0.25  $10^{-43}$  mol2cm-5 results in similar ALP bias (i.e. at a first order, 2-7 % of  $N_{\rm O_2 - O_2}^{\rm s}$ ) results in ALP uncertainties between  $19 \pm 29$  hPa and  $57 \pm 31$  hPa on average for both NN configurations (see Fig. 5c). Note a temperature correction, for real observations, must be taken into account to correctly interpret  $N_{\rm O_2 - O_2}^{\rm s}$  (see Sect. 5.1).

[revised manuscript text omitted]
_{\rm c},N_{\rm O_2-O_2}^{\rm s}}$ ) to 0.7:1.9-1.9 km ( $NN_{\tau,N_{\rm O_2-O_2}^{\rm s}}$ ) depending on the season. When the OMLER is replaced by the MODIS black sky albedo database, the ALH variability continue continues to decrease of about 0.1 km -(cf. Fig. 16).

**6.4 Comparison of OMI aerosol layer height with LIVAS climatology**

The results of 3 years of OMI ALP retrievals over North-East Asia can be statistically compared to a climatology. The LIdar climatology of Vertical Acrosol Structure for space-based lidar simulation studies (LIVAS) is a 3-D multi-wavelength global aerosol and cloud optical database (Amiridis et al., 2015). This database provides averaged profiles of aerosol optical properties over 9 years (1 january 2007 - 31 December 2015) from the Cloud Aerosol Lidar and Infrared Pathfinder Satellite Observations (CALIPSO) data on a uniform grid of 1° x 1°. LIVAS addresses the wavelength dependency of acrosol properties for many laser operating wavelengths including 532 nm. LIVAS data set has been evaluated against AERONET 20 in Amiridis et al. (2015) showing realistic and representative mean state aerosol optical depth values in 532 making this data set ideal for synergistic use with other satellite products. Although the years of the OMI "climatology" and LIVAS do not strictly overlap, it is assumed that the average aerosol layer height (ALH) does not significantly change is assumed not to change significantly between both periods. The comparison is done per season. Spatial average of LIVAS ALH is done over the same area where retrievals are performed. Since large biases are expected at low  $\tau$ , only OMI retrievals acquired for 25 MODIS  $\tau(550 \text{nm}) > 1.0$  are taken into account and then spatially and temporally averaged per season. About 17 % in summer and spring, and between 5 % and 6 % in winter and autumn, of the OMI retrievals over the 3 years were th